**Use of water isotopes and chemistry to infer the type and degree of exchange between groundwater and lakes in an esker complex of northeastern Ontario, Canada**

Maxime P. Boreux[1*], Scott F. Lamoureux[1] and Brian F. Cumming[2,3]

[1]Department of Geography and Planning, Queen's University, Kingston, K7L3N6, Canada

[2]Department of Biology, Queen's University, Kingston, K7L3J9, Canada

[3]School of Environmental Studies, Queen's University, Kingston, K7L3J9, Canada

[*]Corresponding author: m.boreux@queensu.ca

## **Abstract**

While interactions between groundwater and lake-water influence water chemistry, water balance, aquatic organisms, biochemical cycles and contamination levels, they remain a poorly-studied component of lake hydrology. Identifying the controls of groundwater and lake-water interactions at the landscape level and classifying lakes into different categories based on their degree of interaction with the groundwater can provide insights on a lakes' sensitivity and vulnerability to environmental stressors. Such information can also provide baseline conditions for comparison to future changes that are important for water management and conservation. To this end, water chemistry and water isotopic composition were investigated in a set of 50 boreal lakes located at different elevations in an esker system near Timmins, Ontario. Analyses focused on stable isotopic ratios of hydrogen and oxygen, and specific conductance as indicators of the position of a lake with respect to the influence of groundwater. Both isotopic composition and specific conductance distinguished higher-elevation groundwater-recharge lakes from lower-elevation groundwater-discharge lakes. Groundwater-recharge lakes were high elevation lakes characterized by enriched isotopic values and low values of specific conductance. In contrast, groundwater-discharge lakes were isotopically depleted and had higher values of specific conductance, and occurred at lower elevations. An intermediate group of lakes was also defined (termed seepage lakes) and had intermediate isotopic and water chemistry characteristics compared to recharge and discharge lakes. Differences in water geochemistry between field campaigns revealed that upland groundwater-recharge lakes showed evidence of evaporative drawdown, indicating sensitivity to short-term changes in climate, whereas the lower-

elevation groundwater-discharge lakes showed little variation between seasonal samples, and consequently
would likely be affected only by hydroclimatological changes of greater duration and magnitude.
**Keywords**
Water balance, landscape limnology, groundwater, hydrochemistry, lake-water chemistry, stable isotopes
**1 Introduction**
Nearly all surface waters, including lakes, interact with groundwater to some extent (Winter et *al*.,
1998; Winter, 1999; Cohen et *al*., 2016). The degree to which lakes interact with groundwater strongly
influences lake-water chemistry and water balance as well as aquatic biota and biochemical cycles
(Lewandowski et *al*., 2015; Rosenberry et *al.*, 2015). The degree of hydrological connectivity will influence
the sensitivity of lakes to environmental stressors such as climate change, and other anthropogenic
disturbances. Future climate change will likely affect lakes differently depending on groundwater-lake
interactions, further increasing our need to better understand the relations between lake hydrology and lake-
water geochemistry. Moreover, understanding the climatic controls on water balance is essential for
informed ecosystem management and conservation practices (Okkonen and Kløve, 2011).

Interactions between groundwater and lakes are an often poorly-studied component of lake
hydrology. However, recent advances in isotopic techniques and modelling approaches have enabled
researchers the opportunity to better understand hydrological processes in lakes at a local to regional scale
(Fleckenstein et *al.*, 2010; Rosenberry et *al*., 2015). This has included studies using groundwater modelling
(e.g. Winter, 1976; Smerdon et *al*., 2007; Okkonen and Kløve, 2011; Ala-aho et *al*., 2015a; Ala-aho et *al*.,
2015b), as well as empirical studies at a range of different spatial and temporal scales. Studies at the regional
scale (e.g. Gorham et *al*., 1983; Webster et *al*., 2000) emphasize differences in climate and surface geology
as being the important drivers of changes in lake-water chemistry. By contrast, empirical studies performed
at a local to regional scale (e.g. Webster et *al*., 1996; Quinlan et *al*., 2003; Martin and Soranno, 2006; Ala-
aho et *al*., 2013; Thorslund et *al*., 2018) indicate the importance of landscape position and also lake
morphology as being important in understanding lake-groundwater interactions. At the scale of an
individual lake, a number of studies have shown that interactions between groundwater and lakes can vary
temporally according to changes in seasonality and longer-term changes in hydroclimatic conditions (e.g.
Kenoyer and Anderson, 1989; LaBaugh et *al*., 1997; Sebestyen and Schneider, 2001; Schuster et *al*., 2003;
Smerdon et *al*., 2005; Arnoux et *al*., 2017b).
The vast majority of past empirical studies have relied on studying aspects of water chemistry (e.g.
dissolved ions), as an indicator of hydrological connectivity between lakes, because such approaches can
reveal signals of important processes such as mineral weathering and dissolution (Bertrand et *al*., 2014).
Chemical tracers are often referred to as non-conservative tracers because their composition changes as
they react with catchment materials (Kendall and McDonnell, 1998). By contrast, other studies rely on the
isotopic composition of water as a hydrological tracer (e.g. Turner et *al*., 2010; Isokangas et *al*., 2015). The
latter are labelled as conservative tracers because they are relatively conservative in reactions with
catchment materials and retain their distinctive values until they mix with other water sources or they
evaporate (Kendall and McDonnell, 1998). In addition to being a good indicator of source water, stable
isotopes of water constitute an efficient and cost-effective means to quantify lake evaporation and water
balance of water bodies using the Craig-Gordon model (Craig and Gordon, 1965), given that the isotopic
composition of lake-water and precipitation as well as air temperature and relative humidity are known, or
can be estimated (Gibson and Edwards, 2002). The Craig-Gordon model is well established and has been
used extensively to investigate the spatial and temporal variability in lake-water balance in remote boreal
regions of Canada (e.g. Wolfe et *al*., 2007; Bouchard et *al*., 2013; Tondu et *al*., 2013; Turner et *al*., 2014a;
Turner et *al*., 2014b). However, precipitation and groundwater often display similar isotopic signatures as
they both tend to retain their original isotopic composition because they undergo little to no evaporation
(Gibson and Edwards, 2002; Gibson et *al*., 2008; Yi et *al*., 2008). This makes the distinction of the relative
influence of groundwater and precipitation in lake-water balance challenging. Given this, the combination
of chemical and isotopic approaches has the ability to produce more reliable interpretations, especially if
the two approaches converge on a mutually reinforcing interpretation. Chemical and isotopic tracers have
been widely used together to investigate the connectivity between groundwater and lake water within a
single lake (e.g. Labaugh et *al*., 1997; Schuster et *al*., 2003; Rautio and Korkka-Niemi, 2011) or for a cluster
of a few selected lakes (e.g. Gurrieri and Furniss, 2004; Katz et *al*., 1997; Turner and Townley, 2006;
Arnoux et *al*., 2017a; Arnoux et *al*., 2017b). Nonetheless, studies that have combined chemical and isotopic
approaches to investigate the connectivity between groundwater and lake water at the landscape level and
for a large number of lakes in a region are lacking.

The main objective of this study is to examine the importance of landscape position on groundwater-
lake connectivity (*i.e.* the exchange of water between the groundwater system and the lakes) by scrutinising
both water chemistry and isotopic composition of water in a boreal esker complex in northeastern Ontario.
The use of water tracers was preferred to direct measurements because tracers: (i) have proven to be good
indicators of interactions between groundwater and lake water; and (ii) constitute a time and cost-effective
approach that can be applied at a large spatial scale. In this paper, the term "water tracer" defines natural
indicators that provide information on water sources (conservative tracers) and water pathways (non-
conservative tracers). Investigating such interactions in the context of esker hydrology is particularly
relevant as eskers consist of porous and permeable materials that facilitate groundwater flows, are
widespread in boreal regions (Smerdon et *al*., 2005; Ala-aho et *al*., 2015b) and constitute one of the most
common type of aquifers for community water supplies in boreal regions of the globe (Cloutier et *al*., 2007;
Okkonen and Kløve, 2011; Rey et *al*., 2018). Results from this study will be used to develop a lake typology
(*i.e.* a classification or generalisation of lakes into different categories) of hydrological connectivity
(Newton and Driscoll, 1990; Bertrand et *al*., 2014). The resultant typology will provide insights on lakes'
sensitivity to environmental stressors over time such as climate change, acidification or pollution by
accounting for variation among lake types through continuous monitoring. This classification will also
improve and simplify water management and conservation goals as each lake type requires the similar
management strategies in a region where cottage development, recreational fishing, forest operations,
mining activities and aggregate extraction is prevalent (Cochrane, 2006; Rey et *al*., 2018). Additionally,
the typology will provide an important baseline for comparison to future hydrological regimes that may
altered them and be used for site-selection and interpretations of past hydrological changes from
stratigraphic analysis of isotopic and geochemical indicators in sediment cores from lakes. Finally, the
sensitivity of the proposed typology will be assessed by investigating if short-term variations of lake-water
characteristics are more readily detected in higher-elevation groundwater-recharge lakes (*i.e.* lakes that
receive the majority of their water from precipitation and feed the groundwater system) in comparison to
lower-elevation groundwater-discharge lakes (*i.e.* lakes that receive the majority of their water from
groundwater).
**2 Study area**
The study area is a portion of the Kettle Lake Esker between the southern shore of Frederick House
Lake, and the northern shore of Night Hawk Lake, a region located approximately 35-km east of Timmins,
Ontario, Canada (**Fig. 1a**). The Timmins region is characterized by a humid continental climate (dfb in the
Köppen climate classification) with a mean annual temperature of 1.8°C and average precipitation of 835
mm (Environment Canada, 2015). This region has long cold winters and lakes are covered with ice from
early November until early April. Summers are usually wet and mean air temperatures are 17.5°C in July.
Many of the study lakes and streams are located within Kettle Lakes Provincial Park, as well as in lower-
elevation regions that have been moderately influenced by human activities (Cochrane, 2006). The study
region is covered with well-drained orthic humo-ferric podzols, while surrounding clay plains are for the
most part covered with moderately to poorly drained orthic gray luvisols and gleyed gray luvisols
(OMNDM, 2006). Jack pine, poplar, black spruce, white birch, trembling aspen and balsam fir dominate
the well-drained areas, with a dominance of spruce in poorly drained regions.
The regional landscape is dominated by landforms and deposits created by the Laurentide Ice Sheet
during the last glacial maximum and subsequent deglaciation approximately 10,000 years ago (Dyke,
2004). The ablation of the ice sheet was particularly dynamic and led to the formation of relatively large
eskers composed of long sinuous ridges of coarse grained glaciofluvial sediments in deposits oriented in a
north-south direction and mantling the crystalline bedrock (Cloutier et *al*., 2007; Rey et *al*., 2018). The
retreat of the ice sheet was accompanied by ponding of glacial meltwaters that led to the development of
glacial Lake Ojibway that submerged most of the region (Roy et *al*., 2011) and the widespread deposition
of glaciolacustrine clay, followed by the drainage of Lake Ojibway into Hudson Bay ~8,200 years ago (Roy
et *al*., 2011). As Lake Ojibway levels dropped, wave action eroded the surface of the esker and redistributed
some sand materials on the flanks on the esker, forming lateral littoral sand units that drape the
glaciolacustrine clays (Cloutier et *al*., 2007; Rey et *al*., 2018) (**Fig. 1b**). The numerous kettle lakes on the
esker formed as glacial ice that was trapped in the outwash materials melted.
The esker stratigraphy ensures that its groundwater system is highly localized because the esker
generally have a high hydraulic conductivity due to their coarse texture. The esker is surrounded by
underlying bedrock and the adjacent fine-grained glaciolacustrine deposits, both characterized by very low
hydrological conductivity (Stauffer and Wittchen, 1992). As a consequence, the esker can be conceptually
partitioned into zones of unconfined aquifers in its center where coarse material is present at the surface
and zones of confined aquifers at its edges when fine-grained sediment mantles the core of the esker (**Fig.
1b**) (Cloutier et *al*., 2007; Rey et *al*., 2018). Thus, the recharge of the esker will occur through infiltration
of precipitation in the unconfined aquifer and discharge will take place on the esker flanks at the contact of
the clay, where most groundwater springs emerge (Cloutier et *al*., 2007; Rey et *al*., 2018). Confined aquifers
found on the surrounding clay plain are often covered by peatlands and shallow lakes fed by groundwater
springs on the edges of the esker or by streams that drain the esker (Rossi et *al*., 2012).

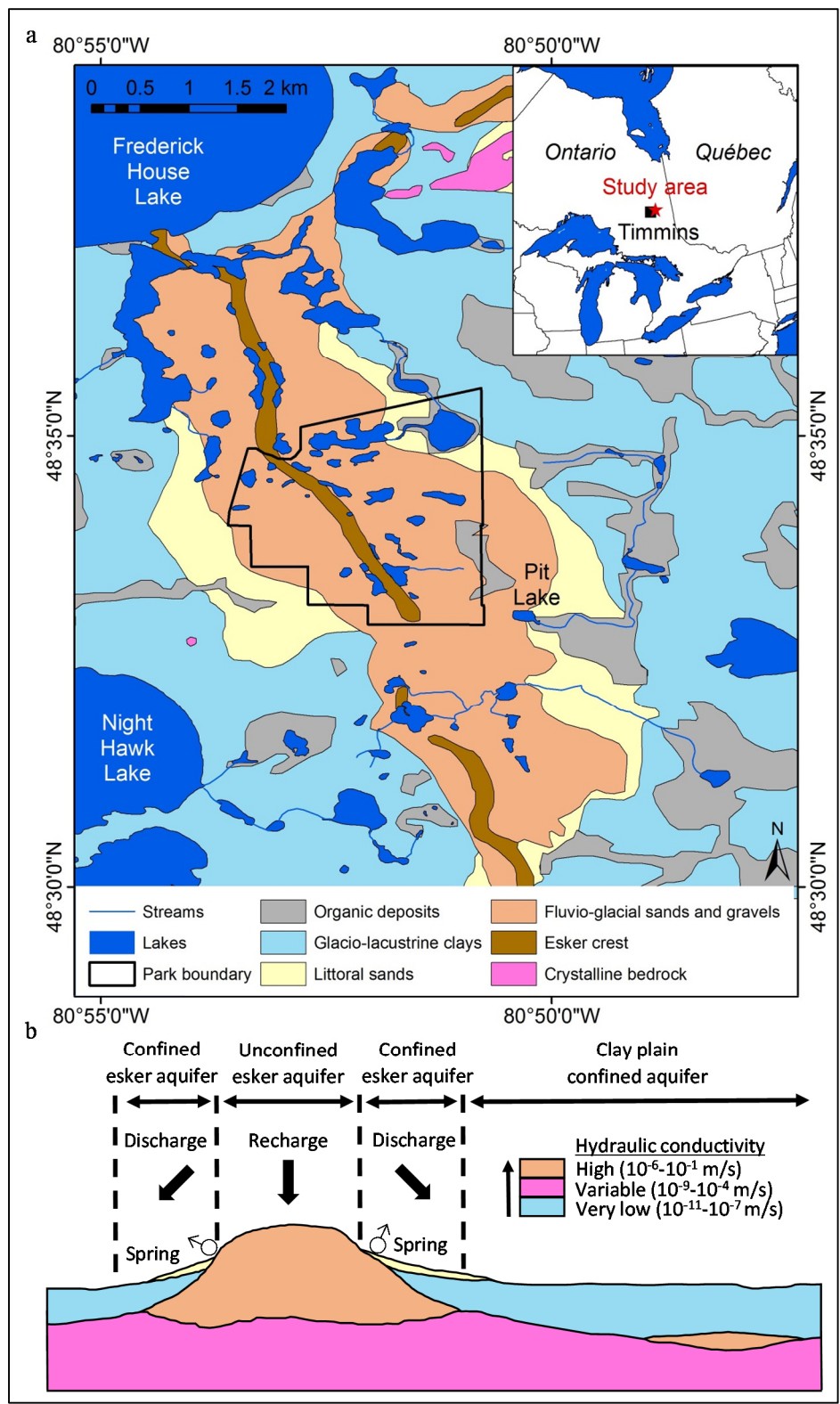

**Fig. 1**: Location of the study area in north east Ontario near Timmins, Ontario (Based on: Richard and McClenaghan, 2000) (a) and conceptual geological transverse section of an esker complex at latitude 48°35'0"N (b) (Figure modified from: Veilette *et al*., 2004, Hydraulic conductivity values from: Cloutier et *al*., 2013).

## 3 Methods

### 3.1 Geomatic and lake morphometric data

Lakes and other geographic features were digitized from Google Earth using the imagery dating from 7/26/2005. Maximum lake depths $Z_{max}$ were obtained from the Ontario Ministry of Natural Resources and Forestry (OMNRF, 2011). Digital Elevation Models (DEM) from the study area were obtained from Natural Resources Canada with a spatial resolution of 20 m (NRC, 2013). Lake elevation values were calculated as the mean elevation of the lake in ArcGIS 10.3 from the available DEM, and a proxy of lake watershed slopes was calculated from the DEM in ArcGIS in a 100 m buffer zone surrounding each lake and termed perimeter slope. This approach was used because of the coarse spatial resolution of the DEM and the close proximity of the lakes made clear individual lake watershed delineation impossible. Because all lakes in the study area are kettle lakes, which are characterized by steep shore slopes, initial buffer zones of different widths were produced. The buffer width of 100 m was chosen as this distance showed the best correlation with water tracers.

### 3.2 Field measurements and water sample collection

Water samples were collected during three field campaigns (7-14 June 2013, 16-23 June 2014, and 12-19 August 2014). 50 lakes were sampled (29, 28 and 50 lakes during the June 2013, June 2014 and August 2014 campaign respectively), as well as a number of streams (lake outlets and lake inlets) and groundwater springs. The lakes were chosen to achieve a wide spatial coverage over this region encompassing a gradient of lake elevation as well as lake types. A few lakes located in the area of interest could not be sampled due to poor accessibility. Groundwater springs and streams flowing into the study lakes were also sampled to characterize the interactions between local groundwater and lakes.

Temperature and specific conductance of the water (corrected to 25°C) at each site were measured with an YSI Salinity Conductivity Temperature meter (accuracy ±0.1 μS/cm and ±0.1°C). The pH was measured with an UP-5 Denver Instrument pH meter calibrated before usage (accuracy ±0.1). Water samples were collected using 1 L Nalgene bottles that were tripled-rinse with distilled water prior to use and again with sample water while sampling (Louiseize et *al*., 2014). To prevent cross-contamination, the tripled-rinsing of bottles with sampling water was carried away from the final sampling point in the lake or downstream of stream/spring sampling points. Bottles were completely filled in order to avoid headspace air and subsequently filtered within 3 hours of sampling.

Samples collected for water stable isotope, dissolved ion analysis and dissolved inorganic nitrogen
were vacuum filtered with 0.45μm Millipore Isopore filters. Vacuum filtering apparati were triple-rinse
with distilled water and the lake water before and after use for each sample. The filtrates were then poured
and placed into pre-cleaned 20ml plastic scintillation vials. Three replicates were collected for each sample.
All vials were previously rinsed with filtered sample water that was discarded. Vials were completely filled
to avoid any headspace, sealed with Parafilm™ to avoid evaporation, and kept in a cool and dark fridge
prior to analysis.

Samples for total dissolved nitrogen (TDN), dissolved organic carbon (DOC) and dissolved inorganic
carbon (DIC) analysis were filtered on-site using a glass filtration and Whatman GF/F glass fiber filters to
avoid any *ex situ* carbon addition. Filters were pre-ashed at 550°C for two hours and wrapped with pre-
ashed aluminum foil prior to utilization to eliminate any residual organic matter (Lamoureux and
Lafrenière, 2014). The glass filtration apparatus was triple-rinsed with distilled water and lake water before
and after use for each sample and was wrapped with new clean pre-ashed aluminum foil overnight. Filtrates
were then poured and stored into pre-cleaned 45-ml amber EPA vials with Teflon-lined septa with no
headspace (Louiseize et *al*., 2014). Two replicates were collected for each sample. Vials were labelled and
were kept cool and in the dark in a fridge prior to analysis.

*3.3 Laboratory analysis*

The stable isotope ratios of water ($\delta^{18}O$ and $\delta^2H$) were measured using a Los Gatos Research Liquid–
Water Isotope Analyzer (LGR) in the FaBRECC laboratory at Queen's University, which vaporizes injected
sample and measures its absorbance relative to Vienna Standard Mean Ocean (‰ V-SMOW). All runs
contain 6 replicate analyses and 3 standards produced by Los Gatos bracketing every 3 samples (e.g.
LGR1A, $\delta^{18}O$=-19.50‰ and $\delta^2H$=-154.3‰; LGR2A, $\delta^{18}O$=-16.14‰ and $\delta^2H$=-123.6‰; LGR3A, $\delta^{18}O$=-
13.10‰ and $\delta^2H$=-96.4‰; LGR4A, $\delta^{18}O$=-7.69‰ and $\delta^2H$=-51.0‰; LGR5A, $\delta^{18}O$=-2.80‰ and $\delta^2H$=-
9.5‰). Sample reproducibility (1σ) was based on repeated measurements of samples and standards fixed
at 0.25‰ for $\delta^{18}O$ and at 1.5‰ for $\delta^2H$.

Concentrations of inorganic ions (Ca, Mg, K, Na, Cl, $SO_4$) were measured by liquid ion
chromatography with a Dionex ICS-3000. Detection limits were as follows: 0.5 ppm for Ca, 0.01 ppm for
Mg, 0.2 ppm for K, 0.3 ppm for Na, 0.05 ppm for Cl and 0.1 ppm for $SO_4$). Concentrations of dissolved
inorganic species ($NO_2$-$NO_3$ and $NH_4$) were measured by colorimetry using an Astoria Pacific FASPac II
Flow Analyser (detection limits of 0.01 ppm). Concentrations of DOC, DIC and TDN were measured by

high-temperature combustion and nondispersive infrared sensor and chemiluminescent detection using a Shimadzu TOC-VPCH/TNM equipped with a high-sensitivity catalyst system (detection limits of 0.08 ppm for DOC and 0.015 ppm for TN) (Louiseize et al., 2014). Total dissolved carbon (TDC) was calculated as the sum of DOC and DIC.

### 3.4 Water balance calculations

Monthly precipitation isotopic data are available from February, 1997 to November, 2010. Precipitation isotopic data were collected at Bonner Lake, about 125-km NW of the study area by the Canadian Network for Isotopes in Precipitation (CNIP) (Birks et al., 2010). The general water (Eq. 1) and isotope balance (Eq. 2) of a well-mixed lake may be written respectively as follow (Darling et al., 2005):

$$\frac{dV}{dt} = I - Q - E \qquad \text{Eq. 1}$$

$$\frac{V d\delta_L + \delta_L dV}{dt} = I\delta_I - Q\delta_Q - E\delta_E \qquad \text{Eq. 2}$$

where $V$ is the volume of the lake, $t$ is time, $dV$ is the change of volume over time $dt$, $I$ is instantaneous inflow where $I = I_F + I_G + P$ ($I_F$ being surface inflow, $I_G$ groundwater inflow and $P$ precipitation on the lake surface); $Q$ is instantaneous outflow where $Q = Q_R + Q_G$ ($Q_R$ is surface outflow and $Q_G$ is groundwater outflow), $E$ is evaporation; and $\delta_L$, $\delta_I$, $\delta_Q$ and $\delta_E$ are the isotopic compositions of the lake, inflow, outflow and evaporative flux respectively. Assuming (i) that the lake maintains a near-constant volume on the long-term (i.e., $dV = 0$ and $dt \to \infty$) (Darling et al., 2005), and (ii) that physical outflow does not cause isotopic fractionation (i.e., $\delta_Q = \delta_L$) (Gibson and Edwards, 2002; Yi et al., 2008), Eq. 1 and 2 can be simplified and rewritten as follows:

$$I = Q + E \qquad \text{Eq. 3}$$

$$I\delta_I = Q\delta_Q + E\delta_E \qquad \text{Eq. 4}$$

$E$ can be related to $I$ assuming that the lakes are in isotopic steady state (i.e. undergoing evaporation while maintaining constant volume). This assumption seems well justified as most of the lakes have had sufficient time in the past to reach their isotopic steady-state which is reflective of the local climate and their mean

hydrological status, and can be defined by its water balance, which corresponds to the ratio of the total
inflow to the evaporation rate (Isokangas et *al*., 2015). The evaporation-to-inflow ratio of the lake $E_L/I_L$ can
be calculated by combining Eq. 3 and 4 (Gibson and Edwards, 2002; Yi et *al*., 2008):

$$\frac{E_L}{I_L} = \frac{\delta_I - \delta_L}{\delta_E - \delta_L} \qquad \text{Eq. 5}$$


where $\delta_I$ was computed as the intersection of the Local Meteoric Water Line (LMWL) with the Local
Evaporation Line (LEL) (Gibson et *al*., 1993; Yi et *al*., 2008); $\delta_L$ is the isotopic composition of the lake
water sample and $\delta_E$ was estimated using the Craig-Gordon model (Craig and Gordon, 1965) formulated
by Gonfiantini (1986) as follows:

$$\delta_E = \frac{(\delta_L - \varepsilon^*)/\alpha^* - h\delta_A - \varepsilon_k}{1 - h + \varepsilon_k} \qquad \text{Eq. 6}$$


where $\delta_L$ is the isotopic composition of lake water, $\varepsilon^*$ is the equilibrium isotopic separation term, $\alpha^*$ is the
liquid–vapour equilibrium fractionation factor, $h$ is the relative humidity, $\delta_A$ is the isotopic composition of
the local atmospheric moisture, and $\varepsilon_k$ is the kinetic separation term between the liquid and vapour phases.
The $\varepsilon^*$ and $\alpha^*$ parameters which are temperature dependent can be calculated using empirical equations for
$\delta^{18}O$ as follows (Horita and Wesolowski, 1994):

$$\varepsilon^* = -7.685 + 6.7123 \left(\frac{10^3}{T}\right) - 1.6664 \left(\frac{10^6}{T^2}\right) + 0.35041 \left(\frac{10^9}{T^3}\right) \qquad \text{Eq. 7}$$


$$\alpha^* = exp\left(-\frac{7.685}{10^3} + \frac{6.7123}{T} - \frac{1666.4}{T^2} + \frac{350410}{T^3}\right) \qquad \text{Eq. 8}$$


where T is the air temperature in Kelvins. $\varepsilon_k$ (Eq. 9) is expressed for $\delta^{18}O$ by (Gonfiantini, 1986):

$$\varepsilon_k = (0.0142\,(1 - h))1000 \qquad \text{Eq. 9}$$


The equation for $\delta_E$ was modified according to Gibson and Edwards (2002) to directly utilize isotopic data
in per mil rather than as a decimal fraction and expressed as follows:

$$\delta_E = \frac{\alpha^* \delta_L - h\delta_A - \varepsilon}{1 - h + 10^{-3}\varepsilon_k} \qquad \text{Eq. 10}$$


where $\varepsilon$ is the total isotopic separation factor that includes both $\varepsilon^*$ and $\varepsilon_k$ expressed as:

$$\varepsilon = \varepsilon^* + \varepsilon_k \qquad \text{Eq. 11}$$


$\delta_A$ was originally estimated with the original model that assumes isotopic equilibrium between atmospheric
moisture and precipitation as follows (Gibson, 2002):

$$\delta_A = \frac{\delta_P - \varepsilon^*}{1 + 10^{-3}\varepsilon^*} \qquad \text{Eq. 12}$$


where $\delta_P$ was computed as the average isotopic composition of annual precipitation from February 1997 to
November 2010 (data collected by CNIP). The same procedure was used to calculate the evaporation-to-
inflow ratio of streams and groundwater springs for comparison, although the assumptions of the
methodology can only be applied to lakes.

However, the hypothesis of steady-state may not be valid for some small lakes that undergo

significant changes in lake levels as evaporation progresses. For those lakes in non-steady state (here
defined as non-alkaline high-elevation lakes), the evaporative loss fraction of the lake volume (f) was
calculated using the original equation of Gonfiantini (1986) rearranged as follows (Skrypek et *al*., 2015):

$$f = 1 - \left[\frac{\delta_L - \delta^*}{\delta_p - \delta^*}\right]^{\frac{1}{m}} \qquad \text{Eq. 13}$$

where $\delta_p$ is the initial value of water in the lake that undergoes evaporation, $\delta_L$ the final value of water in
the lake that undergoes evaporation, $\delta^*$ is the limiting isotope enrichment factor defined as follows (Skrypek
et *al*., 2015):

$$\delta^* = \frac{h\delta_A + \varepsilon}{h - \frac{\varepsilon}{1000}} \qquad \text{Eq. 14}$$

and *m* is the enrichment slope defined as follows (Skrypek et *al*., 2015):

$$m = \frac{h - \frac{\varepsilon}{1000}}{h + \frac{\varepsilon_k}{1000}}$$ Eq. 14


While E/I ratios and f values are two metrics calculated with different formulas, they can be compared
equivalently as they both represent the mass balance of the lakes in dimensionless ratios of water losses
versus available lake water.

*3.5 Numerical analysis*

Linear regressions were used to assess the degree of co-variability between quantitative variables
while logistic regressions were utilized to assess the relations between binary variables and quantitative
variables at the 0.05 level. Breakpoint analysis or segmented regression was used to detect any change of
trends in water tracers along an elevation gradient and to produce subsequent higher-level groupings of
lakes. Breakpoints that were significant at the 0.05 level were averaged to obtain the elevation of the
"breakpoint line". A non-metric multidimensional scaling (NMDS) was run to assess the differences among
lake types in a 2-dimensional ordination space using non-scaled values of electrical conductance, Ca, $\delta^{18}O$
and $\delta^{2}H$ as input variables, and Euclidean distance as a measure of dissimilarity. A Wilcoxon signed-rank
test was subsequently applied as a post-hoc analysis for all lake-water variables that were above detection
limits to determine if differences among the different types of lakes were statistically significant at the 0.05
level as most of the Shapiro-Wilk test for normality revealed that most variables were not normally
distributed. An analysis of similarity (ANOSIM) was also carried out as a complement to determine if
within group similarity was significantly greater than in-between group similarity at the 0.05 level. All
statistical analyses were performed in *R* 3.4 on the data from the August 2014 campaign as it was the one
with the most samples.

**4 Results**

*4.1 Temperature and water stable isotopes*

During the August 2014 field campaign, all types of bodies of water differed from one another by
their temperature: springs are characterized by low temperatures (6 -12°C), streams have slightly higher
temperatures (12 -16°C), and lakes ranged from (16 -18°C). It should be noted however that spring
temperature is not an accurate indication of groundwater temperature due to alterations induced by the
velocity of the discharging spring and the resulting warming of the discharging water as it reaches the
surface.

Monthly precipitation isotopic data from CNIP from February, 1997 to November, 2010 from Bonner
Lake, about 125-km NW of the study area, show progressive enrichment in values between winter, spring,
fall and summer on the global and local meteoric water lines, which are similar (**Fig. 2a**). The water samples
(*i.e.* groundwater springs, streams and lakes) displayed a wide range of isotope values (-14.7‰ to -6.8‰
for $\delta^{18}O$ and -105.5‰ to -68.6‰ for $\delta^2H$), which fell on a Local Evaporation Line (r = 0.99, p < 0.001)
(**Fig. 2b**). Groundwater springs have isotopic values similar to mean annual precipitation (-14.7‰ to -
13.1‰ for $\delta^{18}O$ and 105.5‰ to -96.0‰ for $\delta^2H$) while being more depleted as summer precipitation is more
enriched in heavy isotopes than groundwater (**Fig. 2a**). Streams have comparable isotopic composition to
groundwater springs (-13.8‰ to -11.3‰ for $\delta^{18}O$, and -100.8‰ to -91.9‰ for $\delta^2H$). By contrast, lakes are
characterized by large variations in water isotopic composition, ranging from values comparable to
groundwater springs (*ca.* -14‰ for $\delta^{18}O$ and -100‰ for $\delta^2H$) to more enriched values (*ca.* -6‰ for $\delta^{18}O$
and -70‰ for $\delta^2H$).

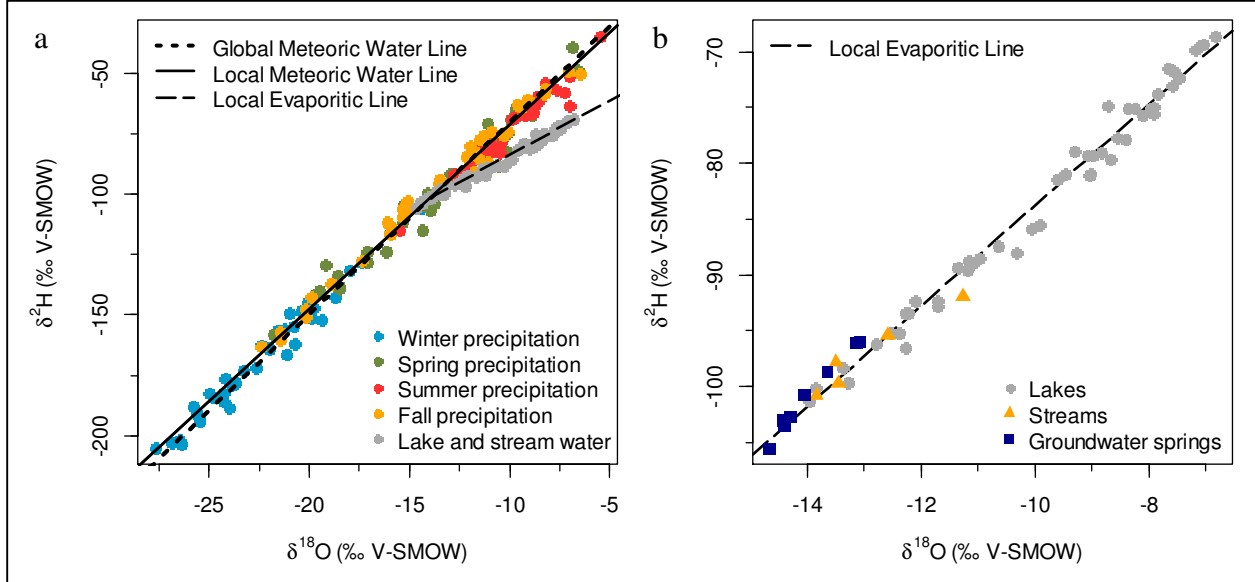

**Fig. 2**: Isotopic composition of precipitation in relation to the Global Meteoric Water and Local Meteoric
Water lines (r = 0.99, n = 166, *p* < 0.001) (a), and isotopic composition of collected water samples in relation
to the Local Evaporation Line (r = 0.99, n = 68, *p* < 0.001) (b).

A significant correlation exists between $\delta^{18}O$ and elevation (r = 0.53, n = 50, *p* < 0.001) (**Fig. 3a**),
suggesting that elevation is an important variable explaining the isotopic composition of water in the study
lakes. Lakes sampled at lower elevations are more depleted while lakes sampled at higher elevations are
enriched in $^{18}$O and $^2$H. Another significant correlation occurs between $\delta^{18}$O and the steepness of the slopes
surrounding the lake (r = -0.33, n = 48, *p* = 0.02), which suggests that morphometric factors may also
influence lake-water balance.

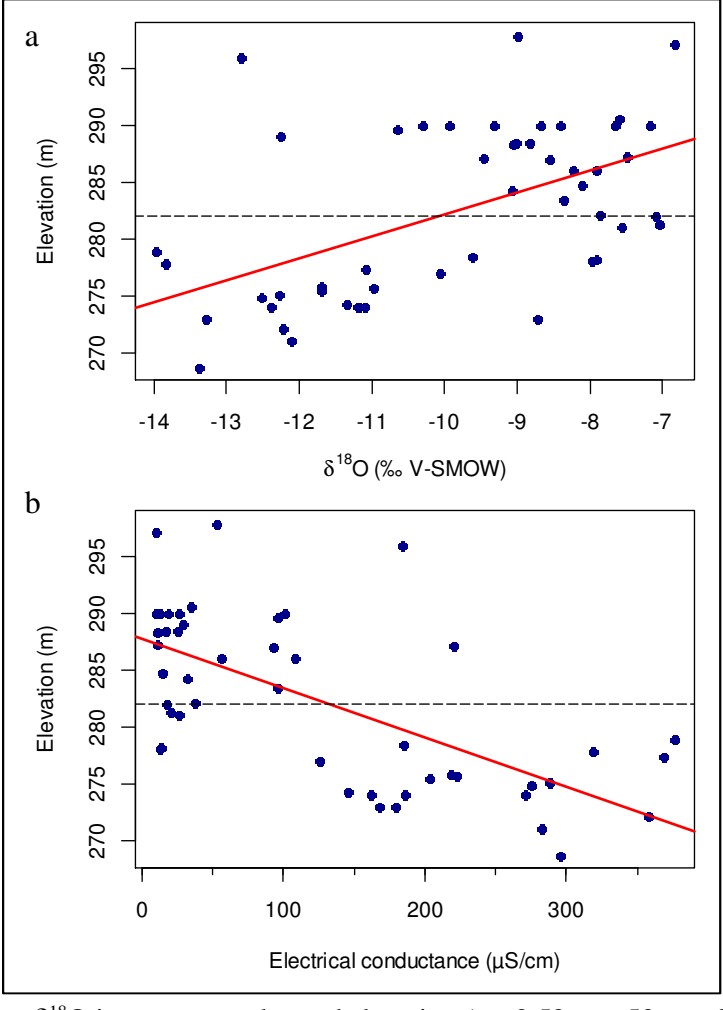


**Fig. 3**: Relation between $\delta^{18}$O in water samples and elevation (r = 0.53, n = 50, *p* < 0.001) (a), and specific
conductance and elevation (r = -0.67, n = 50, *p* < 0.001) (b). The horizontal dashed line indicates the position
of a breakpoint analysis which occurs at an elevation of ~282 m a.s.l. in both cases (**Tab. 1**).

Evaporation-to-inflow ratio calculations (E/I) and evaporative loss fractions of the lake volume (f)
show that groundwater springs and streams have E/I ratios close to 0 due to their short residence times
while lakes have E/I or f values ranging from values similar to groundwater springs and streams to near E/I
or f ~ 1 owing to their longer residence times that expose them to evaporation (**Fig. 4a**).

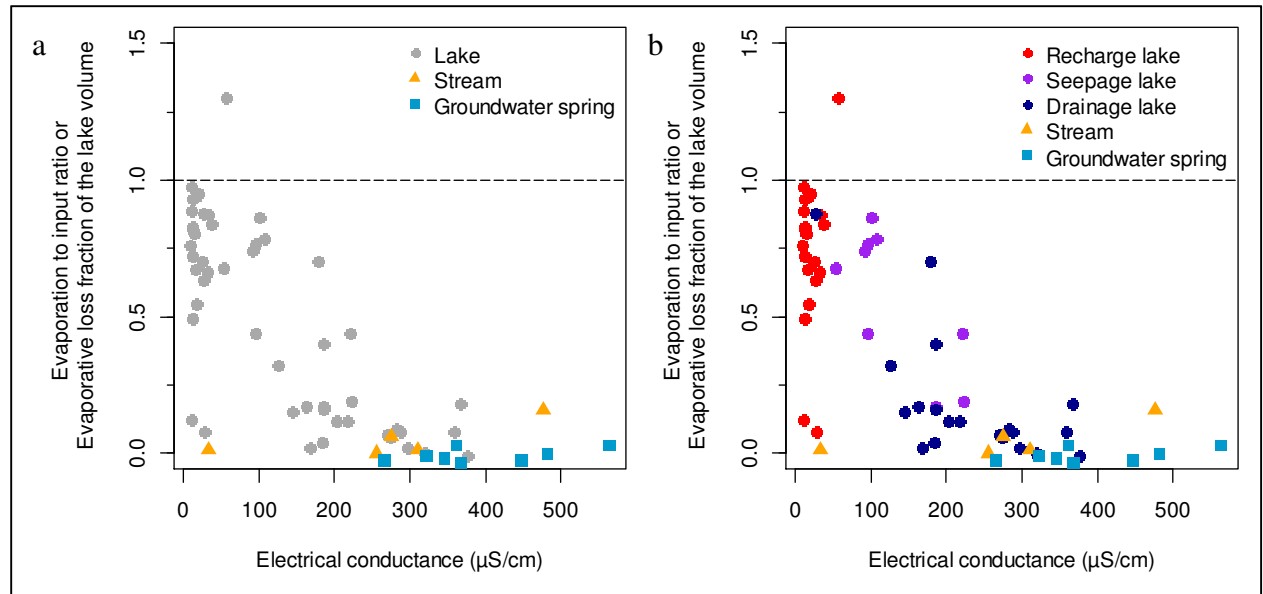

**Fig. 4**: Relation between electrical conductance and calculated evaporation-to-inflow ratios. Lakes in panel b are coded according to a classification scheme developed in this paper.

*4.2 Solutes and dissolved organic matter*

As is the case with stable isotope values, water bodies reveal a wide range for non-conservative ions, and the overall sum of ions indicated by lake-water specific conductance, all of which are significantly correlated (see correlation matrix **Tab. A1**). Groundwater springs have the highest specific conductance (300-550 µS/cm) while streams have values *ca.* 300 µS/cm. Lakes show a wide range of solute content and values of specific conductance, from values similar to groundwater springs (*ca.* 400 µS/cm) to very low values similar to precipitation (as low as 10 µS/cm) (**Fig. 3b**). A significant correlation exists between the specific conductance and elevation (r = -0.67, n = 50, *p* < 0.001), suggesting that elevation is an important variable explaining specific conductance in lakes. There was also a significant relationship between the specific conductance and the ratio of perimeter to surface area of the lakes (r = 0.48, n = 48, *p* < 0.001), which also suggests that lake morphology may also be important.

Only 36 lakes were sampled for dissolved organic carbon (DOC) and nitrogen (TN). Unlike non-conservative ions and conservative isotopic tracers, no significant correlations were found between dissolved organic elements or TN and elevation (r = -0.04, n = 36, *p* = 0.84 for DOC; r = 0.05, n = 36, *p* = 0.77 for TN). However, significant, or marginally significant correlations were observed between DOC and perimeter to area ratio (r = 0.54, n = 33, *p* < 0.001) with more elongated lakes having higher concentrations of DOC, and between DOC and mean lake depth, with deeper lakes having lower concentrations of DOC

(r = -0.58, n = 25, *p* = 0.11). A similar pattern was observed between TN and perimeter to area ratio (r =
0.60, n = 33, *p* = 0.002) and between TN and mean lake depth (r = -0.71, n = 25, *p* = 0.02).

*4.3 Correlations between water tracers*

There is a strong and significant correlation between lake-water isotopic values and specific
conductance (r = 0.80, n = 50, *p* < 0.001). However, the slopes of the linear regressions for the water $\delta^{18}O$
(**Fig. 3a**) and specific conductance (**Fig. 3b**) do not match the data points perfectly as there seem to be a
distinct transition between similar values found in higher- and lower-elevation lakes, which was further
examined using a breakpoint analysis of the lake water properties (the later undertaken to detect any step-
wise changes in trends). Nine available environmental variables had a statistically significant breakpoint
(*i.e.* an era of the line where the relationship between the variables changes trends) when regressed over
elevation and significant breakpoints were within a narrow range of elevation with a mean of 282.4 m a.s.l.
(**Tab. 1**).

**Tab. 1**: Results showing significant breakpoints in nine water chemistry variables and lake elevation (lower
and upper elevation ranges represent the standard deviation).

| Environmental Variable | Mean Elevation | Lower Elevation | Upper Elevation |
|---|---|---|---|
| $\delta^{18}O$ | 282.2 | 280.2 | 284.2 |
| $\delta^2H$ | 282.0 | 279.9 | 284.1 |
| *d** | 282.2 | 280.3 | 284.1 |
| EC | 284.0 | 281.4 | 286.6 |
| $Ca^{2+}$ | 284.2 | 281.9 | 286.5 |
| $Mg^{2+}$ | 282.0 | 280.3 | 283.7 |
| $K^+$ | 281.8 | 278.2 | 285.4 |
| DIC | 281.5 | 279.6 | 283.4 |
| TC | 281.8 | 280.2 | 283.4 |
| Breakpoint line | 282.4 | | |

*d* corresponds to deuterium excess

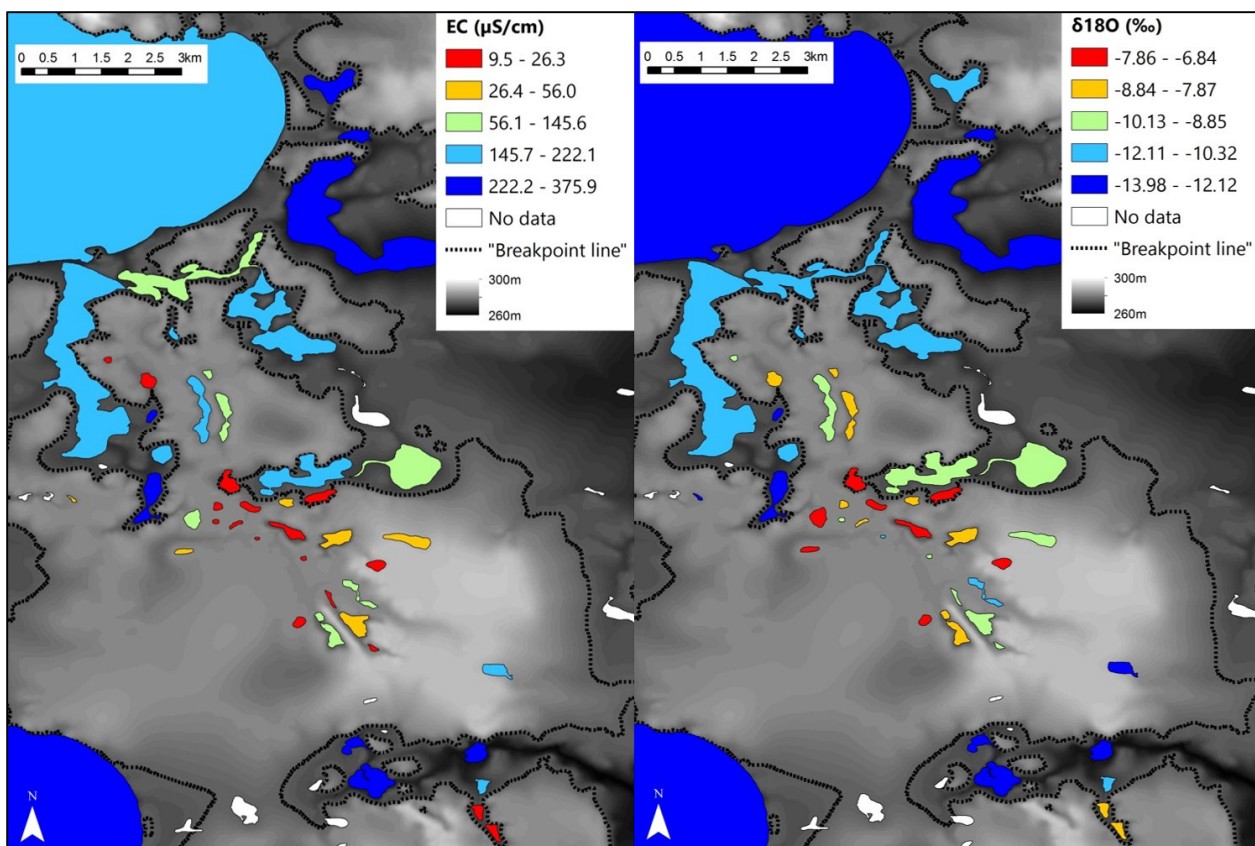

Fig. 5: Spatial depiction between elevation and lake-water specific conductance (µS/cm) (a), and lake-water $\delta^{18}O$ (‰) (b). The elevation of the breakpoint line (284 m) is shown as a dashed line.

To further assess differentiation of the lakes in terms of conservative and non-conservative water tracers, a non-metric multidimensional scaling (NMDS) based on these factors was undertaken, with lakes above an elevation of 282 m a.s.l. coded as groundwater-recharge lakes, and lakes below 282 m a.s.l. coded as groundwater-discharge lakes, indicating good separation of the groups based on water geochemistry (**Fig. 6a**).

The existence of two distinctive types of lakes was used to develop a lake typology to explain changes in water biogeochemistry across the studied lakes. In order to better understand how water tracers vary in those two zones, individual Wilcoxon signed-rank test were undertaken for all lake-water variables that were above detection limits (**Tab. A2**). All anion and cations (and correlated variables including specific conductance), with the exception of K, individually were significantly different above and below an elevation of 282 m a.s.l. This was also the case of the isotopic variables. Variables that were not significantly different included: K and TN.

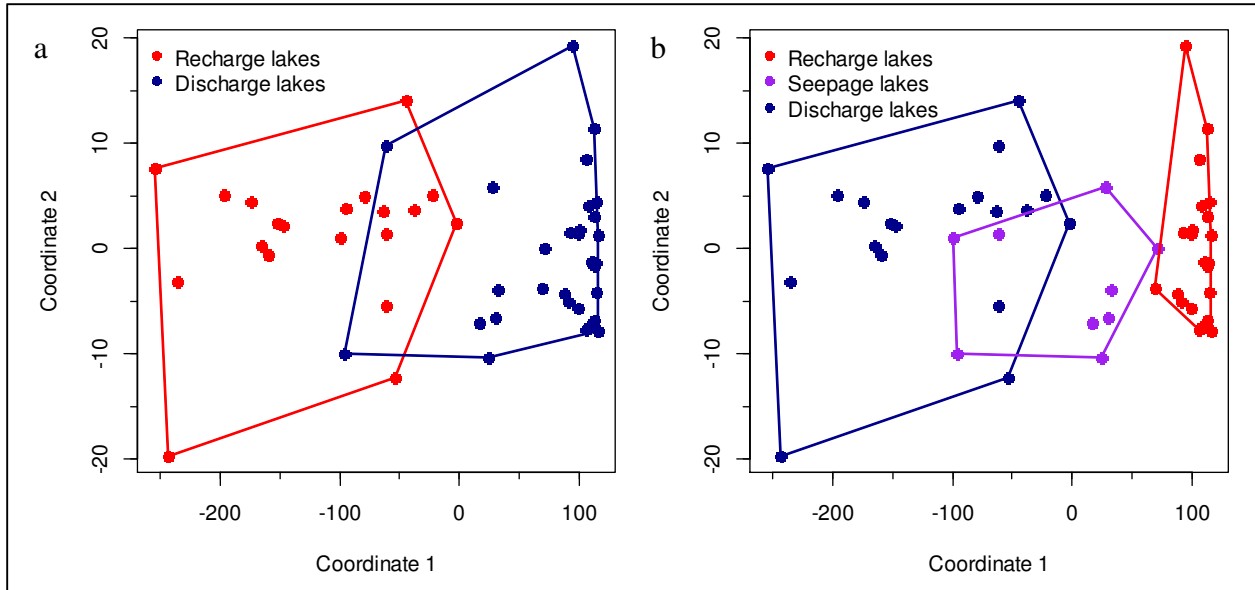

**Fig. 6**: Non-metric multidimensional scaling (NMDS) applied to conservative and non-conservative water
tracers. Lakes that occurred at an elevation of greater than 282 m a.s.l. are labelled as 'recharge' lakes,
whereas lakes as located at an elevation of less than 282 m a.s.l. are labelled as 'discharge' lakes (a). A third
group of lakes (labelled 'seepage') is further discerned based on chemical and isotopic characteristics (b).
The model converged after 35 iterations. Stress value for the NMSD is low (stress = 0.023).

*4.4 Short-term evolution of water tracers*

Short-term water balance variability was observed towards the end of the growing season in 2014
(**Fig. A1b,d** in the appendix) and between the sampling campaign of 2013 and 2014 (**Fig. A2a,c**). Lakes
located at an elevation greater than 282 m a.s.l. underwent marked changes in water balance during the
three sampling campaigns whereas lakes located below 282 m a.s.l. underwent little or no change, especially
the lower lakes (**Fig. A2a,b**). Similarly, the water-chemistry gradient between upland solute-poor lakes and
lowland solute-rich lakes changed seasonally and between years. Lakes above 282 m a.s.l. did not respond
chemically to short-term hydroclimatic change while lakes below 282 m a.s.l. displayed significant solute
changes (**Fig. A2c,d**).

**5 Discussion**

5.1 Interpretation of water tracers

The wide range of lake isotopic values, E/I and f ratios suggest that lakes on the esker are
heterogeneous in terms of water balance and hydrological characteristics. Groundwater springs have the

most depleted isotopic values because their water-residence times are short and they only undergo limited evaporation (Gazis and Feng, 2004; Rey et *al.*, 2018). Streams display isotopic values similar to groundwater springs because they originate from groundwater and experience relatively low evaporation rates due to more continuous water flow. The wide range of isotopic values for lakes can be explained by their position in the landscape, particularly relative to their location in the esker aquifer system for which elevation appears to be a good proxy. This suggests that lowland lakes are primarily fed by groundwater inflow while upland lakes receive a much lower contribution from the local groundwater in their respective water budgets. Nonetheless, it is challenging to distinguish the influence of groundwater from precipitation in lake-water balance as they both have similar isotopic signatures (**Fig. 2b**) (Gibson and Edwards, 2002; Gibson et *al.*, 2008; Yi et *al.*, 2008). Yet, given that: i) the study was carried out on a small spatial scale (*i.e.* a rectangular zone of ~12 km by ~6 km); ii) the close proximity of the lakes; iii) the terrain homogeneity (with boreal forest as the dominant land cover); iv) the limited topography; and v) the strong correlation between water isotopes and specific conductance (r = 0.63, n = 50, *p* < 0.001), it is unlikely that there are significant differences in terms of precipitation patterns within the study area. Therefore, it is a reasonable assumption that groundwater connectivity is the main control on lake water balance. Lake isotopic values can also be influenced by the isotopic composition of surface inflowing waters. But those are mainly groundwater springs (which are made up of groundwater), their volume is small in comparison to the lake volume and, some of those streams are intermittent in the sense that they were not flowing during each of the three field campaigns. Thus, it can reasonably be assumed that this influence is limited.

Similar patterns are observed with the major ions in water. Groundwater springs have the highest solute concentrations likely due to chemical processes associated with mineral surface exchanges and weathering (Ala-aho et *al.*, 2013). Lakes, however, displayed a wide range of water chemistries, and range from high solutes characteristic of groundwater to values close to zero, typical of precipitation (**Fig. 2**), suggesting that the heterogeneity of lakes on the esker are a result to the degree to which they interact with groundwater. As it is the case for isotopic values, chemical composition of a given lake depends on elevation. Lakes sampled at lower elevation are higher in solutes; this indicates that lowland lakes reflect interaction with intermediate or regional groundwater flows subject to more mineral weathering and dissolution (Tóth, 1963). Upland lakes on the other hand reflect interaction with local groundwater flow paths with correspondingly reduced mineral weathering and dissolution (Tóth, 1963). There are only minor differences in terms of the relative solute composition among the samples suggesting that the esker subsurface material is geochemically relatively homogeneous and reflects the carbonate-rich nature of the glaciofuvial outwash that makes up the esker (Cummings et *al.*, 2011). The carbonate-rich sediment

originates from Paleozoic carbonates of the Hudson sedimentary Platform *ca.* 150 km to the north and is
localized to glacial surficial sediments (Roy et *al.*, 2011).

Other characteristics of the lake water (temperature and dissolved organic matter) do not reflect the
changes seen in the stable isotopes and ionic concentration. Water temperatures of groundwater springs are
coolest because the temperature of groundwater is typically close to the mean annual temperature of the
region, while lake-water temperature varies strongly with season. There is little difference in temperature
between lakes and the latter are mainly the result of lake morphology than their connection with cool
groundwater as lakes of smaller volumes have a lower thermal inertia than larger ones. Differences in
dissolved organic content between the lakes is also a result of lake morphology as smaller lakes tend to
have higher concentrations as they mix more easily and receive greater inputs due to their high catchment
size-lake volume ratios (Knoll et *al.*, 2015). However, upland lakes tend to have slightly higher amounts
likely due to their higher water-residence times.

*5.2 Lake hydrological classification*

The correlation between lake-water $\delta^{18}O$ and specific conductance (r = 0.80, n = 50, $p < 0.001$) shows
there is a clear relation between conservative water tracers (which are indicative of water source and
evaporation) and non-conservative water tracers (which are indicative of water flow paths), signifying that
lowland lakes will receive a significant portion of their water as groundwater flows and will geochemically
reflect this origin, while upland lakes will receive most of their water through precipitation and will be
geochemically dilute.

Consequently, the observed breakpoint elevation appears to correspond to an effective hydraulic
separation between groundwater recharge and discharge areas (Winter et *al.*, 1998). This contrast between
the contribution of groundwater flow is evident in the distribution of lake-water composition (**Fig. 5**, **Fig.
A1**). Lakes characterized by groundwater discharge are spatially distinct from higher elevation lakes in the
groundwater recharge zone. Lake position has been used as a classification criterion in several studies (e.g.
Winter, 1977; Born et *al.*, 1979). Thus, upland lakes in the recharge zone are known as groundwater-
recharge lakes or recharge lakes and, conversely, lowland lakes in the discharge zone, also called underflow
zone, will be referred to as groundwater-discharge lakes or discharge lakes (**Fig. 7**) (Winter et *al.*, 1998).
Because discharge lakes receive a substantial amount of water from groundwater, they are considered to be
groundwater-fed or minerotrophic whereas recharge lakes which receive the majority of their water from
precipitation and feed the aquifer are said to be precipitation-fed or ombrotrophic (Webster et *al.*, 1996).
NMDS (**Fig. 6a**) and Wilcoxon signed-rank test (**Tab. A2**) analysis showed that all conservative and non-
conservative tracers are statistically different between the discharge zone and the recharge zone, except for
TN and K. The ANOSIM between of the recharge and discharge lakes shows that within group similarity
is significantly greater than between group similarity, as illustrated by a large and significant r value (r =
0.77, n = 50, significance = 0.001 on 1000 permutations).

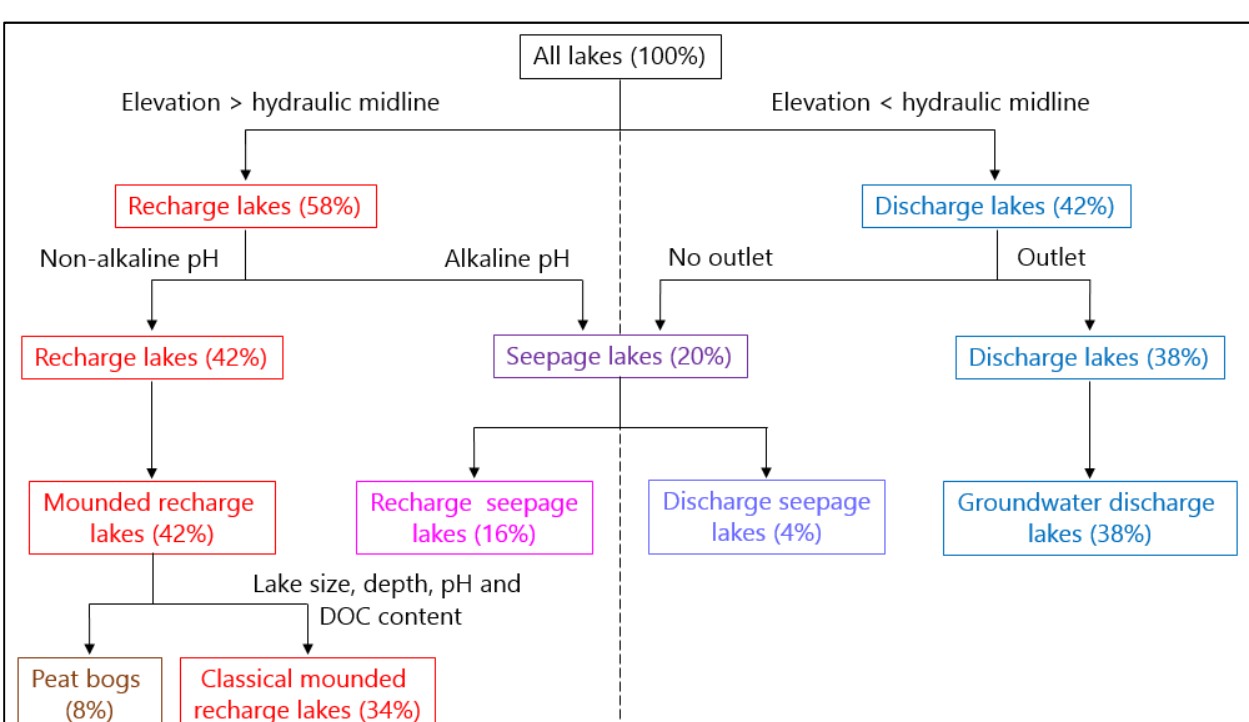

**Fig. 7**: Illustration of a lake typology for the study lakes. Lake elevation was used as the primary factor in
separating lakes, with elevation being an effective surrogate for dividing lakes based on contribution of
groundwater flow to a lake. The lakes were then separated based on presence of an outlet, as well as lake
water pH and other morphological characteristics of the lakes.

Nevertheless, a few solute-rich alkaline lakes are located in the groundwater recharge zone which is

supposed to be depleted in solutes (**Fig. 5a**). Furthermore, the two primary groups in ordinal space display
a small overlap (**Fig. 6a**), suggesting the existence a third category of hybrid lakes referred to as seepage
or flow-through lakes (Winter, 1976; Webster *et al*., 1996; Winter *et al*., 2003). In those lakes, water comes
in as groundwater in-seepage and is returned to the groundwater system as out-seepage (Anderson and
Munter, 1981). Seepage lakes can be found both in the recharge zone and the discharge zone and they can
contribute a recharge or a discharge function (Anderson and Munter, 1981). In the groundwater recharge
zone, seepage lakes differ from recharge lakes by their pH, which is more alkaline while in the groundwater
discharge zone, seepage lakes differ from discharge lakes by the absence of an outlet, meaning they receive
less input than discharge lakes (**Fig. 8**). Lake-water chemistry and the presence of outlets have been used
in several studies as a classification criterion (e.g. Winter, 1977; Newton and Driscoll, 1990). NMDS (**Fig.**
**6b**) and Wilcoxon signed-rank test (**Tab. A3**) analysis showed that all conservative and non-conservative
tracers are statistically different between the three types of lakes, except for $SO_4$.

As noted earlier, lakes can first be classified according to their location within the groundwater

system, particularly above and below the hydraulic midline (*i.e.* the boundary between the groundwater
recharge and discharge areas), in this case, at or near the 282 m a.s.l. elevation in this study that indicated
a breakpoint in the many isotopic and limnological variables (282.4 m a.s.l.) associated with differences in
hydrological inputs. In this study, elevation appears to be a proxy of the boundary between the groundwater
recharge and discharge zones on the esker. Seepage lakes, however, can be found on each side of the
hydraulic midline, thus generating additional classes of lakes (see conceptual diagram, **Fig. 7**) distinguished
from on the presence or absence of a lake outlet (**Fig. A1**) (Stauffer and Wittchen, 1992). Lakes with outlets
were classified as "groundwater discharge lakes" and lakes without outlets as "discharge seepage lakes".

Lakes with outlets are typically found at the contact between the impermeable glaciolacustrine clay

plain and the permeable glaciofluvial esker sand contact where groundwater springs emerge (**Fig. 1b**).
Furthermore, in the study area, there is a significant relation between elevation and lake perimeter maximum
slope ($r = 0.75$, $n = 50$, $p < 0.001$). Surficial geology is also a variable used as a classification criterion in
several studies (Winter, 1977; Martin et *al*., 2011). Lakes located at the edge of the esker tend to have
steeper perimeter slopes and have low elevations, which increases the likelihood of lakes to be in contact
with deeper groundwater flows (Winter, 1976). These lakes receive a substantial inflow of groundwater
due to their geological setting and their water isotopic and chemical composition is similar to the one of
groundwater springs and outlets (Ala-aho et *al*., 2013).

Only two lakes in the groundwater discharge zone are without outlets and are classified as discharge

seepage lakes. The following hypothesis could potentially explain the absence of outlet: the ability of the
lakes to lose water to groundwater; or a reduced input from groundwater or surface inflow. Yet, some lakes
with outlets, like Pit Lake, can be found at elevations corresponding to the groundwater recharge zone (**Fig.**
**5**). The presence of Pit Lake at a distinctly high elevation (296 m a.s.l.) is nonetheless explained by the
location of this lake at the clay-sand interface (**Fig. 1b**). This suggests that elevation is just a putative
variable or, in other words, a proxy of the actual hydraulic midline delineated by the clay-sand interface.
This is confirmed numerically by applying a logistic regression to the presence or absence of a lake outlet
and lake elevation, which provided a poor relation (Mc-Fadden $r = 0.40$, $n = 50$, $p < 0.01$). On the other
hand, the logistic regression of the presence or absence of a lake outlet to the closest distance to the sand-
clay contact provided a better relation (Mc-Fadden r = 0.69, n = 50, $p < 0.001$), although it is challenging
to find the real clay-sand interface as the lateral sands can mask the real exact location (**Fig. 1b**) and surficial
geological maps have errors in tens of meters in comparison to field observations.

Some lakes in the recharge zone, called "recharge seepage lakes", have an alkaline or a circumneutral
pH and higher solute content, signifying that they interact to some extent with groundwater. This could
occur for a number of reasons related to the existence of significant groundwater input including: i) slightly
steeper lake watershed slopes; ii) geographical proximity to the clay-sand interface; and iii) lake relative
deepness allowing interactions with deeper groundwater flows, or a combination of those.

Recharge lakes can be further subdivided into two types of recharge lakes: "classical" recharge lakes
and peat bogs (**Fig. 7**), adding a fifth type of lake in the typology (Newton and Driscoll, 1990). Sampled
peat bogs are very small and shallow lakes rimmed by floating mats of vegetation (typically less than 1 ha
and 1 to 2 m deep), acidic, characterized by a very low amount of solutes (maximum 30 μS/cm), a relatively
high amount of dissolved organic carbon (above 10 ppm) and by water isotopic composition controlled by
short-term hydroclimatic conditions (enriched in heavy water isotopes during drier periods and similar to
the isotopic composition of precipitation during wet periods). This could result from relative hydrological
isolation from the groundwater system due to the thick layer of peat at their bottom formed by the successive
accumulation of sedge and sphagnum characterized by a low hydraulic conductivity (Newton and Driscoll,
1990). As a consequence, direct precipitation would be the predominant source of water, making these lakes
sensitive to hydroclimatic variability.

A sixth type of lake can be added to the proposed lake typology: ephemeral kettle ponds, commonly
called dry kettles, even though they are not lakes *per se*. They can be identified on geological maps, aerial
photography and satellite imagery. Dry kettles consist of small kettle holes adjacent to the esker crest that
are usually but not always too high in elevation and shallow to be connected to the water table and will be
dry during the summer months and most of the year, creating patches of bare land or wetlands in the forested
landscape.

Based on this typology, 42% of the sampled lakes were recharge lakes, 20% seepage lakes and 38%
discharge lakes. Other studies report slightly different proportions (e.g. Anderson and Munter, 1981) as
might be expected based on the geographical location, the size of the study area, and other factors relative
to the groundwater system. The subdivision of lakes into recharge and discharge lakes is consistent with

Ala-aho et *al*. (2013) who worked in similar settings and established their classification based on water solutes. The three-fold typology compares well with Turner et *al*. (2010) who also established a three category lake typology although the terminology of the three lake types differ and E/I and f values were used as the primary classification criteria. Hence, the threefold typology of **Fig. 7** is particularly relevant for water resources management in esker complexes as it uses readily available variables (*i.e.* elevation, specific conductance/pH and presence/absence of an outlet).

Discharge (ground-water discharge) lakes have the highest pH, and amounts of solutes, while the opposite is observed for recharge lakes. Seepage lakes have intermediate values compared to the other two lake types. The same patterns are observed for isotopic values: recharge lakes tend to be enriched in heavier water isotopes, discharge lakes tend to be depleted in heavier isotopes, and seepage lakes characterized by intermediate values. This reinforces the interpretation that seepage lakes are a hybrid type between recharge and discharge lakes. Seepage lakes thus tend to be closer to recharge lakes on an isotopic basis, but more similar to discharge lakes on a chemical basis. This is explained by the fact that most sampled seepage lakes are located in the recharge zone and those receive some intermediate groundwater flows.

Recharge lakes contain higher amounts of dissolved organic matter likely due to substantial water residence times, whereas seepage lakes and discharge lakes can have sustained inflow of groundwater that decreases water residence times. The concentration of DOC in discharge lakes is relatively higher than expected as groundwater usually contain limited organic matter. This is likely an artefact of lake morphology and anthropogenic activities. Indeed, values of DOC for discharge lakes display a high standard deviation due to a subgroup of smaller lakes that are relatively deep and without cottages, combined with another group of larger lakes that are relatively shallow with the presence of cottages. Relatively shallow lakes are known to have significantly higher amounts of DOC than deeper ones as a result of smaller volumes relative to inputs, and potential reworking of bottom organic matter and nutrients due to the shallow water columns. DOC values for discharge lakes also contain two sizeable outliers that skew the average and median: Nighthawk Lake and Frederick House Lake, both of which are two regional high-order and large lakes (10,701 ha and 3,888 ha respectively) that are relatively shallow (maximum depth of 4.6 m and 12.0 m respectively) and are heavily used for recreational purposes.

*5.3 Lake morphometry and water geochemistry*

There are additional morphometric factors that influence lake water isotopic composition in these settings. Evaporation over a water body is enhanced when there is a strong gradient of temperature and

water vapour pressure between the lake water and the adjacent air. Assuming that there is no large
microclimate differences within the esker complex, isotopic values may be influenced by: i) the lake fetch
($r = -0.26$, n = 48, $p = 0.07$) as wind is more efficient at removing moisture over a long distance (Granger
and Hedstrom, 2011); ii) relative depth ($r = -0.32$, n = 32, $p = 0.07$) as lakes that have a large surface
compared to their depth are proportionally more exposed to the atmosphere and thus more susceptible to
evaporation; and iii) the steepness of the slopes surrounding the lake ($r = -0.33$, n = 48, $p = 0.02$). Among
those variables, only the latter variable is significantly related to the isotopic composition of the lake while
the other two may be marginally significant. Steep slopes tend to reduce evaporation rates by blocking air
flows over the lake, thus reducing wind speed, water-air temperature and water vapour pressure contrasts,
and to increase the likelihood of lakes to be in contact with deeper groundwater flows (Winter, 1976).

Lake-water chemistry can also be influenced by lake morphometry. Lake depth has been known to

play a role (Winter, 1976). It seems that *a priori* there is no relation between lake maximum depth and
specific conductance ($r = 0.09$, n = 39, $p = 0.59$). However, to determine whether the relation between
specific conductance and lake maximum depth varied with lake landscape position, two separate
regressions analyses were carried out for recharge ($r = 0.01$, n = 16, $p = 0.66$) and discharge lakes ($r = 0.30$,
n = 23, $p = 0.05$). The two regressions show that maximum depth can act as a control on specific
conductance only in lakes in the discharge zone: the deeper the lake, the deeper solutes-rich groundwater
flows seep into the lake. It is also expected that small lakes with a large ratio of perimeter to surface area ($r$
$= 0.48$, n = 48, $p < 0.001$) will have a higher amounts of solutes as the majority of groundwater seepage
into the lake is typically localised near the shoreline (Rosenberry et *al*., 2015).

Water-quality tracers (DOC and TN) were not correlated to elevation, suggesting there is no apparent

relationship between groundwater inflow and water quality. The correlation between water-quality tracers
and the ratio of lake perimeter to surface area may suggest a relation between lake morphometry and water
quality. Elongated lakes have proportionally a greater interface with terrestrial ecosystem, which favours
the inflow of nutrients in lakes while mean depth is seen as an indicator of water residence times and rates
of mixing (Mulholland, 2003; Knoll et *al*., 2015).

Given that some lacustrine morphometric variables are correlated to some water tracers, it can be

argued that lake morphometry can play a significant role on lake-water biogeochemistry and introduce
some complexity in the lake typology. As a fact, some lake typologies incorporate morphometric features
as defining characteristics (e.g. Winter, 1977; Martin et *al*., 2011; Knoll et *al*., 2015). Even though water
tracers show a stronger correlation with elevation than with morphometric features, it cannot be assessed
with certainty that elevation accounts for much more variance in isotopic composition and specific
conductance as certain other key morphometric variables are not available for comparison. Such variables
include the lake network number or the measure of connections to other lakes, which can modify lake
isotopic composition or the lake to catchment area ratio, which can affect lake-water chemistry. In other
studies, land use has been identified as a key variable explaining lake water balance (as canopy cover reduce
evaporative losses) and water chemistry (as substrate and water pathways influence water chemistry) (e.g.
Turner et *al*., 2010; Turner et *al*., 2014a and Turner et *al*., 2014b). However, given the predominance of the
boreal forest in the study area, the impact of land use on water geochemistry was not considered as an
explanatory variable. Specific yield and soil type are additional variables that can also impact lake isotopic
composition and lake-water chemistry. Hence, several morphometric and geophysical variables can modify
lake chemistry and make the relation between elevation and lake-water balance and/or water chemistry
more complex.

5.4 Implications for water balance, hydrochemistry and response to external stressors

Results from the study show that the water balance of individual lakes is highly dependent on the

nature of their interactions with groundwater system that is in turn determined by landscape position at
highly localized scale as shown in the conceptual model in **Fig. 8a**. Under drier hydroclimatic conditions,
groundwater-discharge lakes undergo minimal to negative changes in water balance as the continuous flow
of isotopically light groundwater masks evaporative enrichment whereas groundwater-recharge lakes are
highly sensitive to evaporation as they rely on precipitation as their primary source of water. Seepage lakes
constitute a hybrid between recharge and discharge and their water balance is between the other two lake
types (**Fig. 4b**). These patterns of change were observed when comparing changes in E/I and f ratios
between sampling campaigns: E/I and f ratios experienced negative noticeable changes in groundwater-
discharge lakes and small changes in seepage lakes while positive changes in E/I and f were observed in
groundwater-recharge lakes (**Fig. 8b** and **Fig. 8c**). Consequently, upland groundwater-recharge lakes will
be more prone to evaporative drawdown and therefore more sensitive to short-term climate change and
droughts, while groundwater-discharge lakes will be buffered by groundwater inflow and affected by
hydroclimatological changes of greater duration and persistence that alter water table position (**Fig. 8a**).
Seepage lakes will presumably be sensitive to drought but not to the same extent as groundwater-recharge
lakes as they have stronger interactions with groundwater (**Fig. 8a**). This suggests that the esker
hydrological system can respond to a large-scale hydroclimatic forcing (e.g. prolonged drought) in a manner
that affects individual lakes differently. The degree of interaction with groundwater by an individual lake
will also dictate the response to strong hydroclimatic forcings and introduce time lags (Webster et *al*., 2000).

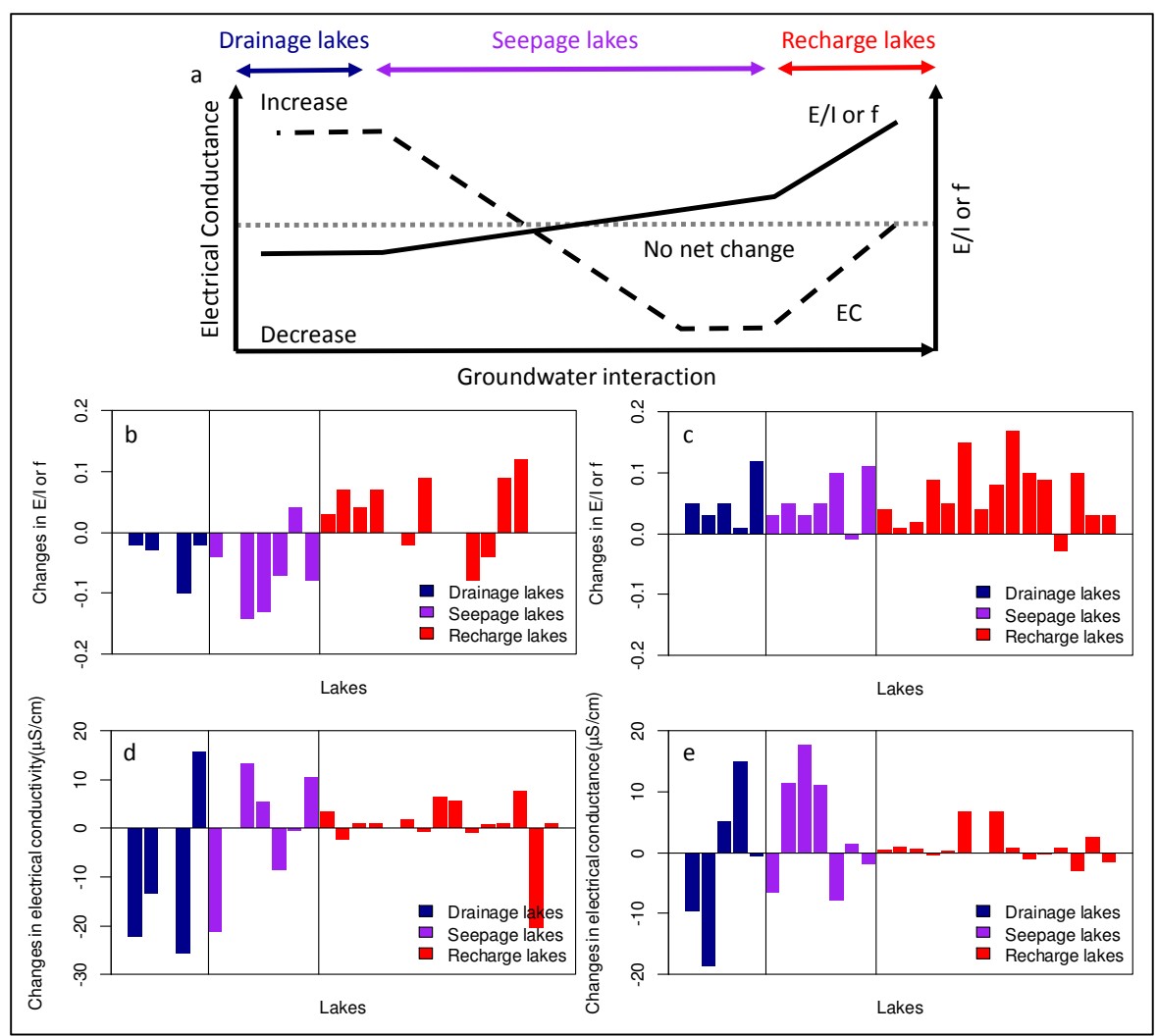

**Fig. 8**: Conceptual model of the relationship between the direction and magnitude of lake E/I or f and EC changes during drier conditions to lake type defined by the degree to which lakes interact with groundwater (modified from Webster et *al*. (1996)) (a) and observed changes in E/I or f between June 2013 and June 2014 (E/I 2014 – E/I 2013) (b) and changes in E/I between August 2014 and June 2014 (E/I or f Aug – E/I or f Jun) (c), and changes in specific conductance between June 2013 and June 2014 (EC 2014 – EC 2013) (d) and changes in electrical conductance between August 2014 and June 2014 (EC Aug – EC Jun) (e) by lake type.

Similarly, the water-chemistry gradient between upland solute-poor lakes and lowland solute-rich lakes could make recharge lakes more vulnerable to acidification than seepage and discharge lakes. The effects of changing hydroclimatic conditions on lake-water chemistry across the landscape observed during successive sampling in June 2013, June 2014 and August 2014 resulted in heterogeneous pattern depending on the lake type, consistent with the findings of Webster et *al*. (1996). Groundwater-recharge lakes did not respond chemically to short-term hydroclimatic change as they feed the groundwater system by lake out-seepage thus producing no net change in solutes, while seepage and groundwater-discharge lakes displayed

significant solute changes (**Fig. 8b** and **Fig. 8c**). Based on the limited sampling frequency, it is difficult to draw conclusions on which of the seepage or discharge lake types undergo the most chemical variation. A greater temporal resolution of sampling would provide more insights on short-term changes in hydrochemistry in relation to short-term hydroclimatic fluctuations and reinforce positively these interpretations. However, Webster et *al*. (1996) suggest that groundwater-discharge lakes and seepage lakes respond chemically to evaporative drawdown in opposite ways, with seepage lakes showing a decline in solutes during droughts as inputs from groundwater diminish due to the lowering of the water table (**Fig. 8a**). By contrast, groundwater-discharge lakes have been noted as susceptible to evaporative enrichment of solutes and increased relative contribution of solute-rich groundwater during drought periods (Kratz et *al*., 1997). For these reasons, Webster et *al*. (1996) suggested that climate change could amplify anthropogenic impacts and make lakes more vulnerable to other stressors, such as lake acidification.

Due to their varied hydrological characteristics, the lake types identified in this study will have a different susceptibility to direct anthropogenic impacts. Because groundwater-recharge lakes have lower groundwater inflow, they are characterized by relatively long water-residence times, making them highly vulnerable to inputs and pollutants in comparison to seepage lakes and groundwater-discharge lakes that tend to have a greater watershed, which results in increased flushing. Given this situation, the limitation of cottage development and disposal sites in the groundwater recharge zone and/or stricter regulations on cottage sceptic tanks in this zone would prevent the downstream contamination and subsequent degradation of water quality in the groundwater discharge zone delimited in **Fig. 5**. However, it can be argued that differences in material permeability between the esker and the clay plain can produce the opposite effect as clays and organic deposits on the esker flanks act as an aquitard that locally confine the aquifer (**Fig. 1b**) (Rossi et *al*., 2012). Sand extraction and mining activities in the groundwater discharge zone could potentially influence the water levels of upland lakes in the recharge zone (Klove et *al*., 2011). Indeed, material excavation and tunnel construction can cause the desiccation of groundwater-dependent systems by reversing flow patterns in the recharge area, and increase the discharge from the esker after ditching on the esker flanks and drain groundwater-dependent systems in the recharge area (Klove et *al*., 2011; Rossi et *al*., 2012). As a result, the implementation of incidence studies that investigate and take into account local groundwater flow patterns and hydrological connections between the recharge and discharge areas prior to any excavation project in those settings would allow the assessment of any undesired effects on hydrological systems and their functions and services. This potential was acknowledged by an impact study undertaken for an aborted aggregate pit project in the study area, and one recommendation was the construction of an engineered frozen earth barrier to prevent ground water flow into the proposed pit in order to minimize the effects on the water table and surrounding lake levels (Cochrane, 2006).

Finally, when considering groundwater-fed lakes for the purposes of paleohydrological
reconstruction (e.g. Laird et *al*., 2012), it is critical to explore their modern hydrology to be able to have a
baseline for comparison to future hydrological regimes that may alter them and subsequently correctly
interpret the probable causes of isotopic, chemical, and biological change and variability recorded in the
sediment through time along with their potential hydroclimatic drivers. Because the degree to which lakes
interact with groundwater produces differences in hydrologic response to the same hydroclimatic forcing,
the interpretation of paleolimnological records can be complex (Fritz, 2000) and multiple site selection
seems necessary as a groundwater-discharge lake may show long-term stability while a groundwater-
recharge lake may display significant short-term and long-term variability (Bennett et *al*., 2007).

**6 Conclusion**

Lakes located in an esker complex in northeast Ontario showed strong systematic and localized
differences in terms of water balance and hydrochemistry, similar to other esker complexes in other settings.
Results from this study indicated that elevation is a critical factor explaining water chemistry and water
balance across the landscape. As eskers are structurally complex and often characterized by high hydraulic
conductivity, groundwater interactions are an important component of lake water hydrology. Low-elevation
lakes are likely interconnected with solute-rich intermediate and regional groundwater flowpaths while
upland lakes are only interacting with local solute-poor groundwater flowpaths. This threshold in water
chemistry is also accompanied by strong contrasts in lake-water balance. Upland lakes tend to be
isotopically-enriched and more sensitive to evaporation, while lowland lakes are more depleted and subject
to groundwater inflows. Limnological variables including lake depth, DOC and forms of nitrogen were
weakly related to elevation, and likely influenced by lake morphometry and watershed activities. Thus,
these results are in agreement with other studies that indicate that at the local scale, landscape position is
the main control on lake water chemistry and balance while lake morphometric characteristics explain
additional variance.

The physical and chemical characteristics of lake water allowed the development of a lake typology
that is made up of three main types of lakes: i) higher-elevation groundwater-recharge lakes, essentially fed
by precipitation, are characterized by higher evaporations rates and lower amounts of solutes; ii) seepage
lakes, that both gain and lose water to the groundwater, are characterized by intermediate rates of
evaporation and amounts of solutes; and iii) lower-elevation groundwater-discharge lakes, are continuously
fed by groundwater inflow and characterized by almost no evaporation rates and higher amounts of solutes.
The obtained typology provides insights about lake vulnerability to environmental stressors, particularly
short- and long-term hydroclimatic change. Groundwater-recharge lakes will be more prone to evaporative
drawdown and therefore more sensitive to short-term droughts, while groundwater-discharge lakes will be
buffered by groundwater inflow and affected by hydroclimatological changes of greater persistence.
Similarly, recharge lakes will likely be more subject to other anthropogenic impacts, in comparison to
groundwater-discharge lakes.

**Acknowledgments**

This work was supported by the Natural Sciences and Engineering Research Council of Canada
(NSERC) to Scott Lamoureux [Discovery grant 227165], and by the Graduate Dean's Doctoral Field Travel
Grant from Queen's University to Maxime Boreux. The authors want to thank Graham Mushet, Gladys
Kong, Cécilia Barouillet and Dan Lamhonwah for their field assistance, Steve Koziar, Allison Rutter, Liz
Kjikjerkovska and Dan Lamhonwah for their assistance in the lab, and Melissa Lafrenière and Ontario
Parks for their logistical support. Many thanks to local property owners, in particular to Franca Adamo-
Wheeler, Jessica Adamo and their family, to Rob and Mary Ann Stewart, and to Ron and Tarja Bouchard
who allowed access to their property for lake sampling, shared precious field information and kindly offered
logistic support. The authors thank Kevin Turner and the three other anonymous reviewers for their valuable
remarks allowing the improvement of the manuscript.

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

**Author contributions**

Maxime Boreux collected data in the field and processed samples in the lab, conducted data analysis and
interpretation (with input from all authors), generated figures and tables. Scott Lamoureux provided
guidance, funding, reviewed and edited the manuscript in his function as supervisor, as did Brian Cumming.
The manuscript was written by Maxime Boreux with input from all authors.
**Data availability**

The data can be made available by contacting the corresponding author.

**Competing interests**

The authors declare that they have no conflict of interest.

















**Appendix**

**Tab. A1a**: Correlation matrix between hydrological tracers and morphometric variables.

| | EC | pH | T | $Ca^{2+}$ | $Mg^{2+}$ | $K^+$ | $Na^+$ | $Cl^-$ | $SO_4^{2-}$ | $NO_3^-$ | $NH_4^+$ | $\delta^{18}O$ | $\delta^2H$ |
|---|---|---|---|---|---|---|---|---|---|---|---|---|---|
| EC | **1.00** | 0.73 | -0.22 | 0.97 | 0.93 | 0.73 | 0.60 | 0.62 | 0.36 | -0.13 | -0.18 | -0.80 | -0.75 |
| pH | 0.73 | **1.00** | -0.05 | 0.76 | 0.73 | 0.41 | 0.29 | 0.29 | 0.20 | -0.15 | -0.18 | -0.47 | -0.41 |
| T | -0.22 | -0.05 | **1.00** | -0.24 | -0.25 | -0.32 | 0.06 | 0.05 | -0.12 | 0.19 | -0.16 | 0.34 | 0.33 |
| $Ca^{2+}$ | 0.97 | 0.76 | -0.24 | **1.00** | 0.94 | 0.66 | 0.43 | 0.48 | 0.40 | -0.12 | -0.18 | -0.83 | -0.78 |
| $Mg^{2+}$ | 0.93 | 0.73 | -0.25 | 0.94 | **1.00** | 0.72 | 0.36 | 0.37 | 0.26 | -0.12 | -0.18 | -0.77 | -0.73 |
| $K^+$ | 0.73 | 0.41 | -0.32 | 0.66 | 0.72 | **1.00** | 0.56 | 0.49 | 0.37 | -0.01 | 0.05 | -0.55 | -0.51 |
| $Na^+$ | 0.60 | 0.29 | 0.06 | 0.43 | 0.36 | 0.56 | **1.00** | 0.95 | 0.12 | -0.06 | -0.07 | -0.31 | -0.29 |
| $Cl^-$ | 0.62 | 0.29 | 0.05 | 0.48 | 0.37 | 0.49 | 0.95 | **1.00** | 0.06 | -0.06 | -0.04 | -0.35 | -0.33 |
| $SO_4^{2-}$ | 0.36 | 0.20 | -0.12 | 0.40 | 0.26 | 0.37 | 0.12 | 0.06 | **1.00** | 0.03 | -0.05 | -0.33 | -0.30 |
| $NO_3^-$ | -0.13 | -0.15 | 0.19 | -0.12 | -0.12 | -0.01 | -0.06 | -0.06 | 0.03 | **1.00** | 0.04 | 0.16 | 0.16 |
| $NH_4^+$ | -0.18 | -0.18 | -0.16 | -0.18 | -0.18 | 0.05 | -0.07 | -0.04 | -0.05 | 0.04 | **1.00** | 0.14 | 0.16 |
| $\delta^{18}O$ | -0.80 | -0.47 | 0.34 | -0.83 | -0.77 | -0.55 | -0.31 | -0.35 | -0.33 | 0.16 | 0.14 | **1.00** | 0.99 |
| $\delta^2H$ | -0.75 | -0.41 | 0.33 | -0.78 | -0.73 | -0.51 | -0.29 | -0.33 | -0.30 | 0.16 | 0.16 | 0.99 | **1.00** |
| *d* | 0.84 | 0.53 | -0.34 | 0.87 | 0.82 | 0.59 | 0.33 | 0.37 | 0.35 | -0.16 | -0.11 | -0.99 | -0.96 |
| E/I f | -0.80 | -0.66 | 0.45 | -0.83 | -0.77 | -0.59 | -0.37 | -0.30 | -0.78 | 0.18 | 0.24 | 0.94 | 0.93 |
| TC | 0.96 | 0.75 | -0.43 | 0.96 | 0.95 | 0.74 | 0.59 | 0.42 | 0.36 | -0.17 | -0.03 | -0.88 | -0.85 |
| DOC | -0.12 | -0.25 | -0.05 | -0.09 | -0.19 | 0.26 | 0.00 | -0.14 | 0.59 | -0.03 | 0.50 | -0.06 | -0.08 |
| DIC | 0.96 | 0.81 | -0.38 | 0.95 | 0.97 | 0.62 | 0.56 | 0.44 | 0.16 | -0.16 | -0.21 | -0.82 | -0.79 |
| TN | -0.22 | -0.30 | -0.09 | -0.21 | -0.29 | 0.16 | 0.00 | -0.09 | 0.44 | -0.06 | 0.53 | 0.08 | 0.06 |
| C/N | -0.13 | -0.21 | 0.28 | -0.10 | -0.10 | 0.12 | -0.10 | -0.20 | 0.35 | 0.04 | 0.17 | -0.07 | -0.08 |
| H | -0.67 | -0.57 | 0.04 | -0.69 | -0.72 | -0.55 | -0.31 | -0.28 | -0.25 | 0.08 | 0.16 | 0.53 | 0.49 |
| Z | 0.09 | 0.20 | 0.17 | 0.06 | 0.18 | 0.06 | -0.01 | 0.00 | -0.18 | 0.08 | 0.01 | 0.01 | 0.05 |
| S | 0.51 | 0.50 | -0.04 | 0.52 | 0.53 | 0.45 | 0.22 | 0.16 | 0.39 | -0.10 | -0.24 | -0.33 | -0.27 |
| A | 0.35 | 0.40 | 0.23 | 0.37 | 0.38 | 0.32 | 0.20 | 0.17 | 0.18 | -0.06 | -0.08 | -0.24 | -0.21 |
| P | 0.42 | 0.53 | 0.25 | 0.45 | 0.45 | 0.35 | 0.21 | 0.17 | 0.24 | -0.07 | -0.13 | -0.29 | -0.27 |
| P/A | -0.48 | -0.73 | -0.28 | -0.48 | -0.49 | -0.29 | -0.29 | -0.23 | -0.40 | 0.00 | 0.21 | 0.10 | 0.02 |

Note: electrical conductance (EC) is expressed in μS/cm, temperature (T) in °C, dissolved ions ($Ca^{2+}$, $Mg^{2+}$,
$K^+$, $Na^+$, $Cl^-$, $SO_4^{2-}$, $NO_2^-$-$NO_3^-$, $NH_4^+$) in ppm, water stable isotopes ($\delta^{18}O$, $\delta^2H$) and deuterium excess (*d*) in
V-SMOW, total carbon (TC), dissolved organic carbon (DOC), dissolved inorganic carbon (DIC) and total
nitrogen (TN) in ppm, elevation (H), lake maximum depth (Z) and perimeter (P) in m, maximum lake
perimeter slope (S) in % and area (A) in ha.

**Tab. A1b**: Correlation matrix between hydrological tracers and morphometric variables (con't).

| | $d$ | E/I f | TC | DOC | DIC | TN | C/N | H | Z | S | A | P | P/A |
|---|---|---|---|---|---|---|---|---|---|---|---|---|---|
| EC | 0.84 | -0.80 | 0.96 | -0.12 | 0.96 | -0.22 | -0.13 | -0.67 | 0.09 | 0.51 | 0.35 | 0.42 | -0.48 |
| pH | 0.53 | -0.66 | 0.75 | -0.25 | 0.81 | -0.30 | -0.21 | -0.57 | 0.20 | 0.50 | 0.40 | 0.53 | -0.73 |
| T | -0.34 | 0.45 | -0.43 | -0.05 | -0.38 | -0.09 | 0.28 | 0.04 | 0.17 | -0.04 | 0.23 | 0.25 | -0.28 |
| $Ca^{2+}$ | 0.87 | -0.83 | 0.96 | -0.09 | 0.95 | -0.21 | -0.10 | -0.69 | 0.06 | 0.52 | 0.37 | 0.45 | -0.48 |
| $Mg^{2+}$ | 0.82 | -0.77 | 0.95 | -0.19 | 0.97 | -0.29 | -0.10 | -0.72 | 0.18 | 0.53 | 0.38 | 0.45 | -0.49 |
| $K^+$ | 0.59 | -0.59 | 0.74 | 0.26 | 0.62 | 0.16 | 0.12 | -0.55 | 0.06 | 0.45 | 0.32 | 0.35 | -0.29 |
| $Na^+$ | 0.33 | -0.37 | 0.59 | 0.00 | 0.56 | 0.00 | -0.10 | -0.31 | -0.01 | 0.22 | 0.20 | 0.21 | -0.29 |
| $Cl^-$ | 0.37 | -0.30 | 0.42 | -0.14 | 0.44 | -0.09 | -0.20 | -0.28 | 0.00 | 0.16 | 0.17 | 0.17 | -0.23 |
| $SO_4^{2-}$ | 0.35 | -0.75 | 0.36 | 0.59 | 0.16 | 0.44 | 0.35 | -0.25 | -0.18 | 0.39 | 0.18 | 0.24 | -0.40 |
| $NO_3^-$ | -0.16 | 0.18 | -0.17 | -0.03 | -0.16 | -0.06 | 0.04 | 0.08 | 0.08 | -0.10 | -0.06 | -0.07 | 0.00 |
| $NH_4^+$ | -0.11 | -0.24 | -0.03 | 0.50 | -0.21 | 0.53 | 0.17 | 0.16 | 0.01 | -0.24 | -0.08 | -0.13 | 0.21 |
| $\delta^{18}O$ | -0.99 | 0.94 | -0.88 | -0.06 | -0.82 | 0.08 | -0.07 | 0.53 | 0.01 | -0.33 | -0.24 | -0.29 | 0.10 |
| $\delta^2H$ | -0.96 | 0.93 | -0.85 | -0.08 | -0.79 | 0.06 | -0.08 | 0.49 | 0.05 | -0.27 | -0.21 | -0.27 | 0.02 |
| $d$ | **1.00** | -0.92 | 0.90 | 0.05 | 0.85 | -0.11 | 0.05 | -0.57 | 0.04 | 0.40 | 0.28 | 0.32 | -0.20 |
| E/I f | -0.92 | **1.00** | -0.81 | 0.37 | -0.81 | 0.49 | -0.05 | 0.48 | -0.07 | -0.37 | -0.22 | -0.31 | 0.24 |
| TC | 0.90 | -0.81 | **1.00** | 0.05 | 0.94 | -0.05 | 0.01 | -0.64 | 0.03 | 0.50 | 0.31 | 0.38 | -0.27 |
| DOC | 0.05 | 0.37 | 0.05 | **1.00** | -0.29 | 0.93 | 0.66 | -0.04 | -0.29 | 0.08 | 0.00 | -0.12 | 0.54 |
| DIC | 0.85 | -0.81 | 0.94 | -0.29 | **1.00** | -0.37 | -0.21 | -0.60 | 0.12 | 0.45 | 0.29 | 0.39 | -0.40 |
| TN | -0.11 | 0.49 | -0.05 | 0.93 | -0.37 | **1.00** | 0.40 | 0.05 | -0.43 | -0.03 | -0.01 | -0.10 | 0.60 |
| C/N | 0.05 | -0.05 | 0.01 | 0.66 | -0.21 | 0.40 | **1.00** | -0.09 | 0.08 | 0.18 | 0.07 | -0.03 | 0.03 |
| H | -0.57 | 0.48 | -0.64 | -0.04 | -0.60 | 0.05 | -0.09 | **1.00** | -0.26 | -0.77 | -0.46 | -0.50 | 0.53 |
| Z | 0.04 | -0.07 | 0.03 | -0.29 | 0.12 | -0.43 | 0.08 | -0.26 | **1.00** | 0.30 | -0.02 | 0.04 | -0.40 |
| S | 0.40 | -0.37 | 0.50 | 0.08 | 0.45 | -0.03 | 0.18 | -0.77 | 0.30 | **1.00** | 0.33 | 0.35 | -0.52 |
| A | 0.28 | -0.22 | 0.31 | 0.00 | 0.29 | -0.01 | 0.07 | -0.46 | -0.02 | 0.33 | **1.00** | 0.91 | -0.49 |
| P | 0.32 | -0.31 | 0.38 | -0.12 | 0.39 | -0.10 | -0.03 | -0.50 | 0.04 | 0.35 | 0.91 | **1.00** | -0.55 |
| P/A | -0.20 | 0.24 | -0.27 | 0.54 | -0.40 | 0.60 | 0.03 | 0.53 | -0.40 | -0.52 | -0.49 | -0.55 | **1.00** |

Note: electrical conductance (EC) is expressed in μS/cm, temperature (T) in °C, dissolved ions ($Ca^{2+}$, $Mg^{2+}$,
$K^+$, $Na^+$, $Cl^-$, $SO_4^{2-}$, $NO_2^-$-$NO_3^-$, $NH_4^+$) in ppm, water stable isotopes ($\delta^{18}O$, $\delta^2H$) and deuterium excess ($d$) in
V-SMOW, total carbon (TC), dissolved organic carbon (DOC), dissolved inorganic carbon (DIC) and total
nitrogen (TN) in ppm, elevation (H), lake maximum depth (Z) and perimeter (P) in m, maximum lake
perimeter slope (S) in % and area (A) in ha.


**Tab. A2**. Results of the Wilcoxon signed-rank test of individual conservative and non-conservative hydrological tracers using the 2-class lake typology as the categorical variable. Non-significant *p*-values are indicated in bold. Lakes in the 'recharge' zone are defined as lakes above an elevation of 282 m a.s.l., whereas lakes in the 'discharge' zone are defined as lakes located at an elevation of less than 282 m a.s.l.

| Variable | Lakes in the recharge zone Mean (SD) | Lakes in the discharge zone Mean (SD) | Wilcoxon signed-rank test V value | p-value |
|---|---|---|---|---|
| EC (μS/cm) | 47.6 (53.5) | 230.4 (88.4) | 1275 | $p < 0.001$ |
| pH | 6.7 (0.9) | 7.9 (0.4) | 1275 | $p < 0.001$ |
| T (°C) | 17.0 (1.9) | 16.5 (2.6) | 1275 | $p < 0.001$ |
| $Ca^{2+}$ (ppm) | 4.8 (6.6) | 29.4 (10.7) | 2138 | $p < 0.001$ |
| $Mg^{2+}$ (ppm) | 1.1 (1.4) | 7.1 (3.1) | 1013 | $p < 0.001$ |
| $K^+$ (ppm) | 0.4 (0.2) | 1.0 (0.4) | **699** | **0.559** |
| $Na^+$ (ppm) | 0.8 (1.9) | 5.3 (7.9) | 867 | 0.027 |
| $Cl^-$ (ppm) | 1.1 (3.9) | 9.5 (15.0) | 1572 | 0.024 |
| $SO_4^{2-}$ (ppm) | 0.9 (0.8) | 6.6 (13.9) | 1003 | $p < 0.001$ |
| $NO_2^--NO_3^-$ (ppb) | 8.3 (25.0) | 56.5 (297.6) | 638 | $p < 0.001$ |
| $NH_4^+$ (ppb) | 22.5 (6.4) | 31.8 (28.6) | 231 | $p < 0.001$ |
| $\delta^{18}O$ (‰) | - 8.7 (1.4) | - 11.5 (1.6) | 0 | $p < 0.001$ |
| $\delta^2H$ (‰) | - 78.2 (7.1) | - 90.6 (7.6) | 0 | $p < 0.001$ |
| *d* (‰) | - 8.5 (4.5) | 1.7 (5.5) | 234 | $p < 0.001$ |
| E/I or f | 0.6 (0.2) | 0.2 (0.3) | 1406 | $p < 0.001$ |
| TC (ppm) | 11.4 (5.9) | 30.4 (8.7) | 666 | $p < 0.001$ |
| DOC (ppm) | 5.9 (3.2) | 6.2 (4.9) | 666 | $p < 0.001$ |
| DIC (ppm) | 5.5 (6.4) | 24.2 (10.6) | 653 | $p < 0.001$ |
| TN (ppm) | 0.5 (0.2) | 0.5 (0.3) | **217** | **0.069** |
| Atomic C/N | 13.2 (3.3) | 13.6 (4.0) | 666 | $p < 0.001$ |
| Elevation (m) | 287.7 (4.8) | 275.1 (2.8) | 1275 | $p < 0.001$ |
| Maximum depth (m) | 12.4 (6.2) | 13.2 (6.7) | 1750 | $p < 0.001$ |
| Maximum lake perimeter slope (%) | 9.8 (6.9) | 20.5 (5.7) | 1273 | $p < 0.001$ |

**Tab. A3**: Results of the Wilcoxon signed-rank test of conservative and non-conservative hydrological
tracers using the 3 class lake typology as the categorical variable. Non-significant *p*-values are indicated in
bold. Lakes in the 'recharge' zone are defined as lakes above an elevation of 282 m a.s.l. and non-alkaline,
lakes in the 'seepage' zone are defined as lakes above an elevation of 282 m a.s.l. and alkaline whereas lakes
in the 'discharge' zone are defined as lakes located at an elevation of less than 282 m a.s.l.

| | Recharge lakes | Seepage lakes | Discharge lakes | Wilcoxon signed-rank test | |
|---|---|---|---|---|---|
| Variable | Mean (SD) | Mean (SD) | Mean (SD) | V value | p-value |
| EC (µS/cm) | 20.5 (11.9) | 135.9 (60.8) | 233.2 (92.5) | 1275 | $p < 0.001$ |
| pH | 6.2 (0.7) | 7.8 (0.3) | 7.9 (0.4) | 1275 | $p < 0.001$ |
| T (°C) | 17.1 (2.0) | 16.4 (1.7) | 16.5 (2.8) | 1275 | $p < 0.001$ |
| $Ca^{2+}$ (ppm) | 1.4 (1.4) | 16.8 (8.6) | 29.5 (11.2) | 1705 | $p = 0.002$ |
| $Mg^{2+}$ (ppm) | 0.4 (0.3) | 3.5 (1.7) | 7.2 (3.3) | 927 | 0.005 |
| $K^+$ (ppm) | 0.5 (0.3) | 0.4 (0.1) | 1.0 (0.4) | 3 | $p < 0.001$ |
| $Na^+$ (ppm) | 0.4 (0.7) | 1.9 (2.9) | 5.6 (8.3) | 876 | 0.009 |
| $Cl^-$ (ppm) | 0.4 (1.3) | 3.3 (6.5) | 9.9 (15.7) | 832 | 0.004 |
| $SO_4^{2-}$ (ppm) | 0.7 (0.6) | 1.7 (1.1) | 7.0 (14.6) | **486** | **0.145** |
| $NO_2^- - NO_3^-$ (ppb) | 78.0 (347.3) | 0.0 (0.0) | 8.8 (25.6) | 1 | $p < 0.001$ |
| $NH_4^+$ (ppb) | 33.9 (32.9) | 25.4 (0.0) | 22.6 (6.5) | 0 | $p < 0.001$ |
| $\delta^{18}O$ (‰) | -8.5 (1.3) | - 9.7 (1.6) | - 11.6 (1.7) | 0 | $p < 0.001$ |
| $\delta^2H$ (‰) | - 77.4 (6.8) | - 82.1 (7.8) | - 90.8 (8.0) | 0 | $p < 0.001$ |
| *d* (‰) | - 9.6 (3.7) | - 4.5 (5.4) | 1.9 (5.7) | 129 | $p < 0.001$ |
| E/I or f | 0.7 (0.3) | 0.6 (0.2) | 0.2 (0.2) | 218 | $p < 0.001$ |
| TC (ppm) | 8.7 (4.0) | 19.9 (6.6) | 30.8 (9.4) | 666 | $p < 0.001$ |
| DOC (ppm) | 6.7 (3.5) | 3.9 (1.2) | 6.7 (5.2) | 649 | $p < 0.001$ |
| DIC (ppm) | 2.0 (1.3) | 15.9 (7.2) | 24.1 (11.5) | 644 | $p < 0.001$ |
| TN (ppm) | 0.6 (0.2) | 0.4 (0.1) | 0.5 (0.3) | 2 | $p < 0.001$ |
| Atomic C/N | 14.1 (3.5) | 11.1 (1.4) | 14.0 (4.3) | 666 | $p < 0.001$ |
| Elevation (m) | 287.0 (4.6) | 286.7 (7.6) | 275.1 (3.0) | 1275 | $p < 0.001$ |
| Maximum depth (m) | 12.0 (6.6) | 12.4 (5.7) | 13.5 (6.8) | 1701 | $p < 0.001$ |
| Maximum lake perimeter slope (%) | 10.2 (7.4) | 11.3 (6.7) | 19.9 (6.7) | 1272 | $p < 0.001$ |





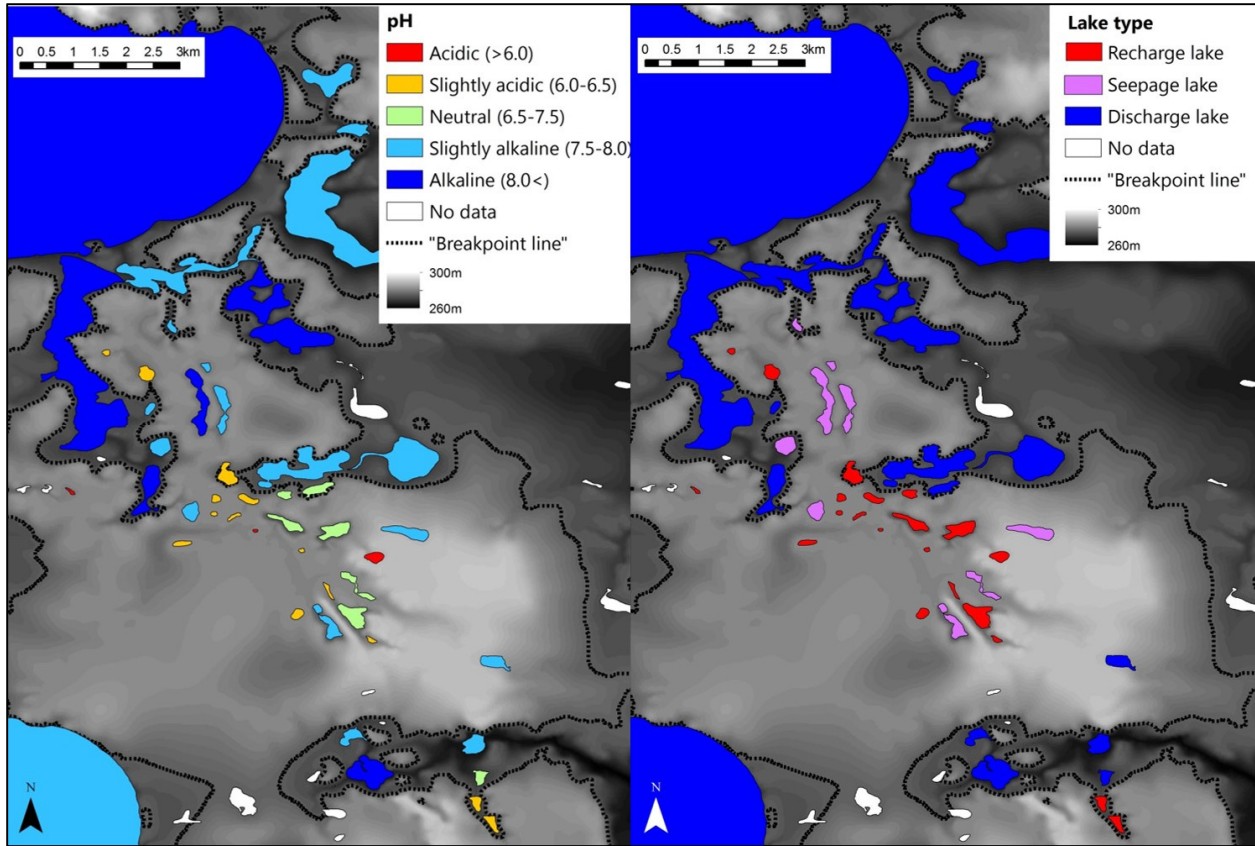

**Fig. A1**: Spatial depiction between elevation and lake-water pH (a) and lake type (b). The elevation of the breakpoint line (284 m) is shown as a dashed line.

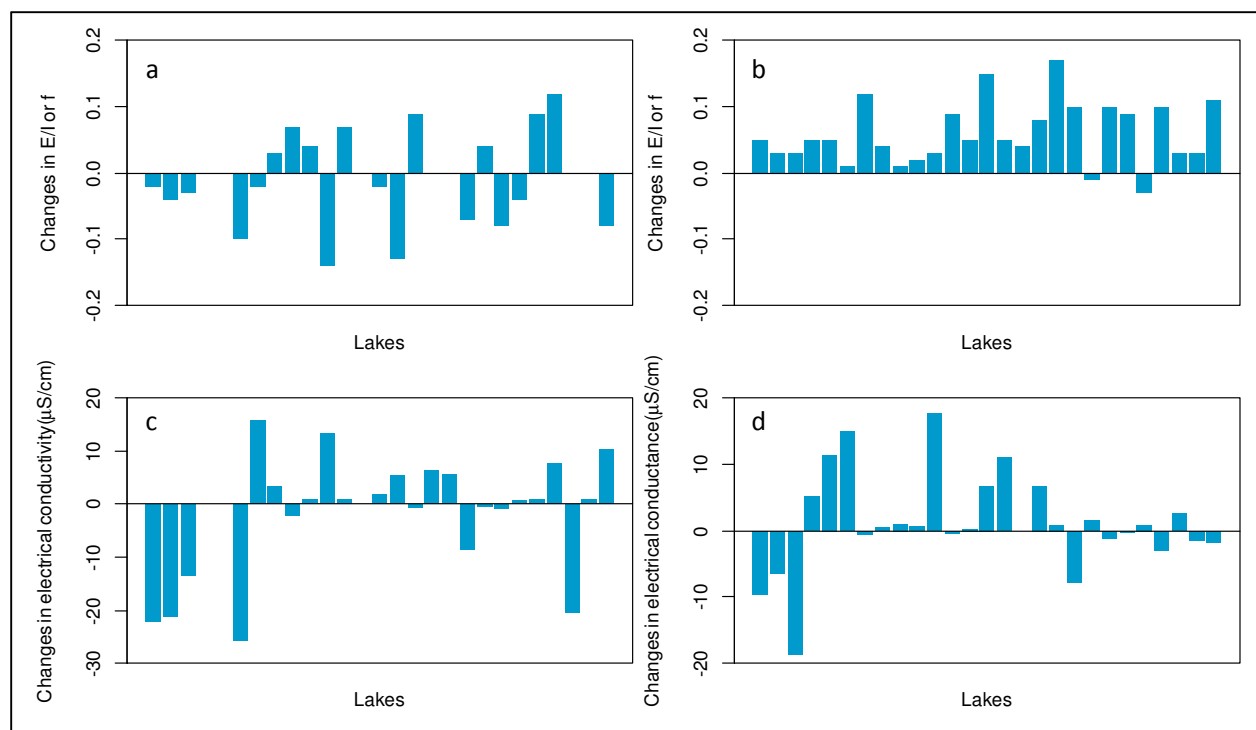

**Fig. A2**: Changes in E/I or f between June 2013 and June 2014 (E/I or f 2014 – E/I or f 2013) (a) and
changes in E/I or f between August 2014 and June 2014 (E/I or f Aug – E/I or f Jun) (b), and changes in
specific conductance between June 2013 and June 2014 (EC 2014 – EC 2013) (c) and changes in electrical
conductance between August 2014 and June 2014 (EC Aug – EC Jun) (d). Lakes are organized along an
elevation gradient (lower to higher elevation).