# Peer review of "Use of water isotopes and chemistry to infer the type and degree of exchange between groundwater and lakes in an esker complex of northeastern Ontario, Canada"

_Hydrology and Earth System Sciences, 2017_

## Referee Comment (RC1) · Anonymous Referee #1 · 20 Sep 2017

General comments

This manuscript describing lake chemistry and physiography relative to elevation and positioning relative to a nearby esker is impressive in its scope (∼50 lakes) and number of variables. The authors nicely show that lakes high in the landscape serve to recharge groundwater and lakes lower in the landscape receive groundwater discharge. Although impressive in the amount of data collected, the overall result of the study is not novel or surprising. The authors state that understanding lake position and relative sources and losses of water are important to lake stability in response

to climate change. Their overall hypothesis is that exchange with groundwater is the most important factor related to the classification that they develop, but they rely on indirect evidence (chemistry and isotopes) to determine the type of lake and degree of groundwater-surface-water exchange. Although interesting, their study does not address uncertainty and lacks confirmation of the extent to which chemistry and isotopes can serve in this function. I don't disagree with their conclusions; they seem logical and well founded. However, discussion about implications of assumptions and the possibility of misinterpretation of lake setting is missing. A slight change to the title may be more representative of the focus of this manuscript, which was to use lake water chemistry and water isotopes to infer exchange between groundwater and lakes. Rather than "Landscape and groundwater controls over boreal lake water chemistry and water balance heterogeneity. . .", it might be more accurate to title the study "Use of water isotopes and chemistry to infer the type and degree of exchange between groundwater and lakes in an esker complex of northeastern Ontario, Canada." A substantial product is the development of a classification scheme based on the authors' results. Numerous other classifications also have been proposed and it would be useful to compare the authors' classification system with one or more of the other lake-classification systems that have bee published. Several that come to mind are: Born, S.M., Smith, S.A., and Stephenson, D.A., 1979, Hydrogeology of glacial-terrain lakes, with management and planning applications: Journal of Hydrology, v. 43, p. 7-43. Bracht-Flyr, B., Istanbulluoglu, E., and Fritz, S., 2013, A hydro-climatological lake classification model and its evaluation using global data: Journal of Hydrology, v. 486, no. 0, p. 376-383. Martin, S.L., Soranno, P.A., Bremigan, M.T., and Cheruvelil, K.S., 2011, Comparing hydrogeomorphic approaches to lake classification: Environmental Management, v. 48, no. 5, p. 957. Wentz, D.A., 1981, Lake classification - Is there method to this madness?: U.S. Geological Survey Circular C0848-B, 15-24 p. Winter, T.C., 1977, Classification of the hydrologic setting of lakes in the north central United States: Water Resources Research, v. 13, no. 4, p. 753-767. This manuscript appears to have been hastily prepared and the numerous errors were not only a distraction, in some cases they led

to substantial confusion. The first example appears in the abstract, where the authors mix up groundwater recharge and groundwater discharge, calling lakes situated high in the landscape groundwater discharge lakes, and lakes situated low in the landscape groundwater recharge lakes. At first, I thought the authors were simply viewing the world from a lake perspective, where groundwater flowing to a lake would provide a recharge function to a lake. But no, the authors did adhere to the common usage for the terms groundwater recharge and discharge; they simply reversed terms in the abstract. I could easily overlook such a mistake, but so many other minor mistakes also are sprinkled liberally throughout the manuscript that they represent a considerable distraction when attempting to follow the logic and conclusions of the manuscript. I provide indications of mistakes where they appear in the first few pages, but I leave it to the authors to carefully read their own manuscript and clean up the rest.

Specific comments pertaining to line numbers

20 Higher elevation lakes are usually groundwater recharge lakes and lower elevation lakes are usually groundwater discharge lakes. Here you state the opposite, which made me think you were viewing flow of groundwater to a lake as providing a recharge function to the lake. Based on the subsequent context of the manuscript, you clearly need to correct this as follows: ". . . higher-elevation groundwater recharge lakes from lower-elevation groundwater discharge lakes." However, this provides an excellent example of how these terms are ambiguous and could easily mean the opposite from the perspective of a lake-centric scientist, who would view the addition of groundwater to a lake as serving a recharge function to the lake. Therefore, you need to define these terms in the introduction or the methods section so the reader clearly knows what you mean. You also use the terms discharge lake and drainage lake interchangeably. This is confusing. Since you first present the term discharge lake, and it is nicely contrary to recharge lake, I suggest you remove the term drainage lake and replace it everywhere with discharge lake.

72-73 You state here that you can determine evaporation and water balance provided

the Craig-Gordon model is used. You state that the Craig Gordon model has to be used to do this, but then you surprisingly cite a paper published in 2015 the title of which implies it is all about software called Hydrocalculator. If you are using the Craig-Gordon model, you should cite Craig and Gordon, 1965. You also need to revise the sentence to indicate that this particular model is either somehow better or more convenient for accomplishing these purposes than the now standard method that is commonly used.

78 You state there are few studies that have used both chemistry and water isotopes, and then you cite 4 such studies. In addition to those four studies, I can think of several more without even perusing the literature. This is not a "very limited number" especially relative to the number of studies in boreal lakes, which likely is smaller. Gurrieri, J.T., and Furniss, G., 2004, Estimation of groundwater exchange in alpine lakes using non-steady mass-balance methods: Journal of Hydrology, v. 297, p. 187–208. Krabbenhoft, D.P., Bowser, C.J., Kendall, C., and Gat, J.R., 1994, Use of oxygen-18 and deuterium to assess the hydrology of groundwater-lake systems, in Baker, L.A., ed., Environmental Chemistry of Lakes and Reservoirs: American Chemical Society, p. 67-90. LaBaugh, J.W., Winter, T.C., Rosenberry, D.O., Schuster, P.F., Reddy, M.M., and Aiken, G.R., 1997, Hydrological and chemical estimates of the water balance of a closed-basin lake in north central Minnesota: Water Resources Research, v. 33, no. 12, p. 2799-2812. Katz, B.G., Coplen, T.B., Bullen, T.D., and Davis, J.H., 1997, Use of chemical and isotopic tracers to characterize the interactions between ground water and surface water in mantled karst: Ground Water, v. 35, no. 6, p. 1014-1028. Turner, J.V., and Townley, L.R., 2006, Determination of groundwater flow-through regimes of shallow lakes and wetlands from numerical analysis of stable isotope and chloride tracer distribution patterns: Journal of Hydrology, v. 320, no. 3-4, p. 451-483. 148-149 Evidently you are only considering watershed slopes for the terrain that is within 100 m of the lake shoreline. For most lakes, this will be a small percentage of the lake watershed. You need to indicate why you are applying this 100-m filter to your analysis.

234-235 Most studies that use water isotopes to look at water budgets go to considerable effort to measure or estimate $\delta E$. Here you simply assume that $\delta E = \delta I$ minus $\delta L$. This method assumes that you know all of the other input and loss volumes and their integrated isotopic signatures. You should address this issue and either indicate that your assumed method for determining $\delta E$ is well supported by other studies or you should state that this method includes uncertainty associated with all of those other water-budget terms.

244 Because this manuscript is about groundwater and lakes, you should indicate what the groundwater temperatures are in this area. Temperatures of springs are not an accurate indication of groundwater temperature because spring temperature at the discharge point is so influenced by the velocity of the discharging spring and the resulting warming of the discharging water as it approaches the surface. Secondly, is there a difference in lake temperature between the higher and lower lakes? If you have substantially more groundwater discharge in the lower lakes you may see a resulting temperature difference in lake temperature. This is unlikely, however, because so many other factors also affect temperature, but with 50 lakes there may be something.

256-257 It would be very useful if you could include here a figure that maps the lakes with indications of the isotopic value (select 18-O or deuterium) and perhaps also the specific conductance. This would allow the reader to see the spatial distribution of the various isotopic values that are indicated in this paragraph. If you do this, please also show the lakes that have a stream outlet and indicate where that stream goes. As I read farther I see that Figure 5 does a reasonable job of showing this but it lacks mapping of the isotopic lake values. Ultimately, what would be most useful is a map showing the resulting distribution of recharge, seepage, and discharge lakes, perhaps overlain on a map that uses shading to show elevation.

266 The slope of your relation is negative but the r-squared value should be positive.

266 It is not the geographic position that is related. It is the elevation that is related. You

may infer from the elevation the position of the lake relative to the crest or the distance from the crest. If that is the case, you should state that here.

268-269 You state the relation between isotopes and elevation. You should, therefore, also provide a sentence and state the relation between EC and elevation.

277 Why are you relating E to I? You stated in the Methods that you can do this but you didn't say why. You should explain what this accomplishes. Also, since you're determining delE from delI minus delO, your equation 5 is really delI-delL/delI-2delL.

277-278 A small E/I could also simply indicate that the lake has an outlet (I∼Q) and, therefore, a short residence time, which has nothing to do with any indication of whether groundwater or springs undergo evaporation or not. Interpretations related to E/I also incorporate error associated with calculating delE rather than measuring it.

Figure 4 I don't understand how you can obtain E/I values for springs and streams when your two variables for making this calculation are delI and delL. What lake do you use to determine delL for a spring or a stream?

297 You can't have a negative R-squared value. These values must instead be correlation coefficients, which are usually indicated as lower-case r. Also, where do you present these values? DOC is not included in the correlation matrix in Table S1b in the Appendix. You do show TN but the value for r when TN is related to H (elevation) is 0.05, not 0.22 as reported here. And since we're talking about the table, why is the table labeled S1b? Is there a table S1A that is missing?

315 What is d in Table 1? Is it deuterium excess? If so, you have not defined or describe this anywhere.

320 Be sure to clearly show any surface-water connections between lakes. The presence of an outlet, or of surface-water flow from one lake to another, is an important part of your analysis and up to now the reader has not been able to see these connections. You could show this either here or in Figure 1.

325 You use an elevation of 242 m as the dividing line between recharge and discharge lakes. However, in Figure 5, and in the previous paragraph, you show a middle elevation, or an elevation break point of just over 280 m. And there is no elevation in Figure 5 lower than 260 m in Figure 5. If all of the lakes shown in Figure 5 are higher than 260 m, and your previously determined break-point elevation was 282.4 m, how can you now choose an elevation of 242 m to separate your lakes?

327 What additional lake characteristics were used to allow you to divide the lakes into three categories?

336 How does this value of 242 m relate to your break-point elevation of 282.4 m? If indeed these are two separate distinguishing elevations, you need to describe how one is important relative to the other.

370-371 Here you finally indicate that your streams are all upgradient of a lake and that you evidently did not sample any stream reaches that were situated between lakes. This is an important distinction that needs to be made clear much earlier in the manuscript.

416-417 You should not assume that a recharge lake is perched. I greatly doubt that lakes situated in permeable material such as you have in the esker are perched. Unless you have separate information indicating otherwise, you should remove the word perched from the manuscript.

433 Here you do report values as R rather than R-squared.

444 The presence or absence of an outlet is a very important variable in this analysis and this is the first time you have brought up this point. I would think this would be a presence-absence type of variable that would have been part of your statistical analysis. In any case, I was looking for outlets in the figures and it was very difficult to determine whether you showed all of the outlets or not. For some lakes, based on their positioning I would expect an outlet and yet one is not shown in the figure. If you

do show all of the stream segments in your study area, there is a surprisingly small number. If not, you should revise the figure to do so. The presence of both inlets and outlets is important to this type of analysis.

455 You should label and show in a map where these two lakes are.

455-460 A lake without an outlet could also be due to the lake being able to easily lose water to groundwater.

473 What is a mounded recharge lake? This needs to be defined. If this is a lake that loses water everywhere to groundwater, you can't call that mounded unless you have hydraulic gradient data to confirm that. It may instead be that the lake is situated in a small local watershed or that it has more organic sediments than others that creates a low pH. Figure 8 What distinguishes a recharge seepage lake from a discharge seepage lake? Is it only the pH? You talk considerably about recharge seepage lakes and the reasons for their classification, but you say nothing about discharge seepage lakes or how they are different from recharge seepage lakes.

520 Here you finally define a drainage lake. Given that this indeed is the same as a discharge lake, I suggest you use one term or the other and do not use both interchangeably.

545-551 Greater lake evaporation also is related to more isotopically enriched water in lakes. If there is lake-to-lake variation in evaporation, that variation affects your interpretation of the extent to which the lake exchanges water with groundwater. Your classification system does not provide any indication of the extent to which other factors affect your interpretation of exchange with groundwater. It would be helpful to point this out in this paragraph.

567-568 You cannot make this statement unless you either have evaporation data to back this up or you can cite some papers that have reached this conclusion. I strongly doubt that evaporation from lakes that receive groundwater discharge is measurably

different from lakes that recharge groundwater. What is different is the degree of iso-
topic enrichment in the lake water. The evaporative enrichment is masked by isotopi-
cally light groundwater.

591-592 This logic is not complete. If drought reduces groundwater flow to a lake
because of reduced water-table elevation, why then would there not be increased flow
of lake water to groundwater on the downgradient side of the lake, in which case there
would be no net change in exchange with solutes?

598-599 Residence time would be much more greatly controlled by local hydraulic
conductivity. If most of your recharge lakes are situated on the esker, then I would
think residence time would be relatively short compared to a lake lower in your system
that is situated in lower-permeability sediments.

607-614 This paragraph is poorly presented and is confusing. What do you mean by
multiple site selection? What sites are you talking about and for what are they being
selected? How is this related to paleohydrological reconstruction?

Technical corrections

22-23 You need to place a comma after contrast and remove the comma after lakes
so the sentence reads "In contrast, groundwater discharge lakes were isotopically de-
pleted . . ." 31-32 You should remove the parentheses here; otherwise, the sentence
is not complete. Also, how can you determine seasonal differences in chemistry when
you only have data between early June and August? That period is during summer
only. 34-35 Recharge lakes also are affected by changes of greater duration and per-
sistence. Please add the word "only" to write ". . . affected only by hydroclimatological
changes of greater duration and persistence." 47-48 The threat of climate change will
not influence lakes. It is the climate change that will influence lakes. Therefore, remove
"The threat of" and start the sentence with "Future climate change will. . ." 49 A word
is missing. Insert "understand" to write ". . . to better understand relationships of lake
hydrology . . ." 55 Another word is missing. Insert "to" to write ". . . the opportunity to

better understand . . ." 64 Now you have an extra word. Delete the word "changing". 70 Change tracers to tracer because it needs to be singular here. 72 What is a "time and cost effective means"? Perhaps you mean "an efficient and cost-effective means" 98 You state this is near Timmins but you do not show Timmins anywhere on the map. Please add the location of Timmins to the map. You also need to add labels to show the locations of Frederick House and Night Hawk Lakes. 110 What is a popular? I suspect you actually mean poplar. 124 I suggest you add the word littoral to write ". . . forming lateral littoral sand units that drape . . .". This addition will provide a better tie with the "littoral sand" indicated in Figure 1. 125 You have an extra and unnecessary word here. Delete the word "in" to write "The numerous kettle lakes on the esker formed . . ." 128 Why does only the esker crest have high-K sediments? I would think the entire esker would be much higher in K than your surrounding sediments. If it is just the upper portion of the esker that serves as an aquifer, you need to explain why this esker is geologically different from other eskers. 145 Depths of what? You should write "maximum lake depths" so the reader will know you are not talking about depths of other geographic features. 147-148 What does spatial resolution have to do with lake-surface elevation? This needs a better explanation. 153 You were only there for one week each time? That certainly does not constitute a season. You should replace the word season with something else; perhaps field campaigns or field trips. 154 Earlier you state that the number of lakes is 50, but here you say that this number is approximate. Surely you know how many lakes you sampled and you should provide that number. Otherwise, this text gives the impression that you can't be bothered with counting up the lakes you sampled or that you were not very meticulous. I suspect this is not the case, but this and the numerous previously identified simple mistakes certainly give a reader pause. 163 The word is triple-rinse, not tripled- rinsed. This mistake is repeated on line 164. 164 You say bottles were "and again with sample water," which I presume to mean that you triple-rinsed the bottle with sample water after the bottle was triple rinsed with distilled water. But then in the next sentence you write that bottles were triple-rinsed with lake water. Is this simply a mistake or redundancy

in your writing or did you rinse the bottles in lake water twice? 170 Apparatus should be replaced with apparati because the following word, "were", is plural. 171 Filtrate is singular but that word is followed by were, which is plural. Either change filtrate to filtrates or change were to was. 172 One does not store into. The samples were either stored or they were placed into. 198 Change detections to detection. 201 Detections limits should be change to detection limit. 202 You already defined these three terms in lines 177-178 so you do not need to do so a second time here. 203 What is NDIR? You need to indicate what this abbreviation indicates. Also, since it is only used once in the manuscript, it would be better to simply indicate what this is. 204 Change detections to detection. 207 You need to indicate what TIC is. I assume it is total inorganic carbon. If so, how was this determined? You do not say. Is TIC only the dissolved portion of the sample? 269-270 You could consider deleting this sentence. It is fairly standard practice to just use 18-O (most common) or 2-H and you've already shown the strong correlation in Figure 2. You might also modify slightly to write ". . . are enriched in 18-O, and due to their strong correlation, 2-H." 288 Delete "it" to write "As is the case with stable isotope values. . ." 332-333 This sentence does not make sense. I can't tell if there are missing words, or if infer is supposed to be inter. In any case, the text in quotations needs to be corrected: "In order to better understand the contribution of infer water chemistry variables," 342 The table caption states that some lakes are labeled in red. The figure is in black and white (although Figure 5 is in color) so nothing is shown in red here. 349 Panel A shows inter-annual change and panel B shows variability over a single summer; the citation should be to Fig. 7a,b). However, "seasonally" implies comparisons over multiple seasons. Your analysis is not seasonal. You only compared values over parts of one season, that being summer. Therefore, you should replace any inference to a seasonal analysis to something like "summertime" or "over-summer". 355 Here you again refer to panel B but the solute-related panels are 7c and 7d. 393 Please remove the hyphen between lake and water. 428 Replace delivery with deliver. In fact, this entire paragraph could use considerable help. 448 Here you use lower-case r. Thus far you have used R, R-squared, r, and r-squared. Do

they all mean the same thing in this manuscript? They certainly mean different things in statistics. You need to thoroughly review the manuscript and present your variables and statistical indicators in a consistent manner. 455-471 There are numerous problems with sentence structure, grammar, and lack of clarity in meaning. 487-489 I have no idea what this means; it seems completely out of context. Please revise for clarity. 554-555 You use relation and relationship interchangeably. I suggest you select one word and use it consistently. I was once told that people have relationships but data indicate relations. 587 This should be Figure 7B, not Figure 8B.

---

## Referee Comment (RC2) · Anonymous Referee #2 · 28 Sep 2017

Landscape and groundwater controls over boreal lake water chemistry and water balance heterogeneity in an esker complex of northeastern Ontario, Canada

Boreux et al.

Review: Boreux et al. utilize lake water isotope and chemistry data to identify lake types in an esker complex in Ontario, and specifically characterize the role of groundwater and landscape position on lake conditions. I enjoyed reading the paper. This is an excellent dataset and, by and large, the interepretations appear to be sound and

supported by the data. So, from an overall perspective, this paper will make for a useful contribution on lake hydrology. In particular, the links between water isotope composition and water chemistry serve as useful example of the strength in such an approach to characterize lake conditions at numerous locations across landscapes. That being said, there are both analytical and structural issues with this manuscript that, in my view, should be addressed. I elaborate on these below.

1. Analytical. Specifically here, I refer to calculation of the isotope-inferred E/I ratios. If the authors intend on retaining this piece of the manuscript, much more detail needs to be provided as to how these calculations were performed. For instance, at the top of page 9 where the E/I ratio is provided, there is an incorrect definition of dE. Several of the papers listed in the reference utilize the Gonfiantini (1986) equation, but I do not know if the authors used this equation. Furthermore, there are no details provided regarding how they computed dA, or what they used for temperature or relative humidity. Fundamentally, insufficient text and details are provided, such that the calculations cannot be reproduced by a reader. So, much more needs to be provided here. But I wonder if this is even necessary? The E/I results are barely mentioned in the Discussion. It seems that, for the apparent purposes of this paper, the authors could simply get away with comments referring to differences in the degree of evaporative isotopic enrichment.

Perhaps it is just my weak understanding of statistics, but I would like to see a bit more background on the breakpoint analysis. This seems to be a very important part of the paper, as it apparently identifies 282 m (or is it 242 m? – both are listed, which added to my confusion) as the landscape position in which the role of groundwater changes. My limited understanding of breakpoint analysis is that it is used to detect when there is a change in a trend (often applied to time-series measurements), but it seems like this analysis is being applied differently here.

2. Structural. In its present form, the manuscript imposes the lake classification scheme on the results, but it is only part-way through the Discussion where the ra-

tionale for the classification scheme is presented. This creates a lot of awkwardness with the paper. The reader is quite literally forced to take the author's word for it that there is some basis for the classification scheme while they read the results (but they have no idea what that is). Since my expertise lies more with the isotope component of the paper, I kept wondering how the authors are going to distinguish the role of groundwater versus precipitation, since they often have very similar isotope compositions. It was only much later, in the Discussion, where I learned that the role of groundwater, in fact, is largely, maybe exclusively (?) based on the water chemistry. There needs to be some re-structuring of the manuscript, so that the reader can examine the results, unencumbered by the imposing of the classification scheme, and then use those results to develop the classification rationale. I think you are likely to end up with the same ultimate intepretations and conclusions, but the path to getting there needs refinement and re-organization so that it will be easier for the reader to digest.

Other comments are listed below:

Line 17: Insert a sentence explaining why the study was conducted. Line 21-22: Interesting that evaporation did not lead to concentration of ions. Seems like an unexpected relation. Line 28: 'characteristics' Line 33: 'discharge lakes showed' Line 34: 'would only be likely affected' Line 64: delete 'changing' Line 84: replace 'to test' with 'examine' . . .. at this point, I'm wondering how the authors might distinguish influence of groundwater from precipitation since they often have similar isotopic signatures. Authors might want to acknowledge this challenge here, and may use this to also rationalize the combined use of water isotopes and chemistry. Line 91-92: re 'water management and conservation goals' . . . some mention of this applied aspect should be stated in the abstract, and furthermore, the abstract might relay more directly how the findings from this study contribute to these aspects Figure 1 caption: 'northeast'; label 'a' and 'b' on figure and label lakes and locations mentioned at the top of page 5 Line 107: delete 'the' Line 129: 'texture. The esker' Line 133: 'aquifers at its edges where' Line 153-154: Can you explain the three-time sampling rationale? This puzzled me a bit because the data are weighted to the early ice-free season, as mostly mean values are used, and there is very little mentioned about seasonal patterns in the data that are obtained. Line 209: As stated above in my comments, I wonder if you even need to perform these calculations. If you would like to retain, more details are required as per comments above. Line 233: Another point of confusion for me was that dI was assumed to be the average of precipitation, and was assumed to be the same for all lakes. Yet knowing that the goal here is to detect influence from groundwater, I wondered then if there might be some variability in dI among your lakes. And, of course, I remained curious to know how groundwater influence might be distinguished from precipitation since they often have similar isotope signatures. I'm not sure any of this was ever explicitly discussed in the manuscript. Line 234-235: Incorrect calculation of dE. Very puzzled by this statement. Line 242-244. I gather you are summarizing here, but you have two springtime samples and one late summer sample, which makes this a bit awkward. Line 250: 'Local Evaporation Line' Line 251-252: Concern about overlapping isotope signatures of gw and precip confirmed! Line 256-257: Awkward incorporation of elevation here. Seems like this should be saved for the next paragraph. Figure 2b: Label your figures 'a' and 'b'. First awkward imposing of lake type here, without rationale being provided. Line 266: A smalll thing but I would replace 'indicating' with 'suggesting'. Of course, this may be just one factor driving the water balance. At this point in the article, it seems the authors are overly anxious to get to the conclusion without proper development of the results and interpretation. Figure 3: I don't understand the application of the breakpoint analysis. Line 278: delete 'high' Line 279: An E/I ratio of '8' cannot be quoted so directly if using a steady-state model. Line 280: More awkward imposing of lake types without rationale. Same with Figure 4, which isn't a very effective way of presenting E/I ratios, as they are calculated from the isotope measurements. Line 296: Are they really conservative isotope tracers if they are changing as a consequence of evaporation? Table 1. More confusion about the breakpoint analysis, confounded by listing of ∼'282' m in the table and '242' m on p. 14 and elsewhere. Lines 323 to the end of this section: Again, imposing the lake

types here is simply not warranted. Extremely confusing to the reader. Data needs to be presented first. Then it can be interpreted. Then a classification system can be developed. P. 15. At this point in the paper, I had too many questions to be able to critically evaluate further. But as I say in my opening comments, I believe the story may largely be ok, it is just the development of that story that really needs to be overhauled. Figure 7. First (and perhaps only) characterization and use of the seasonal differences (although literal use of the very elevated E/I ratios is beyond the limitations of the model). Difficult to detect if this is really utilized in the Discussion. Is it relevant? Or can more be made of it? But again, as written, the reader is forced to accept the lake type classification. Line 369: replace 'high' with 'short' Line 373-374. So now we're finally getting some explanation of the different lake types, but this is confusing too. How do you know that gw is causing the difference here (which is not explicitly stated)? Line 379: replace 'composition' with 'water chemistries' Line 384, 386: How do you know the 1 km statements? Where does that come from? But here, I get the sense that the gw role is mainly based on the water chemistry differences, and less so on the isotope data. Line 404: This classification section has been pre-empted by all the comments before this point on lake types, which is the most awkward aspect of this paper. Sorry to keep emphasizing this. I'm sure it can be resolved! Line 407: 'conservative water tracers (water source)' – I am very confused by this terminology. The isotope data reflects more than just source water. It is modified by varying degrees of evaporative enrichment. Line 428: Awkward sentence. Line 436- 437: 282? Or 242? But I still don't understand how this elevation was defined. Line 451-453: Not a sentence. Line 473: What is meant by the word 'mounded'? Line 545: 'those'? Line 575: 'extent' Line 600-602: Interesting. Shouldn't these activities be mentioned in the Intro to help rationalize the study? Line 632: Awkward phrasing. Line 635: delete 'are' I like the conclusion – well stated and summarized!

---

## Author Comment (AC1) · 17 Nov 2017

**Response to reviewer #1 comments on "Landscape and groundwater controls over boreal lake water chemistry and water balance heterogeneity in an esker complex of northeastern Ontario, Canada" by Boreux et *al*.**

In black: reviewer's comments.
In blue: our answers and/or what we propose to add or to change in the manuscript.

We thank the reviewer for his through comments. These comments helped us to realize that the way our data was interpreted, presented and discussed and consequently how we could improve the manuscript. We will follow the majority of the reviewer's comments and suggestions when preparing the revised version of this manuscript.

**General comments**

This manuscript describing lake chemistry and physiography relative to elevation and positioning relative to a nearby esker is impressive in its scope (∼50 lakes) and number of variables. The authors nicely show that lakes high in the landscape serve to recharge groundwater and lakes lower in the landscape receive groundwater discharge. Although impressive in the amount of data collected, the overall result of the study is not novel or surprising.

As mentioned our introduction, landscape position such as elevation is an important control on water chemistry and water balance. Thus we agree with the fact that the conclusion is "not surprising", but we argue that it is novel and important. This study: (1) uses a large data set to document the importance of landscape positon on lakes; (2) uses both conservative (isotopes) and non-conservative (chemistry) tracers to illustrate the importance of landscape positon on lake hydrology; (3) illustrates the robust nature of this data set since the data in this study were collected at three different times: and (4) develops a lake typology that provides important generalizations. Also, by carrying this study in an esker complex, it can be compared with similar studies on other esker complexes (such as in Fennoscandia), which are common in boreal regions. Also, this study is particularly relevant in the context of paleohydrology/paleoclimatic studies from this region, as allowed us to assess the response of lake position to past climate change, in reference to landscape position over the Holocene.

The authors state that understanding lake position and relative sources and losses of water are important to lake stability in response to climate change. Their overall hypothesis is that exchange with groundwater is the most important factor related to the classification that they develop, but they rely on indirect evidence (chemistry and isotopes) to determine the type of lake and degree of groundwater-surface-water exchange.

We agree that direct evidence of groundwater-surface water interactions is more powerful than indirect evidence. However, setting up direct measurements of groundwater-surface water interactions such as seepage meters in 50 lakes multiple time would have needed a significant amount of time and monetary resources that simply were not available. Water tracers such as chemistry and isotopes are indeed an indirect evidence of such interactions and they have been proven to be reliable indirect indicators in the literature in a wide variety of settings.

Although interesting, their study does not address uncertainty and lacks confirmation of the extent to which chemistry and isotopes can serve in this function. I don't disagree with their conclusions; they seem logical and well founded. However, discussion about implications of assumptions and the possibility of misinterpretation of lake setting is missing.

We agree that our manuscript could benefit from including the implications of the assumption and the probability of misinterpretation. This will be included in the revision.

A slight change to the title may be more representative of the focus of this manuscript, which was to use lake water chemistry and water isotopes to infer exchange between groundwater and lakes. Rather than "Landscape and groundwater controls over boreal lake water chemistry and water balance heterogeneity. . .", it might be more accurate to title the study "Use of water isotopes and chemistry to infer the type and degree of exchange between groundwater and lakes in an esker complex of northeastern Ontario, Canada."

We agree with the reviewer that our main objective was to use tracers to infer groundwater-lake interactions rather than studying the processes *per se* and that the title could be more representative of our study. We thus suggest to rename the paper as suggested by the reviewer: "Use of water isotopes and chemistry to infer the type and degree of exchange between groundwater and lakes in an esker complex of northeastern Ontario, Canada"

A substantial product is the development of a classification scheme based on the authors' results. Numerous other classifications also have been proposed and it would be useful to compare the authors' classification system with one or more of the other lake-classification systems that have been published. Several that come to mind are: Born, S.M., Smith, S.A., and Stephenson, D.A., 1979, Hydrogeology of glacial-terrain lakes, with management and planning applications: Journal of Hydrology, v. 43, p. 7-43. Bracht-Flyr, B., Istanbulluoglu, E., and Fritz, S., 2013, A hydro-climatological lake classification model and its evaluation using global data: Journal of Hydrology, v. 486, no. 0, p. 376-383. Martin, S.L., Soranno, P.A., Bremigan, M.T., and Cheruvelil, K.S., 2011, Comparing hydrogeomorphic approaches to lake classification: Environmental Management, v. 48, no. 5, p. 957. Wentz, D.A., 1981, Lake classification - Is there method to this madness?: U.S. Geological Survey Circular C0848-B, 15-24 p. Winter, T.C., 1977, Classification of the hydrologic setting of lakes in the north central United States: Water Resources Research, v. 13, no. 4, p. 753-767.

We agree with the reviewer that comparing our lake typology with other published typologies would be beneficial to the manuscript and we will do so.

This manuscript appears to have been hastily prepared and the numerous errors were not only a distraction, in some cases they led to substantial confusion. The first example appears in the abstract, where the authors mix up groundwater recharge and groundwater discharge, calling lakes situated high in the landscape groundwater discharge lakes, and lakes situated low in the landscape groundwater recharge lakes. At first, I thought the authors were simply viewing the world from a lake perspective, where groundwater flowing to a lake would provide a recharge function to a lake. But no, the authors did adhere to the common usage for the terms groundwater recharge and discharge; they simply reversed terms in the abstract. I could easily overlook such a mistake, but so many other minor mistakes also are sprinkled liberally throughout the

manuscript that they represent a considerable distraction when attempting to follow the logic and conclusions of the manuscript. I provide indications of mistakes where they appear in the first few pages, but I leave it to the authors to carefully read their own manuscript and clean up the rest.

After reading the reviewer's comments and the manuscript we realized the presence of mistakes throughout the manuscript, which were confusing. We apologize for that and will make the necessary edits. We indeed view the world from a groundwater perspective and adhere to the common usage for the terms groundwater recharge and discharge. We will correct this in the abstract and throughout the manuscript based on the reviewer's comments.

**Specific comments pertaining to line numbers**

20 Higher elevation lakes are usually groundwater recharge lakes and lower elevation lakes are usually groundwater discharge lakes. Here you state the opposite, which made me think you were viewing flow of groundwater to a lake as providing a recharge function to the lake. Based on the subsequent context of the manuscript, you clearly need to correct this as follows: ". . . higher-elevation groundwater recharge lakes from lower-elevation groundwater discharge lakes." However, this provides an excellent example of how these terms are ambiguous and could easily mean the opposite from the perspective of a lake-centric scientist, who would view the addition of groundwater to a lake as serving a recharge function to the lake. Therefore, you need to define these terms in the introduction or the methods section so the reader clearly knows what you mean. You also use the terms discharge lake and drainage lake interchangeably. This is confusing. Since you first present the term discharge lake, and it is nicely contrary to recharge lake, I suggest you remove the term drainage lake and replace it everywhere with discharge lake.

We agree with the reviewer that the terms recharge and discharge are ambiguous and we will define early in the manuscript what we mean by those terms. In this paper, we adhere to the common usage for the terms groundwater recharge and discharge and view the world from a groundwater perspective.

72-73 You state here that you can determine evaporation and water balance provided the Craig-Gordon model is used. You state that the Craig Gordon model has to be used to do this, but then you surprisingly cite a paper published in 2015 the title of which implies it is all about software called Hydrocalculator. If you are using the CraigGordon model, you should cite Craig and Gordon, 1965. You also need to revise the sentence to indicate that this particular model is either somehow better or more convenient for accomplishing these purposes than the now standard method that is commonly used.

The explanation of the method seemed more intuitive in Skrzypek et *al*. (2015) than in the original paper, thus the reason to cite the latter reference. However, we do agree with the reviewer that the original paper should be cited. The change of citation will be changed. The other reviewer pointed flaws in the section on the methodology to determined water balance and this section will be rewritten.

78 You state there are few studies that have used both chemistry and water isotopes, and then you cite 4 such studies. In addition to those four studies, I can think of several more without even perusing the literature. This is not a "very limited number" especially relative to the number of studies in boreal lakes, which likely is smaller. Gurrieri, J.T., and Furniss, G., 2004, Estimation of groundwater exchange in alpine

lakes using non-steady mass-balance methods: Journal of Hydrology, v. 297, p. 187– 208. Krabbenhoft, D.P., Bowser, C.J., Kendall, C., and Gat, J.R., 1994, Use of oxygen- 18 and deuterium to assess the hydrology of groundwater-lake systems, in Baker, L.A., ed., Environmental Chemistry of Lakes and Reservoirs: American Chemical Society, p. 67-90. LaBaugh, J.W., Winter, T.C., Rosenberry, D.O., Schuster, P.F., Reddy, M.M., and Aiken, G.R., 1997, Hydrological and chemical estimates of the water balance of a closed-basin lake in north central Minnesota: Water Resources Research, v. 33, no. 12, p. 2799-2812. Katz, B.G., Coplen, T.B., Bullen, T.D., and Davis, J.H., 1997, Use of chemical and isotopic tracers to characterize the interactions between ground water and surface water in mantled karst: Ground Water, v. 35, no. 6, p. 1014-1028. Turner, J.V., and Townley, L.R., 2006, Determination of groundwater flow-through regimes of shallow lakes and wetlands from numerical analysis of stable isotope and chloride tracer distribution patterns: Journal of Hydrology, v. 320, no. 3-4, p. 451-483.

We do agree with the reviewer that there was not only a limited number of studies that used both water chemistry and isotopes to investigate groundwater-lake water connections. Our phrase was awkwardly worded in the sense that we meant few studies have used such a combined approach on the scale of the landscape and most studies used only one approach (isotopic or chemical). We clarify this in the revision, and provide appropriate citations.

148-149 Evidently you are only considering watershed slopes for the terrain that is within 100 m of the lake shoreline. For most lakes, this will be a small percentage of the lake watershed. You need to indicate why you are applying this 100-m filter to your analysis.

The coarse spatial resolution of the DEM and the close proximity of the lakes in the study area made it impossible for ArcGIS to compute the watershed delineation of each lake. Because all lakes are kettles (which are characterized by steep slopes on their shore over a small distance) and nearby, arbitrary buffer zone widths of 50 m and 100 m were tested and displayed similar correlations with lake isotopic composition: Max slope 100 m (r=-0.33, p=0.02), Max slope 50 m (r=-0.31, p=0.03). We will clarify in the revision.

234-235 Most studies that use water isotopes to look at water budgets go to considerable effort to measure or estimate $\delta E$. Here you simply assume that $\delta E = \delta I$ minus $\delta L$. This method assumes that you know all of the other input and loss volumes and their integrated isotopic signatures. You should address this issue and either indicate that your assumed method for determining $\delta E$ is well supported by other studies or you should state that this method includes uncertainty associated with all of those other water-budget terms.

We agree with the reviewer that the provided equation was in error, and thank the reviewer for pointing out our mistake. We originally aimed at summarizing this section as much as possible as this method has been used in several papers and we accidently listed a wrong formula. Thus, more text and details will be provided and in the rewrite of this section (see response to reviewer #2).

244 Because this manuscript is about groundwater and lakes, you should indicate what the groundwater temperatures are in this area. Temperatures of springs are not an accurate indication of groundwater temperature because spring temperature at the discharge point is so influenced by the velocity of the discharging spring and the resulting warming of the discharging water as it approaches the surface.

Secondly, is there a difference in lake temperature between the higher and lower lakes? If you have substantially more groundwater discharge in the lower lakes you may see a resulting temperature difference in lake temperature. This is unlikely, however, because so many other factors also affect temperature, but with 50 lakes there may be something.

The Provincial Groundwater Monitoring Network of the province of Ontario of Canada (where the study area is located) monitor 3 groundwater wells in the area (water level and water chemistry) but does not report any water temperature measurements. Thus we do not have direct measurements of groundwater temperature.

We agree with the reviewer that "the temperatures of springs are not an accurate indication of groundwater temperature because spring temperature at the discharge point is so influenced by the velocity of the discharging spring and the resulting warming of the discharging water as it approaches the surface. However, by referring to the temperature of springs, we do not interpret it to be an indicator of the temperature of groundwater but we looked at the temperature of spring to infer that those were indeed springs (due to their lower temperature). Some spring samples were taken from wells build by locals, which fore sure introduce a bias in the data. This will be clarified in the revision.

Interestingly the temperature was recorded to test whether lakes that have a greater proportion of their water balance coming from groundwater had colder temperatures or not (even though we did not really expect to find such a relationship). This was not the case as supported by statistical analyses in the appendix. Lake size was correlated with water temperature as bigger bodies warm more slowly and cool down slower in comparison to smaller ones.

256-257 It would be very useful if you could include here a figure that maps the lakes with indications of the isotopic value (select 18-O or deuterium) and perhaps also the specific conductance. This would allow the reader to see the spatial distribution of the various isotopic values that are indicated in this paragraph. If you do this, please also show the lakes that have a stream outlet and indicate where that stream goes. As I read farther I see that Figure 5 does a reasonable job of showing this but it lacks mapping of the isotopic lake values. Ultimately, what would be most useful is a map showing the resulting distribution of recharge, seepage, and discharge lakes, perhaps overlain on a map that uses shading to show elevation.

Because the E/I calculation is based on $\delta^{18}O$, we didn't include a map with the lake isotopic values as we thought this would be redundant. However, we acknowledge the fact that this might be of interest to some readers. The addition of a map showing the distribution of lake types in the landscape is an interesting suggestion. Because we are concerned with not overcrowding the manuscript with figures, we will add those this additional information on a more complex figure as supplemental information.

266 The slope of your relation is negative but the r-squared value should be positive.

This is a major typo. Coefficients of determination cannot have negative values. Following the reviewer's comments, we will make the necessary changes throughout the manuscript and present the correlation coefficient.

266 It is not the geographic position that is related. It is the elevation that is related. You may infer from the elevation the position of the lake relative to the crest or the distance from the crest. If that is the case, you should state that here.

Correct. Elevation that is related to water isotopic composition and chemistry. Geographic position was used as a synonym but we agree that this terminology was confusing, and will correct.

268-269 You state the relation between isotopes and elevation. You should, therefore, also provide a sentence and state the relation between EC and elevation.

We agree with the reviewer and will state the relation between EC and elevation as well.

277 Why are you relating E to I? You stated in the Methods that you can do this but you didn't say why. You should explain what this accomplishes. Also, since you're determining delE from delI minus delO, your equation 5 is really delI-delL/delI-2delL.

We can relate E to I assuming that the lakes are in isotopic steady state. This assumption seems well justified as the lakes have had a lot of time in the past to reach their isotopic steady-state which is reflective of the local climate and their mean hydrological status, and can be defined by its water balance (which corresponds to the ratio of the total inflow to the evaporation rate, Isokangas et al., 2015). As also pointed out by the other reviewer, the formula for δE has been corrected, and the methodology for the calculation of water balance has been rewritten (see response to reviewer #2).

Reference: Isokangas, E., Rozanski, K., Rossi, P.M., Ronkanen, A.K. and Kløve, B.: Quantifying groundwater dependence of a sub-polar lake cluster in Finland using an isotope mass balance approach, Hydrol. Earth Syst. Sci., 19, 1247-1262, https://doi.org/10.5194/hess-19-1247-2015, 2015.

277-278 A small E/I could also simply indicate that the lake has an outlet (I~Q) and, therefore, a short residence time, which has nothing to do with any indication of whether groundwater or springs undergo evaporation or not. Interpretations related to E/I also incorporate error associated with calculating delE rather than measuring it.

We agree with the reviewer's comment that residence times are key here and will address this.

Figure 4 I don't understand how you can obtain E/I values for springs and streams when your two variables for making this calculation are delI and delL. What lake do you use to determine delL for a spring or a stream?

In that case we treated the spring water samples as if they were a spring or a lake and used the same methodology. We will clarify in the methods that we used δG for groundwater and δS for springs.

297 You can't have a negative R-squared value. These values must instead be correlation coefficients, which are usually indicated as lower-case r. Also, where do you present these values? DOC is not included in the correlation matrix in Table S1b in the Appendix. You do show TN but the value for r when TN is

related to H (elevation) is 0.05, not 0.22 as reported here. And since we're talking about the table, why is the table labeled S1b? Is there a table S1A that is missing?

Agree. We will use correlations. DOC is indeed included in Tab. S1b but under the label NPOC, we will change this so that the names of the variables is consistent throughout the manuscript. The errors have been corrected. All correlation values (r) are displayed in Tab S1a and b. There is no Tab A1. The journal asked us to rename Tab A1 to Tab S1 and not all notations were changed.

315 What is d in Table 1? Is it deuterium excess? If so, you have not defined or describe this anywhere.

d indeed corresponds to deuterium excess, we will specify and define it in the manuscript.

320 Be sure to clearly show any surface-water connections between lakes. The presence of an outlet, or of surface-water flow from one lake to another, is an important part of your analysis and up to now the reader has not been able to see these connections. You could show this either here or in Figure 1.

We will add the outlets in Fig 1 as suggested.

325 You use an elevation of 242 m as the dividing line between recharge and discharge lakes. However, in Figure 5, and in the previous paragraph, you show a middle elevation, or an elevation break point of just over 280 m. And there is no elevation in Figure 5 lower than 260 m in Figure 5. If all of the lakes shown in Figure 5 are higher than 260 m, and your previously determined break-point elevation was 282.4 m, how can you now choose an elevation of 242 m to separate your lakes?

The elevation of the breakpoint line is indeed 282 m, not 242 m (as accurately mentioned by the reviewer, there is no elevation lower than 260 in our study area). This is an unfortunate typo.

327 What additional lake characteristics were used to allow you to divide the lakes into three categories?

Other lake characteristics include the lake's pH, the presence or absence of an outlet, the lake size, depth and DOC content. This will be added in the text in brackets.

336 How does this value of 242 m relate to your break-point elevation of 282.4 m? If indeed these are two separate distinguishing elevations, you need to describe how one is important relative to the other.

The elevation is 282 m, not 242 m. A previous typo earlier in the manuscript made it confusing. This will be corrected.

370-371 Here you finally indicate that your streams are all upgradient of a lake and that you evidently did not sample any stream reaches that were situated between lakes. This is an important distinction that needs to be made clear much earlier in the manuscript.

We did sample upgradient streams as well as outlets that flow into other lakes (mainly into Frederick House Lake, which feeds the Frederick House River that drains the whole area). We will make this clearer early

in the manuscript as suggested. All the lakes mentioned in the manuscript will be labelled in the map as previously suggested and streams will be added to the map of the area as well.

416-417 You should not assume that a recharge lake is perched. I greatly doubt that lakes situated in permeable material such as you have in the esker are perched. Unless you have separate information indicating otherwise, you should remove the word perched from the manuscript.

We agree with the reviewer that the assumption that a recharge lake is perched is erroneous as esker are indeed made of permeable material. The term "perched" was wrongly used as a synonym of recharge, which is not. This will be corrected.

433 Here you do report values as R rather than R-squared.

As commented above, changes will be made and only r values will be reported.

444 The presence or absence of an outlet is a very important variable in this analysis and this is the first time you have brought up this point. I would think this would be a presence-absence type of variable that would have been part of your statistical analysis. In any case, I was looking for outlets in the figures and it was very difficult to determine whether you showed all of the outlets or not. For some lakes, based on their positioning I would expect an outlet and yet one is not shown in the figure. If you do show all of the stream segments in your study area, there is a surprisingly small number. If not, you should revise the figure to do so. The presence of both inlets and outlets is important to this type of analysis.

The presence or absence of an outlet was investigated with a logistic regression (lines 462-467). The figure will be revised to include all know outlets and streams in the area. However, the local topographic maps don't display all the streams observed in the field and in satellite imagery due to the presence of a dense forest. Thus the updated map might not be exhaustive of all streams in the area but all known streams and streams that appear distinctively in satellite images will be included.

455 You should label and show in a map where these two lakes are.

Those lakes will be labelled in a map in the appendix as suggested.

455-460 A lake without an outlet could also be due to the lake being able to easily lose water to groundwater.

We totally agree with the reviewer and this argument will be added and commented in the manuscript.

473 What is a mounded recharge lake? This needs to be defined. If this is a lake that loses water everywhere to groundwater, you can't call that mounded unless you have hydraulic gradient data to confirm that. It may instead be that the lake is situated in a small local watershed or that it has more organic sediments than others that creates a low pH.

Mounded lakes are lakes positioned above the water table that receive most of their water input from direct precipitation as defined by the ALSC Lake Classification System (Newton and Driscoll, 1990). We agree

with the reviewer that labelling those lakes as mounded without knowing the hydraulic gradient might be incorrect. Given this, we will remove this terminology. We agree with the reviewer that those lakes are just situated in a smaller local watershed.

Reference: Newton, R.M. and Driscoll, C.T.: Classification of ALSC lakes, in: Adriondack Lakes Survey: An Interpretive Analysis of Fish Communities and Water Chemistry, 1984-87, Adriondack Lakes Survey Corporation, Ray Brook, NY., 2-70 to 2-91, 1990.

Figure 8 What distinguishes a recharge seepage lake from a discharge seepage lake? Is it only the pH? You talk considerably about recharge seepage lakes and the reasons for their classification, but you say nothing about discharge seepage lakes or how they are different from recharge seepage lakes.

Those two lakes are distinguished by their position: in the groundwater discharge zone or the groundwater recharge zone inferred from the breakpoint analysis. This will be highlighted in the manuscript.

520 Here you finally define a drainage lake. Given that this indeed is the same as a discharge lake, I suggest you use one term or the other and do not use both interchangeably.

Agreed.

545-551 Greater lake evaporation also is related to more isotopically enriched water in lakes. If there is lake-to-lake variation in evaporation, that variation affects your interpretation of the extent to which the lake exchanges water with groundwater. Your classification system does not provide any indication of the extent to which other factors affect your interpretation of exchange with groundwater. It would be helpful to point this out in this paragraph.

We agree, and will include this in the discussion. Although lake morphometry can influence isotopic values, elevation accounts for much more variance in isotopic composition (r=0.53, p<0.001) and trumps the influence of variations in evaporation between lakes. We will address this in more details in the revised manuscript.

567-568 You cannot make this statement unless you either have evaporation data to back this up or you can cite some papers that have reached this conclusion. I strongly doubt that evaporation from lakes that receive groundwater discharge is measurably different from lakes that recharge groundwater. What is different is the degree of isotopic enrichment in the lake water. The evaporative enrichment is masked by isotopically light groundwater.

We agree with the reviewer's comment and we realized that that sentence was poorly worded. We will replace "changes in evaporation" by "changes in water balance".

591-592 This logic is not complete. If drought reduces groundwater flow to a lake because of reduced water-table elevation, why then would there not be increased flow of lake water to groundwater on the downgradient side of the lake, in which case there would be no net change in exchange with solutes?

We agree. Recharge lakes would feed the groundwater system by lake outseepage, which would produce no net change in solutes.

598-599 Residence time would be much more greatly controlled by local hydraulic conductivity. If most of your recharge lakes are situated on the esker, then I would think residence time would be relatively short compared to a lake lower in your system that is situated in lower-permeability sediments.

We agree with the reviewer that residence times can be related to the permeability of the material it is embedded in. However, residence times are a reflection of both watershed size and lake volume. In our study area, lakes at lower elevation tend to be much shallower and have larger watersheds, resulting in increased flushing. We will discuss this in the revision.

607-614 This paragraph is poorly presented and is confusing. What do you mean by multiple site selection? What sites are you talking about and for what are they being selected? How is this related to paleohydrological reconstruction?

Site selection is meant in reference to lake selection for a paleohydrology/paleoclimatic studies from the lake sediments. Hence, knowledge of present-day hydrology and connection to the groundwater is important. For example, choosing a groundwater discharge lake for a paleo study may show stability, whereas a higher elevation groundwater recharge lake may be more sensitive. Our study provides the modern-day context for lake selection.

**Technical corrections**

22-23 You need to place a comma after contrast and remove the comma after lakes so the sentence reads "In contrast, groundwater discharge lakes were isotopically depleted . . ."

This will be adapted in the text as suggested.

31-32 You should remove the parentheses here; otherwise, the sentence is not complete. Also, how can you determine seasonal differences in chemistry when you only have data between early June and August? That period is during summer only.

The parenthesis will be removed as suggested. As the reviewer mentions below, "seasonally" implies comparisons over multiple seasons. Technically, early June is still in the spring and August is in the summer. Plus, in those settings, summers are usually shorter than at lower latitudes so the notion of season still makes sense here. But at the same time, we agree that the usage of the term season is debateable. Therefore we will use the term summertime instead.

34-35 Recharge lakes also are affected by changes of greater duration and persistence. Please add the word "only" to write ". . . affected only by hydroclimatological changes of greater duration and persistence."

This will be adapted in the text as suggested.

47-48 The threat of climate change will not influence lakes. It is the climate change that will influence lakes. Therefore, remove "The threat of" and start the sentence with "Future climate change will. . ."

This will be adapted in the text as suggested.

49 A word is missing. Insert "understand" to write ". . . to better understand relationships of lake hydrology . . ."

This will be adapted in the text as suggested.

55 Another word is missing. Insert "to" to write ". . . the opportunity to better understand . . ."

This will be adapted in the text as suggested.

64 Now you have an extra word. Delete the word "changing".

This will be adapted in the text as suggested.

70 Change tracers to tracer because it needs to be singular here.

This will be adapted in the text as suggested.

72 What is a "time and cost effective means"? Perhaps you mean "an efficient and cost-effective means"

Yes, "time and cost effective means" will be replaced by: "an efficient and cost-effective means".

98 You state this is near Timmins but you do not show Timmins anywhere on the map. Please add the location of Timmins to the map. You also need to add labels to show the locations of Frederick House and Night Hawk Lakes.

The city of Timmins will be added to the map and the location of cited lakes will be labelled also.

110 What is a popular? I suspect you actually mean poplar.

The correction to poplar will be made. The error was likely due to an automatic correction by Word.

124 I suggest you add the word littoral to write ". . . forming lateral littoral sand units that drape . . .". This addition will provide a better tie with the "littoral sand" indicated in Figure 1.

This will be adapted in the text as suggested.

125 You have an extra and unnecessary word here. Delete the word "in" to write "The numerous kettle lakes on the esker formed . . ."

128 Why does only the esker crest have high-K sediments? I would think the entire esker would be much higher in K than your surrounding sediments. If it is just the upper portion of the esker that serves as an aquifer, you need to explain why this esker is geologically different from other eskers.

The reviewer is correct that the whole esker has high K-sediments. The word "crest" has no place here. This will be corrected in the text.

145 Depths of what? You should write "maximum lake depths" so the reader will know you are not talking about depths of other geographic features.

We will write "maximum lake depths" to make it clear to the reader.

147-148 What does spatial resolution have to do with lake-surface elevation? This needs a better explanation.

In ArcGIS, the elevation of a polygon (here a lake) can be obtained from a Digital Elevation Model using the spatial statistics extension. When requesting the calculation, the software asks whether you want to use the mean, max or min value and we selected mean.

153 You were only there for one week each time? That certainly does not constitute a season. You should replace the word season with something else; perhaps field campaigns or field trips.

The word season will be replaced by campaign as suggested.

154 Earlier you state that the number of lakes is 50, but here you say that this number is approximate. Surely you know how many lakes you sampled and you should provide that number. Otherwise, this text gives the impression that you can't be bothered with counting up the lakes you sampled or that you were not very meticulous. I suspect this is not the case, but this and the numerous previously identified simple mistakes certainly give a reader pause.

The word approximately will be erased. There was indeed 50 lakes sampled.

163 The word is triple-rinse, not tripled- rinsed. This mistake is repeated on line 164.

This will be adapted in the text as suggested in lines 163 and 164.

164 You say bottles were "and again with sample water," which I presume to mean that you triple-rinsed the bottle with sample water after the bottle was triple rinsed with distilled water. But then in the next sentence you write that bottles were triple-rinsed with lake water. Is this simply a mistake or redundancy in your writing or did you rinse the bottles in lake water twice?

Bottles were triple rinsed with distilled water off-site first, then triple rinsed with sample water on-site. We will reword this phrase to make it clearer.

170 Apparatus should be replaced with apparati because the following word, "were", is plural.

This will be adapted in the text as suggested.

171 Filtrate is singular but that word is followed by were, which is plural. Either change filtrate to filtrates or change were to was.

We will make the correction and change "filtrate" to "filtrates".

172 One does not store into. The samples were either stored or they were placed into.

We will make the correction and delete the word "into".

198 Change detections to detection.

This will be adapted in the text as suggested.

201 Detections limits should be change to detection limit.

This will be adapted in the text as suggested.

202 You already defined these three terms in lines 177-178 so you do not need to do so a second time here.

This will be adapted in the text as suggested.

203 What is NDIR? You need to indicate what this abbreviation indicates. Also, since it is only used once in the manuscript, it would be better to simply indicate what this is.

We agree with the reviewer's point. We will change NDIR to "nondispersive infrared sensor".

204 Change detections to detection.

This will be adapted in the text as suggested.

207 You need to indicate what TIC is. I assume it is total inorganic carbon. If so, how was this determined? You do not say. Is TIC only the dissolved portion of the sample?

TIC is indeed total inorganic carbon. This variable was actually not used in our analysis and is not mentioned anywhere else in the manuscript. It must have been left in the method section by accident.

269-270 You could consider deleting this sentence. It is fairly standard practice to just use 18-O (most common) or 2-H and you've already shown the strong correlation in Figure 2. You might also modify slightly to write ". . . are enriched in 18-O, and due to their strong correlation, 2-H."

We will delete this sentence as suggested. Even though we agree it is a standard practice to use $^{18}$O rather than $^2$H, we thought it might have been worthwhile to mention it in case the reader would wonder why we seem to omit referring to $^2$H.

288 Delete "it" to write "As is the case with stable isotope values. . ."

This will be adapted in the text as suggested.

332-333 This sentence does not make sense. I can't tell if there are missing words, or if infer is supposed to be inter. In any case, the text in quotations needs to be corrected: "In order to better understand the contribution of infer water chemistry variables,"

We agree with the reviewer that the sentence in quotation marks doesn't make sense. We will reword "In order to better understand the contribution of infer water chemistry variables," to "In order to better understand how water tracers vary in those two zones".

342 The table caption states that some lakes are labeled in red. The figure is in black and white (although Figure 5 is in color) so nothing is shown in red here.

We will modify this sentence as the figure is in black and white. The figure used to be in color in a previous draft and was changed to black and white for the journal. We will make the appropriate edits.

349 Panel A shows inter-annual change and panel B shows variability over a single summer; the citation should be to Fig. 7a,b). However, "seasonally" implies comparisons over multiple seasons. Your analysis is not seasonal. You only compared values over parts of one season, that being summer. Therefore, you should replace any inference to a seasonal analysis to something like "summertime" or "over-summer".

We will use the terms at the beginning and towards the end of the growing season to identify the June and the August campaigns. The text will be changed to refer to Fig 7 a,b (although this figure will be moved to the discussion, following the suggestion of the other reviewer).

355 Here you again refer to panel B but the solute-related panels are 7c and 7d.

The text will be changed to refer to Fig 7 c,d (although this figure will be moved to the discussion, following the other reviewer's comments).

393 Please remove the hyphen between lake and water.

This will be adapted in the text as suggested.

428 Replace delivery with deliver. In fact, this entire paragraph could use considerable help.

This will be adapted in the text as suggested. Edits will be made to improve the quality of the entire paragraph as suggested.

448 Here you use lower-case r. Thus far you have used R, R-squared, r, and r-squared. Do they all mean the same thing in this manuscript? They certainly mean different things in statistics. You need to thoroughly review the manuscript and present your variables and statistical indicators in a consistent manner.

Yes, this will all be corrected and made consistent.

455-471 There are numerous problems with sentence structure, grammar, and lack of clarity in meaning.

We will improve the grammar throughout the manuscript.

487-489 I have no idea what this means; it seems completely out of context. Please revise for clarity.

We will rewrite.

554-555 You use relation and relationship interchangeably. I suggest you select one word and use it consistently. I was once told that people have relationships but data indicate relations.

This will be adapted in the text as suggested.

587 This should be Figure 7B, not Figure 8B.

This will be adapted in the text as suggested.

---

## Author Response (AR1)

**Reply to the reviewers**

**Response to reviewer #1 comments on "Landscape and groundwater controls over boreal lake water chemistry and water balance heterogeneity in an esker complex of northeastern Ontario, Canada" by Boreux et _al_.**

In black: reviewer's comments.
In blue: our answers and/or what we added or changed in the manuscript.

We thank the reviewer for his/her thorough comments. These comments helped us to improve the manuscript. We followed the majority of the reviewer's comments and suggestions when we prepared the revised version of this manuscript.

**General comments**

This manuscript describing lake chemistry and physiography relative to elevation and positioning relative to a nearby esker is impressive in its scope (~50 lakes) and number of variables. The authors nicely show that lakes high in the landscape serve to recharge groundwater and lakes lower in the landscape receive groundwater discharge. Although impressive in the amount of data collected, the overall result of the study is not novel or surprising.

As mentioned our introduction, landscape position such as elevation is an important control on water chemistry and water balance. Thus we agree with the fact that the conclusion is "not surprising", but we argue that it is novel and important. This study: (1) uses a large data set to document the importance of landscape positon on lakes; (2) uses both conservative (isotopes) and non-conservative (chemistry) tracers to illustrate the importance of landscape positon on lake hydrology; (3) illustrates the robust nature of this data set since the data in this study were collected at three different times: and (4) develops a lake typology that provides important generalizations. Also, by carrying this study in an esker complex, it can be compared with similar studies on other esker complexes (such as in Fennoscandia), which are common in boreal regions. Also, this study is particularly relevant in the context of paleohydrology/paleoclimatic studies from this region, as allowed us to assess the response of lake position to past climate change, in reference to landscape position over the Holocene.

The authors state that understanding lake position and relative sources and losses of water are important to lake stability in response to climate change. Their overall hypothesis is that exchange with groundwater is the most important factor related to the classification that they develop, but they rely on indirect evidence (chemistry and isotopes) to determine the type of lake and degree of groundwater-surface-water exchange.

We agree that direct evidence of groundwater-surface water interactions is more powerful than indirect evidence. However, setting up direct measurements of groundwater-surface water interactions such as seepage meters in 50 lakes multiple time would have needed a significant amount of time and monetary resources that simply were not available. Water tracers such as chemistry and isotopes are indeed an indirect

evidence of such interactions and they have been proven to be reliable indirect indicators in the literature in a wide variety of settings.

Although interesting, their study does not address uncertainty and lacks confirmation of the extent to which chemistry and isotopes can serve in this function. I don't disagree with their conclusions; they seem logical and well founded. However, discussion about implications of assumptions and the possibility of misinterpretation of lake setting is missing.

We agree that our original manuscript could benefit from including the implications of the assumption and the probability of misinterpretation. This was included in the revision (assumption of steady state, differentiation of the role of groundwater and precipitation in water balance, the role of lake morphometry, *etc*.).

A slight change to the title may be more representative of the focus of this manuscript, which was to use lake water chemistry and water isotopes to infer exchange between groundwater and lakes. Rather than "Landscape and groundwater controls over boreal lake water chemistry and water balance heterogeneity. . .", it might be more accurate to title the study "Use of water isotopes and chemistry to infer the type and degree of exchange between groundwater and lakes in an esker complex of northeastern Ontario, Canada."

We agree with the reviewer that our main objective was to use tracers to infer groundwater-lake interactions rather than studying the processes *per se* and that the title could be more representative of our study. We renamed the paper as suggested by the reviewer: "Use of water isotopes and chemistry to infer the type and degree of exchange between groundwater and lakes in an esker complex of northeastern Ontario, Canada"

A substantial product is the development of a classification scheme based on the authors' results. Numerous other classifications also have been proposed and it would be useful to compare the authors' classification system with one or more of the other lake-classification systems that have been published. Several that come to mind are: Born, S.M., Smith, S.A., and Stephenson, D.A., 1979, Hydrogeology of glacial-terrain lakes, with management and planning applications: Journal of Hydrology, v. 43, p. 7-43. Bracht-Flyr, B., Istanbulluoglu, E., and Fritz, S., 2013, A hydro-climatological lake classification model and its evaluation using global data: Journal of Hydrology, v. 486, no. 0, p. 376-383. Martin, S.L., Soranno, P.A., Bremigan, M.T., and Cheruvelil, K.S., 2011, Comparing hydrogeomorphic approaches to lake classification: Environmental Management, v. 48, no. 5, p. 957. Wentz, D.A., 1981, Lake classification - Is there method to this madness?: U.S. Geological Survey Circular C0848-B, 15-24 p. Winter, T.C., 1977, Classification of the hydrologic setting of lakes in the north central United States: Water Resources Research, v. 13, no. 4, p. 753-767.

Thank you for the citations. We agree with the reviewer that comparing our lake typology with other published typologies is beneficial to the manuscript and we did so by very briefly comparing our typology with two similar ones (Turner et *al*., 2010 and Ala-aho et *al*., 2013) and by citing studies that have use the same criteria we did to establish our typology.

This manuscript appears to have been hastily prepared and the numerous errors were not only a distraction, in some cases they led to substantial confusion. The first example appears in the abstract, where the authors

mix up groundwater recharge and groundwater discharge, calling lakes situated high in the landscape groundwater discharge lakes, and lakes situated low in the landscape groundwater recharge lakes. At first, I thought the authors were simply viewing the world from a lake perspective, where groundwater flowing to a lake would provide a recharge function to a lake. But no, the authors did adhere to the common usage for the terms groundwater recharge and discharge; they simply reversed terms in the abstract. I could easily overlook such a mistake, but so many other minor mistakes also are sprinkled liberally throughout the manuscript that they represent a considerable distraction when attempting to follow the logic and conclusions of the manuscript. I provide indications of mistakes where they appear in the first few pages, but I leave it to the authors to carefully read their own manuscript and clean up the rest.

After reading the reviewer's comments and the manuscript we realized the presence of mistakes throughout the manuscript, which were confusing. We apologize for that and we made the necessary edits. We indeed view the world from a groundwater perspective and adhere to the common usage for the terms groundwater recharge and discharge. We corrected this in the abstract and throughout the manuscript based on the reviewer's comments.

**Specific comments pertaining to line numbers**

20 Higher elevation lakes are usually groundwater recharge lakes and lower elevation lakes are usually groundwater discharge lakes. Here you state the opposite, which made me think you were viewing flow of groundwater to a lake as providing a recharge function to the lake. Based on the subsequent context of the manuscript, you clearly need to correct this as follows: ". . . higher-elevation groundwater recharge lakes from lower-elevation groundwater discharge lakes." However, this provides an excellent example of how these terms are ambiguous and could easily mean the opposite from the perspective of a lake-centric scientist, who would view the addition of groundwater to a lake as serving a recharge function to the lake. Therefore, you need to define these terms in the introduction or the methods section so the reader clearly knows what you mean. You also use the terms discharge lake and drainage lake interchangeably. This is confusing. Since you first present the term discharge lake, and it is nicely contrary to recharge lake, I suggest you remove the term drainage lake and replace it everywhere with discharge lake.

We agree with the reviewer that the terms recharge and discharge are ambiguous and we defined early in the manuscript in the introduction what we mean by those terms. In this paper, we adhere to the common usage for the terms groundwater recharge and discharge and view the world from a groundwater perspective.

72-73 You state here that you can determine evaporation and water balance provided the Craig-Gordon model is used. You state that the Craig Gordon model has to be used to do this, but then you surprisingly cite a paper published in 2015 the title of which implies it is all about software called Hydrocalculator. If you are using the CraigGordon model, you should cite Craig and Gordon, 1965. You also need to revise the sentence to indicate that this particular model is either somehow better or more convenient for accomplishing these purposes than the now standard method that is commonly used.

The explanation of the method seemed more intuitive in Skrzypek et *al*. (2015) than in the original paper, thus the reason to cite the latter reference. However, we do agree with the reviewer that the original paper

should be cited. The change of citation was made. The other reviewer pointed flaws in the section on the methodology to determined water balance and this section was rewritten.

78 You state there are few studies that have used both chemistry and water isotopes, and then you cite 4 such studies. In addition to those four studies, I can think of several more without even perusing the literature. This is not a "very limited number" especially relative to the number of studies in boreal lakes, which likely is smaller. Gurrieri, J.T., and Furniss, G., 2004, Estimation of groundwater exchange in alpine lakes using non-steady mass-balance methods: Journal of Hydrology, v. 297, p. 187– 208. Krabbenhoft, D.P., Bowser, C.J., Kendall, C., and Gat, J.R., 1994, Use of oxygen- 18 and deuterium to assess the hydrology of groundwater-lake systems, in Baker, L.A., ed., Environmental Chemistry of Lakes and Reservoirs: American Chemical Society, p. 67-90. LaBaugh, J.W., Winter, T.C., Rosenberry, D.O., Schuster, P.F., Reddy, M.M., and Aiken, G.R., 1997, Hydrological and chemical estimates of the water balance of a closed-basin lake in north central Minnesota: Water Resources Research, v. 33, no. 12, p. 2799-2812. Katz, B.G., Coplen, T.B., Bullen, T.D., and Davis, J.H., 1997, Use of chemical and isotopic tracers to characterize the interactions between ground water and surface water in mantled karst: Ground Water, v. 35, no. 6, p. 1014-1028. Turner, J.V., and Townley, L.R., 2006, Determination of groundwater flow-through regimes of shallow lakes and wetlands from numerical analysis of stable isotope and chloride tracer distribution patterns: Journal of Hydrology, v. 320, no. 3-4, p. 451-483.

We do agree with the reviewer that there was not only a limited number of studies that used both water chemistry and isotopes to investigate groundwater-lake water connections. Our phrase was awkwardly worded in the sense that we meant few studies have used such a combined approach on the scale of the landscape and most studies used only one approach (isotopic or chemical). We clarified this in the revision, and provided appropriate citations.

148-149 Evidently you are only considering watershed slopes for the terrain that is within 100 m of the lake shoreline. For most lakes, this will be a small percentage of the lake watershed. You need to indicate why you are applying this 100-m filter to your analysis.

The coarse spatial resolution of the DEM and the close proximity of the lakes in the study area made it impossible for ArcGIS to compute the watershed delineation of each lake. Because all lakes are kettles (which are characterized by steep slopes on their shore over a small distance) and nearby, arbitrary buffer zone widths of 50 m and 100 m were tested and displayed similar correlations with lake isotopic composition: Max slope 100 m (r=-0.33, p=0.02), Max slope 50 m (r=-0.31, p=0.03). We clarified this in the revision.

234-235 Most studies that use water isotopes to look at water budgets go to considerable effort to measure or estimate $\delta E$. Here you simply assume that $\delta E = \delta I$ minus $\delta L$. This method assumes that you know all of the other input and loss volumes and their integrated isotopic signatures. You should address this issue and either indicate that your assumed method for determining $\delta E$ is well supported by other studies or you should state that this method includes uncertainty associated with all of those other water-budget terms.

We agree with the reviewer that the provided equation was not the right one, and we thank the reviewer for pointing out our mistake. We originally aimed at summarizing this section as much as possible as this

method has been used in several papers and we accidently listed a wrong formula. Thus, more text and details was provided and in the rewrite of this section (see response to reviewer #2).

244 Because this manuscript is about groundwater and lakes, you should indicate what the groundwater temperatures are in this area. Temperatures of springs are not an accurate indication of groundwater temperature because spring temperature at the discharge point is so influenced by the velocity of the discharging spring and the resulting warming of the discharging water as it approaches the surface. Secondly, is there a difference in lake temperature between the higher and lower lakes? If you have substantially more groundwater discharge in the lower lakes you may see a resulting temperature difference in lake temperature. This is unlikely, however, because so many other factors also affect temperature, but with 50 lakes there may be something.

The Provincial Groundwater Monitoring Network of the province of Ontario of Canada (where the study area is located) monitor 3 groundwater wells in the area (water level and water chemistry) but does not report any water temperature measurements. Thus, we do not have direct measurements of groundwater temperature.

We agree with the reviewer that "the temperatures of springs are not an accurate indication of groundwater temperature because spring temperature at the discharge point is so influenced by the velocity of the discharging spring and the resulting warming of the discharging water as it approaches the surface". However, by referring to the temperature of springs, we do not interpret it to be an indicator of the temperature of groundwater but we looked at the temperature of spring to infer that those were indeed springs (due to their lower temperature). Some spring samples were taken from wells build by locals, which fore sure introduce a bias in the data. This was clarified in the revision.

Interestingly the temperature was recorded to test whether lakes that have a greater proportion of their water balance coming from groundwater had colder temperatures or not (even though we did not really expect to find such a relationship). This was not the case as supported by statistical analyses in the appendix (Tab A2). Lake size was correlated with water temperature as bigger bodies warm more slowly and cool down slower in comparison to smaller ones.

256-257 It would be very useful if you could include here a figure that maps the lakes with indications of the isotopic value (select 18-O or deuterium) and perhaps also the specific conductance. This would allow the reader to see the spatial distribution of the various isotopic values that are indicated in this paragraph. If you do this, please also show the lakes that have a stream outlet and indicate where that stream goes. As I read farther I see that Figure 5 does a reasonable job of showing this but it lacks mapping of the isotopic lake values. Ultimately, what would be most useful is a map showing the resulting distribution of recharge, seepage, and discharge lakes, perhaps overlain on a map that uses shading to show elevation.

Because the E/I calculation is based on $\delta^{18}O$, we didn't include a map with E/I values as we thought this would be redundant. However, we acknowledge the fact that this might be of interest to some readers. The addition of a map showing the distribution of lake types in the landscape was an interesting suggestion. Because we are concerned with not overcrowding the manuscript with figures, we added those this additional information on a more complex figure as supplemental information (pH was replaced with $\delta^{18}O$

and put in the appendix along with a map of lake type (Fig. A1)). The representation of outlets was added to Fig. 1 and not to Fig. 5 as the latter figure is too small to see those features. The direction of flow was not added as it would overcrowd the map.

266 The slope of your relation is negative but the r-squared value should be positive.

This is a major typographical error. Coefficients of determination cannot have negative values. Following the reviewer's comments, we made the necessary changes throughout the manuscript and present the correlation coefficient.

266 It is not the geographic position that is related. It is the elevation that is related. You may infer from the elevation the position of the lake relative to the crest or the distance from the crest. If that is the case, you should state that here.

Correct. Elevation that is related to water isotopic composition and chemistry. Geographic position was used as a synonym but we agree that this terminology was confusing, and corrected it.

268-269 You state the relation between isotopes and elevation. You should, therefore, also provide a sentence and state the relation between EC and elevation.

We agree with the reviewer and will state the relation between EC and elevation as well. A sentence was added in the next section on solutes.

277 Why are you relating E to I? You stated in the Methods that you can do this but you didn't say why. You should explain what this accomplishes. Also, since you're determining delE from delI minus delO, your equation 5 is really delI-delL/delI-2delL.

We can relate E to I assuming that the lakes are in isotopic steady state. This assumption seems well justified as the lakes have had a lot of time in the past to reach their isotopic steady-state which is reflective of the local climate and their mean hydrological status, and can be defined by its water balance (which corresponds to the ratio of the total inflow to the evaporation rate, Isokangas et *al*., 2015). This was clarified in the method section. As also pointed out by the other reviewer, the formula for δE has been corrected, and the methodology for the calculation of water balance has been rewritten (see response to reviewer #2).

Reference: Isokangas, E., Rozanski, K., Rossi, P.M., Ronkanen, A.K. and Kløve, B.: Quantifying groundwater dependence of a sub-polar lake cluster in Finland using an isotope mass balance approach, Hydrol. Earth Syst. Sci., 19, 1247-1262, https://doi.org/10.5194/hess-19-1247-2015, 2015.

277-278 A small E/I could also simply indicate that the lake has an outlet (I~Q) and, therefore, a short residence time, which has nothing to do with any indication of whether groundwater or springs undergo evaporation or not. Interpretations related to E/I also incorporate error associated with calculating delE rather than measuring it.

We agree with the reviewer's comment that residence times are key here and we clarified this.

Figure 4 I don't understand how you can obtain E/I values for springs and streams when your two variables for making this calculation are delI and delL. What lake do you use to determine delL for a spring or a stream?

In that case we treated the spring water samples as if they were a spring or a lake and used the same methodology. We will clarify in the methods that we used δG for groundwater and δS for streams.

297 You can't have a negative R-squared value. These values must instead be correlation coefficients, which are usually indicated as lower-case r. Also, where do you present these values? DOC is not included in the correlation matrix in Table S1b in the Appendix. You do show TN but the value for r when TN is related to H (elevation) is 0.05, not 0.22 as reported here. And since we're talking about the table, why is the table labeled S1b? Is there a table S1A that is missing?

Agree. We made the changes and used correlations. DOC was indeed included in Tab. S1b but under the label NPOC, we changed this so that the names of the variables is consistent throughout the manuscript. The errors have been corrected. All correlation values (r) are displayed in Tab A1a and b. There is no Tab S1. The journal asked us to rename Tab S1 to Tab a1 and not all notations were changed.

315 What is d in Table 1? Is it deuterium excess? If so, you have not defined or describe this anywhere.

d indeed corresponds to deuterium excess. We have defined this as a note below Table 1.

320 Be sure to clearly show any surface-water connections between lakes. The presence of an outlet, or of surface-water flow from one lake to another, is an important part of your analysis and up to now the reader has not been able to see these connections. You could show this either here or in Figure 1.

We added the outlets in Fig 1 as suggested.

325 You use an elevation of 242 m as the dividing line between recharge and discharge lakes. However, in Figure 5, and in the previous paragraph, you show a middle elevation, or an elevation break point of just over 280 m. And there is no elevation in Figure 5 lower than 260 m in Figure 5. If all of the lakes shown in Figure 5 are higher than 260 m, and your previously determined break-point elevation was 282.4 m, how can you now choose an elevation of 242 m to separate your lakes?

The elevation of the breakpoint line is indeed 282 m, not 242 m (as accurately mentioned by the reviewer, there is no elevation lower than 260 in our study area). This is an unfortunate typo. This was corrected throughout the manuscript.

327 What additional lake characteristics were used to allow you to divide the lakes into three categories?

Other lake characteristics include the lake's pH, the presence or absence of an outlet. This was added in the text in brackets.

336 How does this value of 242 m relate to your break-point elevation of 282.4 m? If indeed these are two separate distinguishing elevations, you need to describe how one is important relative to the other.

The elevation is 282 m, not 242 m. A previous typo earlier in the manuscript made it confusing. This was corrected.

370-371 Here you finally indicate that your streams are all upgradient of a lake and that you evidently did not sample any stream reaches that were situated between lakes. This is an important distinction that needs to be made clear much earlier in the manuscript.

We did sample upgradient streams as well as outlets that flow into other lakes (mainly into Frederick House Lake, which feeds the Frederick House River that drains the entire area). We made this clearer early in the manuscript as suggested. All the lakes mentioned in the manuscript were labelled in the map as previously suggested and streams were added to the map of the area as well.

416-417 You should not assume that a recharge lake is perched. I greatly doubt that lakes situated in permeable material such as you have in the esker are perched. Unless you have separate information indicating otherwise, you should remove the word perched from the manuscript.

We agree with the reviewer that the assumption that a recharge lake is perched is erroneous as esker are indeed made of permeable material. The term "perched" was wrongly used as a synonym of recharge, which is not. This was corrected.

433 Here you do report values as R rather than R-squared.

As commented above, changes were made and only r values were reported.

444 The presence or absence of an outlet is a very important variable in this analysis and this is the first time you have brought up this point. I would think this would be a presence-absence type of variable that would have been part of your statistical analysis. In any case, I was looking for outlets in the figures and it was very difficult to determine whether you showed all of the outlets or not. For some lakes, based on their positioning I would expect an outlet and yet one is not shown in the figure. If you do show all of the stream segments in your study area, there is a surprisingly small number. If not, you should revise the figure to do so. The presence of both inlets and outlets is important to this type of analysis.

The presence or absence of an outlet was investigated with a logistic regression (lines 462-467). Fig. 1 was revised to include all know outlets and streams in the area. However, the local topographic maps don't display all the streams observed in the field and in satellite imagery due to the presence of a dense forest. Thus the updated map may not be exhaustive of all streams in the area but all known streams and streams that appear distinctively in satellite images were included.

455 You should label and show in a map where these two lakes are.

Those lakes were labelled on Fig. 1 as suggested.

455-460 A lake without an outlet could also be due to the lake being able to easily lose water to groundwater.

We totally agree with the reviewer and this argument was added in the manuscript.

473 What is a mounded recharge lake? This needs to be defined. If this is a lake that loses water everywhere to groundwater, you can't call that mounded unless you have hydraulic gradient data to confirm that. It may instead be that the lake is situated in a small local watershed or that it has more organic sediments than others that creates a low pH.

Mounded lakes are lakes positioned above the water table that receive most of their water input from direct precipitation as defined by the ALSC Lake Classification System (Newton and Driscoll, 1990). We agree with the reviewer that labelling those lakes as mounded without knowing the hydraulic gradient might be incorrect. Given this, we removed this terminology. We agree with the reviewer that those lakes are just situated in a smaller local watershed.

Reference: Newton, R.M. and Driscoll, C.T.: Classification of ALSC lakes, in: Adirondack Lakes Survey: An Interpretive Analysis of Fish Communities and Water Chemistry, 1984-87, Adirondack Lakes Survey Corporation, Ray Brook, NY., 2-70 to 2-91, 1990.

Figure 8 What distinguishes a recharge seepage lake from a discharge seepage lake? Is it only the pH? You talk considerably about recharge seepage lakes and the reasons for their classification, but you say nothing about discharge seepage lakes or how they are different from recharge seepage lakes.

Those two lakes are distinguished by their position: in the groundwater discharge zone or the groundwater recharge zone inferred from the breakpoint analysis. This was adapted in the manuscript as follows: "In the groundwater recharge zone, seepage lakes differ from recharge lakes by their pH, which is more alkaline while in the groundwater discharge zone, seepage lakes differ from discharge lakes by the absence of an outlet, meaning they receive less input than discharge lakes".

520 Here you finally define a drainage lake. Given that this indeed is the same as a discharge lake, I suggest you use one term or the other and do not use both interchangeably.

Agreed. We replaced "drainage" by "discharge" throughout the manuscript.

545-551 Greater lake evaporation also is related to more isotopically enriched water in lakes. If there is lake-to-lake variation in evaporation, that variation affects your interpretation of the extent to which the lake exchanges water with groundwater. Your classification system does not provide any indication of the extent to which other factors affect your interpretation of exchange with groundwater. It would be helpful to point this out in this paragraph.

We agree, and included this in the discussion. Although lake morphometry can influence isotopic values, elevation accounts for much more variance in isotopic composition (r=0.53, p<0.001) and trumps the influence of variations in evaporation between lakes.

567-568 You cannot make this statement unless you either have evaporation data to back this up or you can cite some papers that have reached this conclusion. I strongly doubt that evaporation from lakes that receive groundwater discharge is measurably different from lakes that recharge groundwater. What is different is the degree of isotopic enrichment in the lake water. The evaporative enrichment is masked by isotopically light groundwater.

We agree with the reviewer's comment and we realized that that sentence was poorly worded. We replaced "changes in evaporation" by "changes in water balance".

591-592 This logic is not complete. If drought reduces groundwater flow to a lake because of reduced water-table elevation, why then would there not be increased flow of lake water to groundwater on the downgradient side of the lake, in which case there would be no net change in exchange with solutes?

We agree. Recharge lakes would feed the groundwater system by lake outseepage, which would produce no net change in solutes. This was included in this section.

598-599 Residence time would be much more greatly controlled by local hydraulic conductivity. If most of your recharge lakes are situated on the esker, then I would think residence time would be relatively short compared to a lake lower in your system that is situated in lower-permeability sediments.

We agree with the reviewer that residence times can be related to the permeability of the material it is embedded in. However, residence times are a reflection of both watershed size and lake volume. In our study area, lakes at lower elevation tend to be much shallower and have larger watersheds, resulting in increased flushing. We discussed this in the revision.

607-614 This paragraph is poorly presented and is confusing. What do you mean by multiple site selection? What sites are you talking about and for what are they being selected? How is this related to paleohydrological reconstruction?

Site selection is meant in reference to lake selection for a paleohydrology/paleoclimatic studies from the lake sediments. Hence, knowledge of present-day hydrology and connection to the groundwater is important. For example, choosing a groundwater discharge lake for a paleo study may show stability, whereas a higher elevation groundwater recharge lake may be more sensitive. Our study provides the modern-day context for lake selection.

**Technical corrections**

22-23 You need to place a comma after contrast and remove the comma after lakes so the sentence reads "In contrast, groundwater discharge lakes were isotopically depleted . . ."

This was adapted in the text as suggested.

31-32 You should remove the parentheses here; otherwise, the sentence is not complete. Also, how can you determine seasonal differences in chemistry when you only have data between early June and August? That period is during summer only.

The parenthesis were removed as suggested. As the reviewer mentions below, "seasonally" implies comparisons over multiple seasons. Technically, early June is still in the spring and August is in the summer. Plus, in those settings, summers are usually shorter than at lower latitudes so the notion of season still makes sense here. But at the same time, we agree that the usage of the term season is debateable. Therefore we used the term summertime instead.

34-35 Recharge lakes also are affected by changes of greater duration and persistence. Please add the word "only" to write ". . . affected only by hydroclimatological changes of greater duration and persistence."

This was adapted in the text as suggested.

47-48 The threat of climate change will not influence lakes. It is the climate change that will influence lakes. Therefore, remove "The threat of" and start the sentence with "Future climate change will. . ."

This was adapted in the text as suggested.

49 A word is missing. Insert "understand" to write ". . . to better understand relationships of lake hydrology . . ."

This was adapted in the text as suggested.

55 Another word is missing. Insert "to" to write ". . . the opportunity to better understand . . ."

This was adapted in the text as suggested.

64 Now you have an extra word. Delete the word "changing".

This was adapted in the text as suggested.

70 Change tracers to tracer because it needs to be singular here.

This was adapted in the text as suggested.

72 What is a "time and cost effective means"? Perhaps you mean "an efficient and cost-effective means"

Yes, "time and cost effective means" was replaced by: "an efficient and cost-effective means".

98 You state this is near Timmins but you do not show Timmins anywhere on the map. Please add the location of Timmins to the map. You also need to add labels to show the locations of Frederick House and Night Hawk Lakes.

The city of Timmins was added to the map and the location of cited lakes was labelled also.

110 What is a popular? I suspect you actually mean poplar.

The correction to poplar was made. The error was likely due to an automatic correction.

124 I suggest you add the word littoral to write ". . . forming lateral littoral sand units that drape . . .". This addition will provide a better tie with the "littoral sand" indicated in Figure 1.

This was adapted in the text as suggested.

125 You have an extra and unnecessary word here. Delete the word "in" to write "The numerous kettle lakes on the esker formed . . ."

This was corrected in the text as suggested.

128 Why does only the esker crest have high-K sediments? I would think the entire esker would be much higher in K than your surrounding sediments. If it is just the upper portion of the esker that serves as an aquifer, you need to explain why this esker is geologically different from other eskers.

The reviewer is correct that the whole esker has high K-sediments. The word "crest" has no place here. This was corrected in the text.

145 Depths of what? You should write "maximum lake depths" so the reader will know you are not talking about depths of other geographic features.

We wrote "maximum lake depths" to make it clear to the reader.

147-148 What does spatial resolution have to do with lake-surface elevation? This needs a better explanation.

In ArcGIS, the elevation of a polygon (here a lake) can be obtained from a Digital Elevation Model using the spatial statistics extension. When requesting the calculation, the software asks whether you want to use the mean, max or min value and we selected mean. It has indeed nothing to do with spatial resolution. This was made clearer in the manuscript.

153 You were only there for one week each time? That certainly does not constitute a season. You should replace the word season with something else; perhaps field campaigns or field trips.

The word season was replaced by campaign as suggested.

154 Earlier you state that the number of lakes is 50, but here you say that this number is approximate. Surely you know how many lakes you sampled and you should provide that number. Otherwise, this text gives the

impression that you can't be bothered with counting up the lakes you sampled or that you were not very meticulous. I suspect this is not the case, but this and the numerous previously identified simple mistakes certainly give a reader pause.

The word approximately was erased. There was indeed 50 lakes sampled.

163 The word is triple-rinse, not tripled- rinsed. This mistake is repeated on line 164.

This was adapted in the text as suggested in lines 163, 164 and 170.

164 You say bottles were "and again with sample water," which I presume to mean that you triple-rinsed the bottle with sample water after the bottle was triple rinsed with distilled water. But then in the next sentence you write that bottles were triple-rinsed with lake water. Is this simply a mistake or redundancy in your writing or did you rinse the bottles in lake water twice?

Bottles were triple rinsed with distilled water off-site first, then triple rinsed with sample water on-site. We reworded this phrase to make it clearer.

170 Apparatus should be replaced with apparati because the following word, "were", is plural.

This will be adapted in the text as suggested.

171 Filtrate is singular but that word is followed by were, which is plural. Either change filtrate to filtrates or change were to was.

We made the correction and change "filtrate" to "filtrates".

172 One does not store into. The samples were either stored or they were placed into.

We changed "stored" by "placed".

198 Change detections to detection.

This was adapted in the text as suggested.

201 Detections limits should be change to detection limit.

This will be adapted in the text as suggested.

202 You already defined these three terms in lines 177-178 so you do not need to do so a second time here.

This was adapted in the text as suggested.

203 What is NDIR? You need to indicate what this abbreviation indicates. Also, since it is only used once in the manuscript, it would be better to simply indicate what this is.

We agree with the reviewer's point. We changed "NDIR" to "nondispersive infrared sensor".

204 Change detections to detection.

This was adapted in the text as suggested.

207 You need to indicate what TIC is. I assume it is total inorganic carbon. If so, how was this determined? You do not say. Is TIC only the dissolved portion of the sample?

TIC is indeed total inorganic carbon. This variable was actually not used in our analysis and is not mentioned anywhere else in the manuscript. It must have been left in the method section by accident. Therefore, this sentence was deleted.

269-270 You could consider deleting this sentence. It is fairly standard practice to just use 18-O (most common) or 2-H and you've already shown the strong correlation in Figure 2. You might also modify slightly to write ". . . are enriched in 18-O, and due to their strong correlation, 2-H."

We deleted this sentence as suggested. Even though we agree it is a standard practice to use $^{18}$O rather than $^{2}$H, we thought it might have been worthwhile to mention it in case the reader would wonder why we seem to omit referring to $^{2}$H.

288 Delete "it" to write "As is the case with stable isotope values. . ."

This was adapted in the text as suggested.

332-333 This sentence does not make sense. I can't tell if there are missing words, or if infer is supposed to be inter. In any case, the text in quotations needs to be corrected: "In order to better understand the contribution of infer water chemistry variables,"

We agree with the reviewer that the sentence in quotation marks doesn't make sense. We reworded "In order to better understand the contribution of infer water chemistry variables," to "In order to better understand how water tracers vary in those two zones".

342 The table caption states that some lakes are labeled in red. The figure is in black and white (although Figure 5 is in color) so nothing is shown in red here.

We changed this sentence as the figure is in black and white. The figure used to be in color in a previous draft and was changed to black and white for the journal. We made the appropriate edits.

349 Panel A shows inter-annual change and panel B shows variability over a single summer; the citation should be to Fig. 7a,b). However, "seasonally" implies comparisons over multiple seasons. Your analysis

is not seasonal. You only compared values over parts of one season, that being summer. Therefore, you should replace any inference to a seasonal analysis to something like "summertime" or "over-summer".

We used the terms at the beginning and towards the end of the growing season to identify the June and the August campaigns. The text was changed to refer to Fig 7 a,b (although this figure was moved to the discussion, following the suggestion of the other reviewer).

355 Here you again refer to panel B but the solute-related panels are 7c and 7d.

The text was changed to refer to Fig 7 c,d (although this figure was moved to the discussion, following the other reviewer's comments).

393 Please remove the hyphen between lake and water.

This was adapted in the text as suggested.

428 Replace delivery with deliver. In fact, this entire paragraph could use considerable help.

This sentence was changed. Edits were made to improve the quality of the entire paragraph as suggested.

448 Here you use lower-case r. Thus far you have used R, R-squared, r, and r-squared. Do they all mean the same thing in this manuscript? They certainly mean different things in statistics. You need to thoroughly review the manuscript and present your variables and statistical indicators in a consistent manner.

Yes, this was all corrected and made consistent.

455-471 There are numerous problems with sentence structure, grammar, and lack of clarity in meaning.

Efforts were made to improve the paragraph.

487-489 I have no idea what this means; it seems completely out of context. Please revise for clarity.

This paragraph was deleted for a better clarity and flow of the discussion.

554-555 You use relation and relationship interchangeably. I suggest you select one word and use it consistently. I was once told that people have relationships but data indicate relations.

This was adapted in the text throughout the manuscript as suggested.

587 This should be Figure 7B, not Figure 8B.

This was adapted in the text as suggested.

**Response to reviewer #2 comments on "Landscape and groundwater controls over boreal lake water chemistry and water balance heterogeneity in an esker complex of northeastern Ontario, Canada" by Boreux et _al._**

In black: reviewer's comments.
In blue: our answers and/or what we added or changed in the manuscript.

We thank the reviewer for the comments on our manuscript which aided in improving a new version of the manuscript. We found the comments and suggestions useful and the majority of suggestions were incorporated in the revised version of the manuscript.

Boreux et al. utilize lake water isotope and chemistry data to identify lake types in an esker complex in Ontario, and specifically characterize the role of groundwater and landscape position on lake conditions. I enjoyed reading the paper. This is an excellent dataset and, by and large, the interpretations appear to be sound and supported by the data. So, from an overall perspective, this paper will make for a useful contribution on lake hydrology. In particular, the links between water isotope composition and water chemistry serve as useful example of the strength in such an approach to characterize lake conditions at numerous locations across landscapes.

Thank you.

That being said, there are both analytical and structural issues with this manuscript that, in my view, should be addressed. I elaborate on these below.

1. Analytical. Specifically here, I refer to calculation of the isotope-inferred E/I ratios. If the authors intend on retaining this piece of the manuscript, much more detail needs to be provided as to how these calculations were performed. For instance, at the top of page 9 where the E/I ratio is provided, there is an incorrect definition of dE. Several of the papers listed in the reference utilize the Gonfiantini (1986) equation, but I do not know if the authors used this equation. Furthermore, there are no details provided regarding how they computed dA, or what they used for temperature or relative humidity. Fundamentally, insufficient text and details are provided, such that the calculations cannot be reproduced by a reader. So, much more needs to be provided here. But I wonder if this is even necessary? The E/I results are barely mentioned in the Discussion. It seems that, for the apparent purposes of this paper, the authors could simply get away with comments referring to differences in the degree of evaporative isotopic enrichment.

We agree after reading your comment that the calculations cannot be reproduced by a reader. We originally aimed at summarizing this as much as possible as this method has been used in several papers, but the external perspective of the reviewer made us realize this mistake. Thus, more text and details were provided and the section was rewritten as follows.

$\delta_E$ was estimated using the Craig-Gordon model (Craig and Gordon, 1965) formulated by Gonfiantini (1986) as follows:

$$\delta_E = \frac{(\delta_L - \varepsilon^*)/\alpha^* - h\delta_A - \varepsilon_k}{1 - h + \varepsilon_k}$$

where $\delta_L$ is the isotopic composition of lake water, $\varepsilon^*$ is the equilibrium isotopic separation term, $\alpha^*$ is the liquid–vapour equilibrium fractionation factor, $h$ is the relative humidity, $\delta_A$ is the isotopic composition of the local atmospheric moisture, and $\varepsilon_k$ is the kinetic separation term between the liquid and vapour phases. The $\varepsilon^*$ and $\alpha^*$ parameters which are temperature dependent can be calculated using empirical equations for $\delta^{18}O$ as follows (Horita and Wesolowski, 1994):

$$\varepsilon^* = -7.685 + 6.7123\left(\frac{10^3}{T}\right) - 1.6664\left(\frac{10^6}{T^2}\right) + 0.35041\left(\frac{10^9}{T^3}\right)$$

$$\alpha^* = exp\left(-\frac{7.685}{10^3} + \frac{6.7123}{T} - \frac{1666.4}{T^2} + \frac{350410}{T^3}\right)$$

where T is the air temperature in Kelvins. $\varepsilon_k$ (Eq. 9) is expressed for $\delta^{18}O$ by (Gonfiantini, 1986):

$$\varepsilon_k = (0.0142\,(1 - h))1000$$

The equation for $\delta_E$ was modified according to Gibson and Edwards (2002) to directly utilize isotopic data in per mil rather than as a decimal fraction and expressed as follows:

$$\delta_E = \frac{\alpha^*\delta_L - h\delta_A - \varepsilon}{1 - h + 10^{-3}\varepsilon_k}$$

where $\varepsilon$ is the total isotopic separation factor that includes both $\varepsilon^*$ and $\varepsilon_k$ expressed as:

$$\varepsilon = \varepsilon^* + \varepsilon_k$$

$\delta_A$ was originally estimated with the original model that assumes isotopic equilibrium between atmospheric moisture and precipitation as follows (Gibson, 2002):

$$\delta_A = \frac{\delta_P - \varepsilon^*}{1 + 10^{-3}\varepsilon^*}$$

where $\delta p$ was computed as the average isotopic composition of annual precipitation from February 1997 to November 2010 (data collected by CNIP).

References:
Craig, H. and Gordon, L.I.: Deuterium and Oxygen-18 Variations in the Ocean and the Marine Atmosphere, in: Stable Isotopes in Oceanographic Studies and Paleotemperatures, edited by: Tongiorgi, E., Laboratorio di geologia nucleare, Pisa, Italy, 9–130, 1965.
Gibson, J.J.: A new conceptual model for predicting isotope enrichment of lakes in seasonal climates, PAGES News, 10, 10-11, 2002.

Gibson, J.J. and Edwards, T.W.D.: Regional surface water balance and evaporation–transpiration partitioning from a stable isotope survey of lakes in northern Canada, Global Biogeochem. Cycles, 16, 1-14, doi:10.1029/2001GB001839, 2002.

Gonfiantini, R: Environmental isotopes in lake studies, In: Handbook of Environmental Isotope Geochemistry, Vol. 2, The Terrestrial Environment, edited by: Fritz, B.P. and Fontes, J.C., Elsevier, Amsterdam, the Netherlands, 113–168, 1986.

Horita, J. and Wesolowski, D.: Liquid-vapour fractionation of oxygen and hydrogen isotopes of water from the freezing to the critical temperature, Geochim. Cosmochim. Acta, 58, 3425–3437, doi:10.1016/0016-7037(94)90096-5, 1994.

Perhaps it is just my weak understanding of statistics, but I would like to see a bit more background on the breakpoint analysis. This seems to be a very important part of the paper, as it apparently identifies 282 m (or is it 242 m? – both are listed, which added to my confusion) as the landscape position in which the role of groundwater changes. My limited understanding of breakpoint analysis is that it is used to detect when there is a change in a trend (often applied to time-series measurements), but it seems like this analysis is being applied differently here.

As the reviewer accurately mentioned, breakpoint analysis is used to detect when there is a change in a trend and it is often applied to time-series measurements. In our case, we have a gradient of lakes ranked according to elevation (thus a continuous series of data just like a time series) and we observed that lakes behaved isotopically and chemically differently in two groups according to elevation. A breakpoint analysis was done and indeed revealed that the breakpoint was significant for most variables at around the same elevation. We understand the use of breakpoint analysis is a little unusual in this context as clustering is usually the prefer method. Clustering would provide a grouping of the lakes but this method would not provide with a value of elevation that divides the two zones like the breakpoint analysis does. This is particularly relevant because: (1) the breakpoint line allows us to map what we believe to be the groundwater recharge and discharge zones; and (2) the elevation of the breakpoint line actually slightly varies according to the sampling campaign as does the water table level and allow us to track short term changes while the grouping of the lakes remains the same. The elevation of the breakpoint line is indeed 282 m, not 242 m. This is an unfortunate typo. Thank you for pointing this out.

2. Structural. In its present form, the manuscript imposes the lake classification scheme on the results, but it is only part-way through the Discussion where the rationale for the classification scheme is presented. This creates a lot of awkwardness with the paper. The reader is quite literally forced to take the author's word for it that there is some basis for the classification scheme while they read the results (but they have no idea what that is). Since my expertise lies more with the isotope component of the paper, I kept wondering how the authors are going to distinguish the role of groundwater versus precipitation, since they often have very similar isotope compositions. It was only much later, in the Discussion, where I learned that the role of groundwater, in fact, is largely, maybe exclusively (?) based on the water chemistry. There needs to be some re-structuring of the manuscript, so that the reader can examine the results, unencumbered by the imposing of the classification scheme, and then use those results to develop the classification rationale. I think you are likely to end up with the same ultimate intepretations and conclusions, but the path to getting there needs refinement and re-organization so that it will be easier for the reader to digest.

We agree with the reviewer's diagnostic that: (1) the manuscript needed some restructuring so that the results and their interpretation are better distinguished; and that (2) some challenges such as the distinction between the roles of groundwater versus precipitation should be acknowledged and explained in the manuscript.

Other comments are listed below:

Thank you for the details comments, which will improve the quality and the clarity of the manuscript.

Line 17: Insert a sentence explaining why the study was conducted.

We developed more on the relevance on the study (as also suggested by the reviewer's comments on line 600-602 below). The main reason for developing this study is to assess the lakes' sensitivity to changes in water balance which is necessary for an informed discussion of paleoenvironmental/climatic studies on a subset of these lakes, as well as to develop a more detailed understanding on the local hydrology (the region has a lot of mining activities and cottages).

Line 21-22: Interesting that evaporation did not lead to concentration of ions. Seems like an unexpected relation.

The reviewer is correct in stating that evaporation can lead to the evapoconcentration of ions but this doesn't really apply in this context because of the mesic climate in this region. In our case, water residence is low and the amount of solutes is a reflection of the rain water. Evaporation leads to changes in ion concentration for recharge lakes as showed in Fig. 7. Interestingly when we look at the percentage of change vs. the raw amount of change in electric conductance, the percentage is higher for recharge lakes. However, the amount of solutes in those lakes is so low that this enrichment in ions is insignificant. If the study area was more arid the results likely would be different.

Line 28: 'characteristics'

This was corrected in the text.

Line 33: 'discharge lakes showed'

This was adapted in the text as suggested.

Line 34: 'would only be likely affected'

This was adapted in the text as suggested.

Line 64: delete 'changing'

We adapted the sentence to "interactions can vary temporally according to changes in seasonality and longer term changes in hydroclimatic conditions" as suggested by the other reviewer.

Line 84: replace 'to test' with 'examine'

This was adapted in the text as suggested.

At this point, I'm wondering how the authors might distinguish influence of groundwater from precipitation since they often have similar isotopic signatures. Authors might want to acknowledge this challenge here, and may use this to also rationalize the combined use of water isotopes and chemistry.

The reviewer raises an excellent point here. Indeed, as shown in Fig.2, the isotopic signature of groundwater and mean annual precipitation are very similar (which is not surprising). As accurately suggested, we will acknowledge this challenge and further support the rational of using both water isotopes and chemistry in our study. However, given that: (1) our study was carried out on a small spatial scale (our study area consist basically of a rectangular zone of 12 km by 6 km); (2) the close proximity of the lakes; (3) the terrain homogeneity (with boreal forest as the dominant land cover); and (4) the limited topography, it is unlikely that there are significant differences in terms of precipitation patterns within our study area. Due to the particular setting of our study area, in this case, we believe this is a fair assumption that groundwater connectivity is the main control on lake water balance for our 50 lakes. This challenge was taken into account in the beginning of the discussion.

Line 91-92: re 'water management and conservation goals' . . . some mention of this applied aspect should be stated in the abstract, and furthermore, the abstract might relay more directly how the findings from this study contribute to these aspects

As suggested by the reviewer, we mentioned the usefulness of this study in the abstract and how our findings and the developed typology can contribute to the conservation and the better management of the local environment.

Figure 1 caption: 'northeast'; label 'a' and 'b' on figure and label lakes and locations mentioned at the top of page 5

We labelled part a and part b of Fig. 1 as suggested by the reviewer and labelled the mentioned locations on the map.

Line 107: delete 'the'

This was adapted in the text as suggested.

Line 129: 'texture. The esker'

This was adapted in the text as suggested.

Line 133: 'aquifers at its edges where'

This was corrected in the text.

Line 153-154: Can you explain the three-time sampling rationale? This puzzled me a bit because the data are weighted to the early ice-free season, as mostly mean values are used, and there is very little mentioned about seasonal patterns in the data that are obtained.

There were three field campaigns to ensure the reproducibility of the data. Lakes were sampled mostly in the early ice-free season due to logistical reasons. The description of the observed changes between sampling campaigns are commented in the end of the manuscript in relation to the lake typology as the observed changed were specific to lake type.

Line 209: As stated above in my comments, I wonder if you even need to perform these calculations. If you would like to retain, more details are required as per comments above.

While we agree with the reviewer's previous statement that this study could be done with just the water stable isotope values, calculating the water balance gives us a better and more intuitive sense of the extent to which lakes are sensitive to evaporation. Thus, we took the above comments into consideration and added details to the calculation of E/I ratios.

Line 233: Another point of confusion for me was that dI was assumed to be the average of precipitation, and was assumed to be the same for all lakes. Yet knowing that the goal here is to detect influence from groundwater, I wondered then if there might be some variability in dI among your lakes. And, of course, I remained curious to know how groundwater influence might be distinguished from precipitation since they often have similar isotope signatures. I'm not sure any of this was ever explicitly discussed in the manuscript.

Even though $\delta_P$ is an approximation of $\delta_I$ (Gibson, 2002), we decided to estimate $\delta_I$ as the intersection of the LMWL with the LEL in the revised version as the latter method is more conventional (Gibson et *al*., 1993; Yi et *al*., 2008). While we agree with the challenge raised by the reviewer, the particularity of our study area (small spatial extend, homogeneity of land cover and limited differences in elevation), we assume that hypothesizing that $\delta p$ is the same for all the lakes is a fair assumption as explained above. This is of course something that we mentioned in the revised manuscript. We also recognize that $\delta I$ can also be influenced by the isotopic composition of surface inflowing waters (which for the most part are groundwater springs and thus made up of groundwater). Only two lakes receive inflowing water from local streams (the volume of both of those is supposed to be small in comparison to the lake volume and those streams are intermittent in the sense that they were not flowing during each of the three field campaigns; the latter observation was confirmed by local park officials). We thus assume that inflowing water can change the isotopic composition of a handful of lakes but this change is assumed to be limited. This limitation was stated in the revised manuscript.

References:
Gibson, J.J.: A new conceptual model for predicting isotope enrichment of lakes in seasonal climates, PAGES News, 10, 10-11, 2002.

Gibson, J.J., Edwards, T.W.D., Bursey, G.G., and Prowse, T.D.: Estimating evaporation using stable isotopes: quantitative results and sensitivity analysis for two catchments in northern Canada, Nord. Hydrol., 24, 79–94, 1993.

Yi, Y., Brock, B.E., Falcone, M.D., Wolfe, B.B. and Edwards, T.W.D.: A coupled isotope tracer method to characterize input water to lakes, J. Hydrol., 350, 1–13, doi:10.1016/j.jhydrol.2007.11.008, 2008.

Line 234-235: Incorrect calculation of dE. Very puzzled by this statement.

Indeed, this is an incorrect calculation of δE. This is obviously a major typo. The right formula was inserted in the rewriting of that section (see above).

Line 242-244. I gather you are summarizing here, but you have two springtime samples and one late summer sample, which makes this a bit awkward.

Temperature was only measured during the August 2014 field campaign. Thus we are only comparing samples collected during the same week. This was specified in the manuscript.

Line 250: 'Local Evaporation Line'

This was corrected in the text.

Line 251-252: Concern about overlapping isotope signatures of gw and precip confirmed!

This justified concern was addressed above.

Line 256-257: Awkward incorporation of elevation here. Seems like this should be saved for the next paragraph.

We agree with the reviewer that the incorporation of elevation should be done in the following paragraph. This was changed accordingly.

Figure 2b: Label your figures 'a' and 'b'. First awkward imposing of lake type here, without rationale being provided.

Labels a and b were added to Figure 2. As to not impose the lake typology early in the manuscript, we aggregated all the lake water samples as one category.

Line 266: A small thing but I would replace 'indicating' with 'suggesting'. Of course, this may be just one factor driving the water balance. At this point in the article, it seems the authors are overly anxious to get to the conclusion without proper development of the results and interpretation.

This was adapted in the text as suggested. We agree with the reviewer that the result section should only state the results and that interpretation should be withhold at this point.

Figure 3: I don't understand the application of the breakpoint analysis.

Please see our response to your earlier comment.

Line 278: delete 'high'

This was adapted in the text as suggested.

Line 279: An E/I ratio of '8' cannot be quoted so directly if using a steady-state model.

We agree with the reviewer that an E/I ratio of 8 shouldn't be quoted directly due to the limitations of the Craig-Gordon model and we adapted the quotation.

Line 280: More awkward imposing of lake types without rationale. Same with Figure 4, which isn't a very effective way of presenting E/I ratios, as they are calculated from the isotope measurements.

We agree about the non-effectiveness of Fig.4 as E/I is indeed computed from $\delta^{18}O$ (and we realized this beforehand). The original goal of this figure was to show the reader the range of E/I that exist in our data set. We thus plotted E/I and the electrical conductance instead.

We also agree that we awkwardly impose our lake typology but at the same time adding the typology adds more information to the figure and allows the reader to infer the importance of lake type on lake E/I as she or he reads further in the manuscript. The details related to the lake typology was advised in the figure caption. In order to take the reviewer's concern into account, we duplicated this figure as Fig. 4a (without the lake typology, which was referred to in the result section) and Fig. 4b (with the lake typology, which was referred to in the discussion).

Line 296: Are they really conservative isotope tracers if they are changing as a consequence of evaporation?

The reviewer is correct to challenge the conservativeness of isotope tracers as they indeed change as a consequence of evaporation. We did label isotope tracers as conservative in comparison to chemical tracers because isotopes are relatively conservative in reactions with catchment materials and retain their distinctive values until they mix with water of different isotopic composition (Kendall and McDonnell, 1998). We acknowledge that this terminology can be a little confusing and we included a small definition of what we mean by conservative and non-conservative tracers in the introduction.

Reference: Kendall, C. and Caldwell, E.A.: Fundamentals of Isotope Geochemistry. In: "Isotope Tracers in Catchment Hydrology", Edited by: Kendall, C. and McDonnell J.J., Elsevier Science B.V., Amsterdam, pp. 51-86, 1998.

Table 1. More confusion about the breakpoint analysis, confounded by listing of _'282' m in the table and '242' m on p. 14 and elsewhere.

The elevation is 282 m, not 242 m. A previous typo earlier in the manuscript made it confusing. This was corrected throughout the manuscript.

Lines 323 to the end of this section: Again, imposing the lake types here is simply not warranted. Extremely confusing to the reader. Data needs to be presented first. Then it can be interpreted. Then a classification system can be developed. P. 15. At this point in the paper, I had too many questions to be able to critically evaluate further. But as I say in my opening comments, I believe the story may largely be ok, it is just the development of that story that really needs to be overhauled.

As commented earlier in our response, we agree with the reviewer's diagnostic that the manuscript needed to be structurally improved so that the results should be stated first, then interpreted, after which the typology could be developed and explained.

Figure 7. First (and perhaps only) characterization and use of the seasonal differences (although literal use of the very elevated E/I ratios is beyond the limitations of the model). Difficult to detect if this is really utilized in the Discussion. Is it relevant? Or can more be made of it? But again, as written, the reader is forced to accept the lake type classification.

We understand that it is confusing to impose the lake typology in the result section. To make things clearer and less confusing for the readers, we moved Fig. 7 (now Fig. 8) to the discussion section (although introducing figures in the discussion is not very conventional) and we added a figure in the appendix that was referred to in the result section with part a showing changes in water balance between the sampling campaigns for each lake ranked by elevation and with part b showing changes in water chemistry between the sampling campaigns for each lake ranked by elevation (Fig. A2). The interpretation of those changes is mostly relevant when relating to the lake typology given that the observed changes were dependent on the lake typology. For this reason, we think this is relevant to include it in the manuscript.

Line 369: replace 'high' with 'short'

This was adapted in the text as suggested.

Line 373-374. So now we're finally getting some explanation of the different lake types, but this is confusing too. How do you know that gw is causing the difference here (which is not explicitly stated)?

We already addressed those two concerns above: (1) the manuscript was restructured as not to impose the lake typology too early to the reader; (2) we addressed the issue of differencing the relative contribution of precipitation and groundwater to balance and the assumption that in our study area, groundwater is causing the difference in water balance as precipitation is supposed to be uniform in our study area due to its small size.

Line 379: replace 'composition' with 'water chemistries'

This was adapted in the text as suggested.

Line 384, 386: How do you know the 1 km statements? Where does that come from? But here, I get the sense that the gw role is mainly based on the water chemistry differences, and less so on the isotope data.

The 1 km value was quoted from the literature (Bertrand et *al*., 2011) as a typical example of spatial scale. However we acknowledge it can be hazardous to cite values that applies to general settings so we deleted this value. Because we labelled isotopes as conservative tracers and chemical tracers as non-conservative (with respect to their ability to react chemically with the watershed materials), we indeed rely on water chemistry to infer whether the flows are local or intermediate.

Reference: Bertrand, G., Goldscheider, N., Gobat, J.M. and Hunkeler, D.: Review: from multi-scale conceptualization to a classification system for inland groundwater-dependent ecosystems, Hydrogeol. J. 20, 5–25, 2012.

Line 404: This classification section has been pre-empted by all the comments before this point on lake types, which is the most awkward aspect of this paper. Sorry to keep emphasizing this. I'm sure it can be resolved!

As commented earlier in our response, we agree with the reviewer's diagnostic that the manuscript had to be structurally improved so that the results are stated first, then interpreted, after which the typology can be developed and explained and that the typology shouldn't be imposed too early to the reader. Efforts were made to restructure the manuscript to address this concern.

Line 407: 'conservative water tracers (water source)' – I am very confused by this terminology. The isotope data reflects more than just source water. It is modified by varying degrees of evaporative enrichment.

We agree with the reviewer that isotopic tracers reflect water sources and their level of evaporative enrichment as well. As mentioned above we defined conservative and non-conservative tracers with respect to their reactiveness with the watershed materials, which was specified in the introduction.

Line 428: Awkward sentence.

We agree with the reviewer that the sentence is awkwardly phrased and is confusing. We rephrased it as: "In seepage lakes, water comes in as groundwater in-seepage and is returned to the groundwater system as out-seepage".

Line 436- 437: 282? Or 242? But I still don't understand how this elevation was defined.

The elevation is 282 m. A previous typo earlier in the manuscript made it confusing. The details of how it was estimated is provided above.

Line 451-453: Not a sentence.

We agree with the reviewer that those two lines didn't constitute a proper sentence. We will rephrase it as: "These lakes receive a substantial inflow of groundwater due to their geological setting and their water isotopic and chemical composition is similar to the one of groundwater springs and outlets".

Line 473: What is meant by the word 'mounded'?

Mounded lakes are lakes positioned above the water table that receive most of their water input from direct precipitation as defined by the ALSC Lake Classification System (Newton and Driscoll, 1990). The word mounded was discarded following the comments of the other reviewer (see response to reviewer #1).

Reference: Newton, R.M. and Driscoll, C.T.: Classification of ALSC lakes, in: Adirondack Lakes Survey: An Interpretive Analysis of Fish Communities and Water Chemistry, 1984-87, Adirondack Lakes Survey Corporation, Ray Brook, NY., 2-70 to 2-91, 1990.

Line 545: 'those'?

Those was changed to these.

Line 575: 'extent'

This was corrected in the text.

Line 600-602: Interesting. Shouldn't these activities be mentioned in the Intro to help rationalize the study?

As suggested we added this to the rationale in the introduction.

Line 632: Awkward phrasing.

We agree with the reviewer that the sentence is awkwardly phrased and is confusing, we rephrased it as: "The physical and chemical characteristics of lake water allowed the development of a lake typology that is made up of three main types of lakes".

Line 635: delete 'are'

This was corrected in the text.

I like the conclusion – well stated and summarized!

Thank you.

---

## Referee Report (RR1)

Review Comments

**General comments on the manuscript:**

It is evident that the authors maximized their use of conservative and non-conservative tracers to effectively characterize lake typology. They carefully considered most physical variables influencing their results and their data shows strong distinction among groupings, as backed by their statistical analyses.

This paper provides important data that characterizes hydrological conditions in an area that is expected to experience more landscape (development) changes. The typology of lakes provides an important baseline for comparison to future hydrological regimes that may be altered.

I am a fan of the study site description.

The field measurements and water sample collection provide important details for other researchers to consider when doing this type of analysis.

251 – I'm a bit confused about their assumption that the lakes are at isotopic steady state when their data shows that recharge lakes have EI > 1.

350 – Is the lower DOC because of dilution over a greater water volume?  Would be interesting to know if catchment land cover could explain some of that difference.  I see they get at that in the discussion at 453 with catch area:lake area and again at 605, but land cover has been left out. That's fine, but the possibility that land cover influences non-conservative tracers should be mentioned.

**Minor edits:**

95 – Strange end to the sentence.

164 – Change 'digitalized' to digitized.  What imagery was being used in Google Earth?  That is what should be mentioned.

171-173 – Sentence should be cleaned up.

Fig 2b – Typo – change to 'Local Evaporation Line'

213 – I assume the isotope work was done in their own lab since no other one is mentioned.

Table 1 – Sort rows in order of nutrients, ions, isotopes. Caption should just say that 'lower and upper elevation ranges represent the standard deviation'

425 – 'But those are for the most…'??

483 – Use different choice of word/phrase for 'supposed to be'

634 – While it was noted that the recharge lakes are more susceptible to evaporative-drawdown during dry conditions, it could also be noted that discharge lakes may be more susceptible to contamination as development encroaches into the source water locations. The point could also be made around 669.

683 – Could be mentioned that paleo work could provide a reference for evaluating whether present hydrological conditions are within the range of natural variability. Furthermore his would significantly complement their baseline knowledge of hydrological conditions as development continues in the area.

**Notes on previous reviewer comments:**

The previous reviewer had many useful comments for the authors to consider, and overall the authors responded with the necessary revisions. I agree with the authors' responses where the reviewer comments questioned the utility of their approach. In particular, the reviewers comment about the authors' 'indirect' evidence (chemistry and isotopes) of findings suggests his/her lack of confidence in the approach despite the clear evidence presented in the paper. As the authors note, the resources required to make the necessary direct measurements would be immense, but are clearly detectable using more feasible and sustainable approaches that can be applied at greater spatial scales. This point could even be showcased more in the paper.

---

## Author Response (AR3)

**Response to reviewer #3 (Kevin Turner) comments on "Use of water isotopes and chemistry to infer the type and degree of exchange between groundwater and lakes in an esker complex of northeastern Ontario, Canada" by Boreux et al.**

In black: reviewer's comments.
In blue: our answers and/or what we added or changed in the manuscript.

We thank reviewer Kevin Turner for his comments which aided in improving a new version of the manuscript.

**General comments on the manuscript:**

It is evident that the authors maximized their use of conservative and non-conservative tracers to effectively characterize lake typology. They carefully considered most physical variables influencing their results and their data shows strong distinction among groupings, as backed by their statistical analyses. This paper provides important data that characterizes hydrological conditions in an area that is expected to experience more landscape (development) changes. The typology of lakes provides an important baseline for comparison to future hydrological regimes that may be altered. I am a fan of the study site description. The field measurements and water sample collection provide important details for other researchers to consider when doing this type of analysis.

251 – I'm a bit confused about their assumption that the lakes are at isotopic steady state when their data shows that recharge lakes have EI > 1.

Indeed, a significant proportion of the lakes in the study area are small closed basin lakes that receive no direct surface inflow. As such, the assumption of steady state (*i.e.* undergoing evaporation while maintaining constant volume) may not be valid for all the lakes. We thus calculated the evaporative loss fraction of the pool volume for recharge lakes and kept the E/I ratios for discharge and seepage lakes as those likely receive significant groundwater flow to satisfy the steady state assumption. E/I ratios have been used in small lakes with no direct surface inflow but significant groundwater input in similar settings (e.g. Arnoux et al., 2017a; Arnoux et al., 2017b). Details of those calculations have been added to the method section.

350 – Is the lower DOC because of dilution over a greater water volume? Would be interesting to know if catchment land cover could explain some of that difference. I see they get at that in the discussion at 453 with catch area:lake area and again at 605, but land cover has been left out. That's fine, but the possibility that land cover influences non-conservative tracers should be mentioned.

DOC tends to be lower in deeper lakes, likely due to differences in residence times and/or mixing rates. The reviewer is absolutely correct in stating that land cover affects lake trophic status and water chemistry. Many previous studies have demonstrated this. However, as mentioned in the study site description and the discussion, the study area is almost completely covered with boreal forest. As such, our study area does not offer the possibility to assess the influence of land cover on lake water tracers. We also discussed the influence of other factors as suggested by the other reviewer.

**Minor edits:**

95 – Strange end to the sentence.

The sentence was reworded as follows: "Nonetheless, studies that have combined chemical and isotopic approaches to investigate the connectivity between groundwater and lake water at the landscape level and for a large cluster of lakes are lacking".

164 – Change 'digitalized' to digitized. What imagery was being used in Google Earth? That is what should be mentioned.

We changed 'digitalized' to 'digitized' as suggested. The imagery date (7/26/2005) was mentioned.

171-173 – Sentence should be cleaned up.

The sentence was rephrased as follows: "Since all lakes in the study area are kettle lakes, which are characterized by steep slopes on their shore over a small distance, buffer zone of different widths were produced. The buffer width of 100 m was chosen as this distance showed the best correlation with water tracers."

Fig 2b – Typo – change to 'Local Evaporation Line'

This was adapted in the text as suggested.

213 – I assume the isotope work was done in their own lab since no other one is mentioned.

The name of the lab where the isotopes were processed (FaBRECC lab at Queen's University) was mentioned as suggested.

Table 1 – Sort rows in order of nutrients, ions, isotopes. Caption should just say that 'lower and upper elevation ranges represent the standard deviation'

Rows were sorted and the caption was adapted as suggested.

425 – 'But those are for the most…'??
We replaced "for the most" by "mainly".

483 – Use different choice of word/phrase for 'supposed to be'

We deleted "supposed to be" as suggested.

634 – While it was noted that the recharge lakes are more susceptible to evaporative-drawdown during dry conditions, it could also be noted that discharge lakes may be more susceptible to contamination as development encroaches into the source water locations. The point could also be made around 669.

The point was made and developed around line 669 as suggested. We did not mention this in line 634 as we wanted to keep this paragraph solely on hydroclimatic conditions.

683 – Could be mentioned that paleo work could provide a reference for evaluating whether present hydrological conditions are within the range of natural variability. Furthermore this would significantly complement their baseline knowledge of hydrological conditions as development continues in the area.

We mentioned that the typology of lakes provides an important baseline for comparison to future hydrological regimes that may alter them as suggested. This was also stated in the introduction.

**Notes on previous reviewer comments:**

The previous reviewer had many useful comments for the authors to consider, and overall the authors responded with the necessary revisions. I agree with the authors' responses where the reviewer comments questioned the utility of their approach. In particular, the reviewers comment about the authors' 'indirect' evidence (chemistry and isotopes) of findings suggests his/her lack of confidence in the approach despite the clear evidence presented in the paper. As the authors note, the resources required to make the necessary direct measurements would be immense, but are clearly detectable using more feasible and sustainable approaches that can be applied at greater spatial scales. This point could even be showcased more in the paper.

We totally agree with the reviewer and we added the following sentence in the introduction: "The use of water tracers was preferred to direct measurements because tracers (1) have proven to be good indicators of groundwater-lake water interactions and (2) constitute a time and cost-effective approach that can be applied at a greater spatial scale for a given time."

**Response to reviewer #4 comments on "Use of water isotopes and chemistry to infer the type and degree of exchange between groundwater and lakes in an esker complex of northeastern Ontario, Canada" by Boreux et *al.**

In black: reviewer's comments.
In blue: our answers and/or what we added or changed in the manuscript.

This study examines spatial variability in conservative and non-conservative tracers across 50 "kettle lakes" in northern Ontario. Authors employ a combination of correlation, ordination, and simple water balance analyses to develop a "lake typology." The authors argue that landscape position is dominant driver of lake hydrology, suggesting lakes at higher elevations are "recharge" lakes (ie they contribute to local groundwater aquifer) and lakes at lower elevation are "discharge" lakes (ie they receive water from local groundwater). Finally, authors suggest recharge lakes are more sensitive to short term changes in hydroclimatic conditions than discharge lakes.

While this manuscript is in revision, I am reviewing this manuscript for the first time. In general, I found the manuscript quite interesting and worthy of publication. However, unfortunately, I believe both the narrative structure and analyses should be developed further before publication. In particular, more information about analyses is needed in the methods section, results associated with landscape position/morphology are overstated, and conclusions about inter-annual variability are confusing/unconvincing.

Below I provide both general and specific comments in an effort to help the authors improve their manuscript.

We thank the reviewer for his/her thorough comments. These comments helped us to improve the manuscript. We followed the majority of the reviewer's comments and suggestions when we prepared the second revised version of this manuscript.

**General Comments:**

1) The authors should work to further develop their narrative. In particular, authors should work to make their main points more easily accessible to readers. One approach is to structure the paper to target three different types of readers: (1) readers who will only review the abstract/conclusion/figures, (2) readers who will lightly skim discussion, and finally, (3) readers who will thoroughly read the manuscript. Because the vast majority of readers will fall into the first category, it is imperative that authors tell a coherent story with the abstract/conclusions/figures.

The abstract was adjusted to mirror the manuscript, a new section in the method was added, more results on the relation between water tracers and lake morphology was added in the result section, and efforts were made to highlight the rationale and the implication of this study in the introduction and discussion.

2) The abstract should be streamlined. (Refer to comment #1 above.) The current abstract reads like a list of facts, and provides minimal synthesis of those facts. While this is a stylistic decision, I prefer abstracts that mirror the structure of the paper [eg intro, broad objective statement, study design, 2-3 major results, and a brief conclusion that links to objective statement]. This helps the reader extract needed information, and for readers who are taking a deeper dive, it gives them a road map to refer back to if needed. Also, in general, the abstract should be constrained to one paragraph.

The abstract was constrained into one paragraph and modified according to the reviewer's comment to mirror the manuscript more.

3) The methods are incomplete. There is no description of the ordination analysis, comparative statistics, correlation, or breakpoint analyses. While it is often acceptable to omit simple analyses from a methods section, I do not believe it is appropriate in this case. Further, this will help guide readers as they pick through the results section.

We added a section in the methods named "Statistical analysis" as suggested as follows: "Linear regressions were used to assess the degree of co-variability between quantitative variables while logistic regressions were utilized to assess the relations between binary variables and quantitative variables at the 0.05 level. Breakpoint analysis or segmented regression was used to detect any change of trends in water tracers along an elevation gradient and to produce subsequent higher level groupings of lakes. Breakpoints that were significant at the 0.05 were averaged to obtain the elevation of the "breakpoint line". A non-metric multidimensional scaling (NMDS) was run to assess the differences among lake types in a 2 dimensions ordination space using non-scaled values of electrical conductance, Ca, $\delta^{18}O$ and $\delta^2H$ as input variables and Euclidean distance as dissimilarity measure; no rotation was applied. A Wilcoxon signed-rank test was subsequently applied as a post-hoc analysis for all lake-water variables that were above detection limits to determine if differences among the different types of lakes were statistically significant at the 0.05 level as most of the Shapiro-Wilk test for normality revealed that most variables were not normally distributed. An analysis of similarity (ANOSIM) was also carried out as a complement to determine if within group similarity was significantly greater than in-between group similarity at the 0.05 level. All statistical analysis were performed in $R$ on the data from the August 2014 campaign as it was the one with the most samples".

4) The description of the sampling design is ambiguous. For the synoptic sampling, 50 lakes were sampled across three sampling campaigns. Were all lakes sampled during each campaign, or were 50 total lakes sampled and only some sampled multiple times?

We agree with the reviewer that the sampling design should be more clearly described. We reworded the sentence at line 178-179 as follows: "50 lakes were sampled (29, 28 and 50 lakes during the June 2013, June 2014 and August 2014 campaign respectively), as well as a number of streams (lake outlets and lake inlets) and groundwater springs".

5) Authors should discuss the limitations of the temporal component of their sampling design. Throughout the document, the authors suggest the groundwater and precipitation observations are too similar to differentiate. I'm not sure if I agree with this statement, as it vastly oversimplifies associated isotopic fractionation and solute mixing processes. (Specifically for water isotopes, see recent review paper by Sprenger et al., 2016 in Reviews of Geophysics. Note, the detail presented in the Sprenger review is admittedly beyond the scope of this study.) While I think the sampling design is adequate for the given inference, it may be worth discussing the benefits of greater temporal resolution sampling.

We touched on the resemblance between the isotopic composition of precipitation and groundwater as suggested by one previous reviewer. Indeed, mean precipitation and groundwater often display similar isotopic signatures as they both tend to retain their original isotopic composition because they undergo little to no evaporation. This is shown by the data: groundwater springs have isotopic values similar to mean annual precipitation, respectively -14.7‰ to -13.1‰ for $\delta^{18}O$ and 105.5‰ to -96.0‰ for $\delta^2H$. One of the previous reviewers was concerned that the spatial extent of the study area would be large enough to have significant differences in precipitation (and their isotopic signature), which could introduce a bias in the interpretation. We do agree with the reviewer that the isotopic signature of precipitation (Fig. 2a) and groundwater vary on an annual basis, and although they have similar values, those are different: groundwater values were more depleted than the ones of precipitations during the sampling campaigns that took place in the growing season (and those would be likely more enriched during the winter as snow is more depleted). We specified this briefly in our results section.

The authors acknowledge the benefits of having a greater temporal resolution of sampling. When comparing the variability between sampling campaigns, we highlighted briefly the fact that a greater temporal resolution of sampling would reinforce positively our interpretations in the discussion.

6) I am concerned about the assumption dV=0 in the water balance analysis. In large lakes with consistent inflow and outflow channels, this assumption is likely appropriate. However, in small lakes without surface connections, this assumption is absolutely not valid. [As noted in the authors

discussion, ephemeral lakes are actually defined by their seasonal variation in volume.] It appears the lakes sampled in this study are in-between these two endmembers. Therefore, the authors should caution readers about the assumption dV=0, and further, cite other papers that use a similar approach to develop I/E ratios to justify their methods.

As for the two to three "ephemeral lakes" in our study area those correspond to dry kettles as specified in the manuscript and those are not considered lakes per se or even ponds. Those areas refer to small kettles depressions that become muddy during moist conditions. But, indeed, their presence is indicative of the variation of the water table in the study area and changes in volumes of lakes deprived of surface connections. As such, we agree with the reviewer than the hypothesis of steady state may not hold for specific lakes. As mentioned to the other reviewer, we thus calculated the evaporative loss fraction of the pool volume for recharge lakes and kept the E/I ratios for discharge and seepage lakes as those likely receive significant groundwater flow to satisfy the steady state assumption. E/I ratios have been used in small lakes with no direct surface inflow but significant groundwater input in similar settings (e.g. Arnoux et al., 2017a; Arnoux et al., 2017b). Details of those calculations have been added to the method section.

7) I encourage the authors to use non-parametric statistics. (eg Use a Wilcoxon rank-sum test instead of an ANOVA.) I would wager that their data likely violate normality assumptions.

Some of our variables (but not all) indeed do fail the normality test (tested with a Shapiro-Wilk test), although most of the variables that do fail the normality test are near normal as evidenced by QQ plots. In order to improve the statistical analysis of the manuscript we switched our post-hoc analysis from an ANOVA to a Wilcoxon signed-rank test as suggested by the reviewer. Tab A2 and A3 were adjusted in the appendix. The outcome of the test was very similar to the ANOVA performed (except for a limited amount of variables).

8) I would like more information about the NMDS analysis.

In the methods section, the following information should be provided [at a minimum]: What was the input data? [Was it from across all 50 sites and all three synoptic sampling events?] Was the data scaled before running the analysis? What was the dissimilatory measure? How many iterations were used to fit the model? Were the final results rotated so the dominant gradient varies along the primary axis? What statistical package did the authors use (R, SAS, PC-ORD?).

In the results section: How many axes were utilized, and why was that decision made? Did the model converge, and if so, after how many iterations? What was the resulting stress?

> What was the input data? [Was it from across all 50 sites and all three synoptic sampling events?] All 50 lakes from the August 2014 campaign as it was the one with the most samples.

> Was the data scaled before running the analysis?
No scaling was applied to the data prior to the calculation of the dissimilarity measure (we tested the raw data and the data transformed with a z-standardisation. The latter gave different coordinates values but the same relative distances in ordination space and the exact same polygons as displayed in Fig. 6).

> What was the dissimilatory measure?
Euclidean distance

> How many iterations were used to fit the model?
35 iterations.

> Were the final results rotated so the dominant gradient varies along the primary axis?
No rotation was used.

> What statistical package did the authors use (R, SAS, PC-ORD?).
The software R was used.

> How many axes were utilized, and why was that decision made?
Two axes were utilized to visualize the data in two dimensions.

> Did the model converge, and if so, after how many iterations?
The model converged after 35 iterations.

> What was the resulting stress?
Stress = 0.023

A summary of those details was added in the method section as suggested.

9) I appreciate the authors attempt to classify lakes in Figure 7. This is one of the more compelling parts of the manuscript. To further clarify the discussion surrounding the differences between higher level groups, I encourage the authors to add an additional plot that displays differences between the higher level groups. Personally, I love boxplots because of the amount of information they provide! Also, authors may consider showing differences across groups in either ordination or dataspace. (A multivariate, nonparametric test like MRPP or NP-MANOVA may be appropriate.) Further, cluster or CART analyses could be used to differentiate groups. However, just to be clear, the listed analyses are simply suggestions for improvement, and authors should not feel obligated to use them!

We really appreciate the reviewer's suggestions, especially for guiding us to specific statistical analysis. An ANOVA (now changed to a Wilcoxon signed-rank test) and NMDS analysis were processed as a post-hoc analysis: the first one in the appendix and the second one as a plot and a commentary in the discussion. While we totally agree with the reviewer that boxplot are effective in showing differences among groups, a lot of figures would need to be included while a Wilcoxon signed-rank test table show those differences in a single table that is more synthetic. The NMDS showed the differences among groups for the upper two higher level groups. Plus, an ANOSIM was also done as a complementary analysis of the NMDS. Post-hoc analysis used in this study all converged to the same conclusions. As such, we think including more post-hoc analysis would be overwhelming for the reader.

As for additional clustering analysis, clusters were made using different dissimilarity measures and those provided similar groups as the method we used (while sometimes slightly different groups). Following this, we thought that including another method of classification would be redundant and confusing for the reader.

10) The authors' conclusions about landscape vs local drivers of hydrology are likely overstated given the sampling design. The authors develop a compelling argument that landscape position (and associated geologic setting) drive lake hydrology. However, their analysis of hydrogeomorphic features at each site is very limited due to data availability. Important site-specific variables like network order (ie how many lakes drain into the lake in question), contributing watershed area, and soil characteristics like specific yield can all be used to explain observed variability in hydrologic data. While it is understandable/acceptable that authors did not collect these variables, I think they are over reaching by saying landscape position is more important than local hydrogeomorphic characteristics. I would encourage the authors to add a caveat to this statement, highlighting measures that could be used to better explore local geomorphic drivers.

As suggested by the reviewer (and the other reviewer as well), we developed this section further and we listed other factors that may explain the observed variability in the hydrological data. We also added some correlation results in the result section to better link the result and discussion section as suggested earlier in the reviewer's general comments.

11) The results presented in figure 8 are quite confusing. The discussion should restructured in such a way that clearly delineates annual and inter-annual variability in ET and E/I ratios. As presented, the conclusions (L708-712) that recharge wetlands are more sensitive to climate extremes that discharge are unconvincing. I would encourage authors to present a conceptual model of annual variability for both recharge and discharge wetlands, highlighting how results presented in Figure 8 support the proposed model. Also, consider visualizing this information using another format.

We thank the reviewer for his/her input on this section. Our goal was not to discuss annual or inter-annual variability but observed differences between sampling campaigns, which were characterized by slightly different hydroclimatic conditions. Evaporation rates should be relatively similar among lakes as our study area consist basically of a rectangular zone of 12 km by 6 km. E/I ratios differ among lakes as groundwater inflow (I) can buffer evaporative losses (E). ET (evapotranspiration) could not be computed using the Penman formula as no solar radiation data was available for the period of interest. Our real goal was to relate observations in changes in E/I ratios and EC along a groundwater gradient between our sampling campaigns and how it relates to the existing literature. As for the format, bar plots seem to be the most appropriate format as we compare different groups of lakes while tracking changes in time.

But we do agree with the reviewer that a conceptual model combined with the existing data from Fig.8 would produce interesting synergies and further support our observations. While the literature and our general knowledge of our study area could allow us to produce a conceptual model displaying the annual and inter-annual variability, such a model would remain hypothetical as no continuous significant data set could back up such a model. As a result, we decided to adapt the conceptual model developed by Webster et *al*. (1996) in similar settings. The conceptual model of Fig 8a describes the relationship between the direction and magnitude of lake E/I and EC changes during drier conditions to lake type defined by the degree to which lakes interact with groundwater. We think that this type of conceptual model is more appropriate to support the observations we comment in this section of the manuscript.

Besides developing a conceptual model, (1) we acknowledged that some of our observations (such as changes in EC between discharge and seepage lakes) were not clear and used the conceptual model to explain trends; (2) we also cautioned the reader that it is difficult to draw conclusions on the inter campaign variability based on the limited sampling frequency; and (3) we deleted the mention of extreme conditions as we do not have such data.

12) Finally, I would encourage authors to explicitly link their results to management activities in northern Ontario. What information from this study will be useful for managers? Maybe even write this section for them. Authors begin to do this when describing mining activities. However, there is no discussion how the "cottage development" industry could use the information derived from this study.

We agree with the reviewer that the implications of our results for management purposes would benefit from further development. This point was also brought up by the other reviewer. We explained a little bit further in the paragraph how mining activities and cottage development can impact groundwater-dependent systems in the area.

**Specific line-by-line comments:**

L33 Abstracts should be 1 paragraph.

The abstract was restructured into a single paragraph and was revised based on comment 2) of the reviewer.

L72 Typically, anions such as Cl and Br are considered conservative tracers. While they may not exhibit conservative behavior in all settings, they are much less reactive than tracers like DOM, N, and P.

We agree with the reviewer that Cl and Br anions are generally viewed as conservative tracers (especially with respect to NPOC, N or P) but we defined those as non-conservative in comparison to water stable isotopes. Also, Cl and Br don't always behave conservatively as they can be affected by residence times.

L102 This is an odd place for a comma.

The coma was likely a typo and was deleted. Thanks for pointing this out.

L103 Lake typology is a new term for me, and potentially for other readers as well. Maybe consider defining it here?

A definition of lake typology was added in the introduction as suggested and we better developed the rationale for it.

L130 Please rephrase. "The esker are" is somewhat ambiguous, especially since the term eskers has not been defined.

We agree with the reviewer that the sentence was ambiguous. The sentence at line 130 was deleted and the term esker was defined as follows a little bit further in the text: "long sinuous ridges of coarse grained glaciofluvial sediments in deposits oriented in a north-south direction".

L164. How were features digitalized in ArcGIS 10.3 from google earth? I think I know what you mean, but please be more explicit here.

We agree with the reviewer that the sentence was confusing. It was reworded as follows as suggested by the other reviewer: "Lakes and other geographic features were digitized from Google Earth using the imagery dating from 7/26/2005".

L171-173 I'm not sure what this is referring too. Please rephrase to be more clear.

The sentence was rephrased as follows: "Since all lakes in the study area are kettle lakes, which are characterized by steep slopes on their shore over a small distance, buffer zone of different widths were produced. The buffer width of 100 m was chosen as this distance showed the best correlation with water tracers."

L233 How far away were these samples taken from the study area. Note, it's acceptable that precipitation samples were taken "off-site" given the level of detail/inference in this paper. However, this is still an important detail.

The distance and the orientation from the station was referenced according to the closest city (Timmins). We adapted the distance the orientation (*i.e.* 125 km NW) in the text as suggested.

L251 Please define hydrologic and isotopic steady state, and then use previous studies to confirm the validity (and limitations) of those assumptions.

We added a definition and cited a couple of studies that used the Craig-Gordon model in similar settings as suggested above by the reviewer (point 6).

L326 Please add a parenthetical designation of "E/I" ratio

This was adapted in the text as suggested.

L361 Please describe the breakpoint analysis in more detail. This detail should likely go in the methods section.

Breakpoint analysis was used to detect when there was a change in a trend in the data. In our case, we have a gradient of lakes ranked according to elevation (thus a continuous series of data just like a time series) and we observed that lakes behaved isotopically and chemically differently in two groups according to elevation. A breakpoint analysis was done and indeed revealed that the breakpoint was significant for most variables at around the same elevation. This is particularly relevant because the breakpoint line allows us to map what we believe to be the groundwater recharge and discharge zones. A brief sentence was added in the method section.

L522 Again, please describe logistic regression in the methods section.

A brief sentence was added in the method section.

Figure 1. Maybe add a line to map to indicate the potential location of conceptualized cross section presented in b

We agree with the reviewer that indicating the potential location of the conceptualized cross section could be useful to the reader. However, in order not to overcrowd the map, we specified that thus could be located at latitude 48°35'0"N in the figure caption (which is displayed on the figure on the left and right).

Figure 4 and 5. Authors should consider flipping their axis.

While we do agree with the reviewer that elevation should be displaying in the x-axis as we relate the isotopic and chemical signature of the lakes to elevation, displaying elevation in the y-axis allows the reader to see the abrupt transition that occur near the breakpoint. Elevation is typically displayed on the y-axis when looking at variables along an elevation gradient.

Figure 6. How were these grouping boundaries defined? Are you connecting points at the fringes, or are these polygons the result of an analysis?

Those boundaries consist of the connection of points at the fringes of the groupings defined by the lake typology.

---

## Author Response (AR4)

**Response to reviewer #4 comments on "Use of water isotopes and chemistry to infer the type and degree of exchange between groundwater and lakes in an esker complex of northeastern Ontario, Canada" by Boreux et *al*.**

In black: reviewer's comments.
In blue: our answers and/or what we added or changed in the manuscript.

I am reviewing this manuscript for the second time. I believe the authors have sufficiently responded to my previous requests, and thus recommend acceptance after minor revisions. This is an interesting manuscript, and I think the results will be a useful contribution to both hydrologists working in northern latitudes and practioners working in the study region.

Below are several detailed comments for the authors to address. While I would happily review this manuscript one more time, I believe these changes can be made without further peer review.

We thank the reviewer for reading the manuscript one more time and for his/her additional comments. These comments helped us to further improve the manuscript. We followed the majority of the reviewer's comments and suggestions when we prepared the third revised version of this manuscript.

L29: "The hydraulic midline" is a bit of an ambiguous term. I would encourage authors to remove it from the abstract and then define it later in the manuscript before using it.

We agree with the reviewer that this terminology is ambiguous for an abstract. We removed the term from the abstract and defined it further in the manuscript as suggested below in the comments.

L60: Authors may want to consider citing Thorslund et al., 2018 Thorslund, J., M.J. Cohen, J.W. Jawitz, G. Destouni, I.F. Creed, M.C. Rains, P. Badiou, and J. Jarsjö, 2018. Solute Evidence for Hydrological Connectivity of Geographically Isolated Wetlands. Land Degradation & Development 29:3954–3962.

We cited Thorslund et *al*. (2018) as suggested.

L73: "To as" appears to be a typo

This was adapted in the text as suggested.

L95: Authors should define groundwater connectivity.

We defined "groundwater connectivity" as suggested.

L97: Authors use multiple terms to describe "tracers". These terms should be clearly defined, then implemented consistently throughout.

We clarified what we meant by "tracers" as suggested.

Fig 1. When this goes to print, authors should make sure the text is large enough to be useful.

We increased the front size as suggested. We also took the liberty to add colours to all the figures in the manuscript for more clarity as well.

L175: While this metric is certainly useful, I'm not sure that it's appropriate to call it watershed slope, as the actual lake's watershed is certainly larger than the delineated 100m buffer.

We replaced watershed slope with the term "perimeter slope" throughout the manuscript.

L291: Here, while I understand the utility of calculating E/I ratios of springs and rivers for comparison, the assumptions of the approach breaks down. If nothing else, authors should at least make a statement indicating as much.

We agree with the reviewer that E/I ratios of springs and rivers were included only by way of comparison. We made a clarification as suggested.

L305: The authors should justify using E/I ratio and f values as equivalents.

We justified this as follows: "While E/I ratios and f values are two metrics calculated with different formulas, they can be compared equivalently as they both represent the mass balance of the lakes in dimensionless ratios of water losses versus available lake water."

L514: Consider citing Spence et al., 2019: Spence, C., G. Ali, C.J. Oswald, and C. Wellen, 2019. An Application of the T-TEL Assessment Method to Evaluate Connectivity in a Lake-Dominated Watershed after Drought. JAWRA Journal of the American Water Resources Association 55:318–333.

We thank the reviewer for the suggestion. However, after reading the paper, we did not think the citation related enough to the argument we are trying to make at line 514. Therefore, the citation was not included.

L542: See comment above, please define hydraulic midline.

The midline was defined as "the boundary between the groundwater recharge and discharge areas".

L554: What do you mean by "lithology" here?

We replaced the word "lithology" with "surficial geology" for more clarity as suggested.

L650: While I appreciate the authors attempt at showing p-values here [which I would encourage them to keep], it seems like the p-value threshold here [p<0.05] has little relevance when comparing across metrics.

We changed "isotopic values are influenced by" to "isotopic values may be influenced by" to highlight the fact that the p-values are not significant in the strictest sense.

L673: What do the authors mean by "this"

We clarified that sentence in the manuscript.

L676: Sentence should be re-written [maybe into multiple sentences?] for clarity.

We clarified that sentence in the manuscript.

L685: Again, please rewrite and be more specific.

We clarified that sentence in the manuscript.

L741: It feels like the authors are overstepping their bounds here a bit. In my opinion, our role as scientist is not to tell policy makers and practioners what to do. More appropriately, I argue that our role is to provide information about how different actions will affect management outcomes. Thus, instead of saying "managers should implement policy XYZ," I would encourage authors to say something like "In order to obtain ABC outcome, policy makers should consider implementing practice XYZ because…."

We took the reviewer's remark into account and adapted the paragraph accordingly.

Note: we additionally took the liberty to 1) do some reordering in the reference citation in the manuscript, 2) to add a handful of new references, some of them recently published and 3) to check the consistency of the citations in the reference section.